# Development of high-order global dynamical core using discontinuous Galerkin method for atmospheric LES and proposal of test cases: SCALE-DG v0.8.0

Yuta Kawai[1] and Hirofumi Tomita[1,2]

[1]RIKEN Center for Computational Science, Kobe, Japan
[2]RIKEN Cluster for Pioneering Research, Wako, Japan

**Abstract.** Focusing on the future global atmospheric simulations with grid spacing of O(10–100 m), we developed a global nonhydrostatic atmospheric dynamical core with high-order accuracy by applying a discontinuous Galerkin method (DGM) for horizontal and vertical discretization. Furthermore, considering a global large-eddy simulation (LES), a Smagorinsky–Lilly turbulence model was introduced to the proposed global dynamical core in the DGM framework. By conducting several tests with various polynomial orders ($p$), the impact of high-order DGM on the accuracy of the numerical simulations of atmospheric flows was investigated. To show high-order numerical convergence, a few modifications were made in the experimental setup of existing test cases. In addition, we proposed an idealized test case to verify global LES models, which is a global extension of idealized planetary boundary layer (PBL) turbulence experiment performed in our previous studies. The error norms from the deterministic test cases, such as linear advection and gravity wave tests, showed an optimal convergence rate achieved by approximately $p + 1$-order spatial accuracy when the temporal and round-off errors were sufficiently small. In the climatic test cases, such as the Held-Suarez test, the kinetic energy spectra indicated the advantage of effective resolution when large polynomial orders were used. In the LES experiment, the global model provided a reasonable vertical structure of PBL and energy spectra because the results under shallow atmosphere approximation well reproduced those obtained in the plane computational domain.

## 1 Introduction

Recently developed supercomputers have enabled us to conduct high-resolution global atmospheric simulations using a sub-kilometer horizontal grid spacing. For example, Miyamoto et al. (2013) conducted a global simulation at a horizontal grid spacing of 870 m and discussed the numerical convergence of statistical properties of deep moist convections. In the near future, this continuous development of computer technology is expected to enable us to perform global simulations using O(10–100 m) grid spacing (Satoh et al., 2019), which begin to explicitly represent turbulence in the inertial sub-range. Then, large-eddy simulation (LES) is a promising strategy, since in LES, the turbulence in a spatial scale larger than a spatial filter is explicitly calculated, whereas the effect of turbulence in a smaller spatial scale is parameterized using eddy viscosity and diffusion terms. By explicitly representing the large-scale eddies in boundary layers and the low-level clouds such as shallow

cumuli, we expect to reduce a source of uncertainty associated with the parameterizations and improve representation of global radiation budget in a realistic Earth's atmosphere.

Considering future high-resolution atmospheric simulations such as global LES, we discussed the problem of low numerical accuracy of conventional atmospheric dynamical cores in Kawai and Tomita (2021, 2023) (hereafter referred to as KT2021 and KT2023, respectively). To perform LES precisely, we must ensure that the discretization errors do not dominate over sub-grid scale (SGS) terms of turbulent models. Otherwise, the SGS terms might lose their physical significance. KT2021 investigated the order of accuracy necessary for advection schemes in the framework of conventional grid-point methods. In particular, the study derived two ratios associated with numerical diffusion and numerical dispersion: the ratio of decay time with the SGS terms to that of the numerical diffusion error terms and that of phase speed due to the error in advection terms to that of the SGS terms. Moreover, we pointed out that the advection scheme requires at least seventh- or eighth-order accuracy to ensure that both ratios are less than $10^{-1}$ at wavelengths longer than eight grid lengths for grid spacing simulations of O(10) m. However, in conventional grid-point methods, the required stencil grows larger as the order of accuracy increases. This can degrade computational performance of the high-order methods in recent massively parallel computers. Thus, we focused on applicability of the discontinuous Galerkin method (DGM). At element boundaries, the representation of flow field is allowed to be discontinuous, and the flux shared by two neighboring elements is calculated using approximated Riemann solvers. These computational features provide a straightforward strategy to achieve high-order discretization and computational compactness. KT2023 extended the discussion presented in KT2021 to the DGM framework and investigated a polynomial order necessary for precisely conducting LES. It indicated that the polynomial order needs to be higher than or equal to four when the upwind numerical fluxes and sufficiently scale-selective modal filters are used to ensure numerical stability.

The basis of the state-of-the-art global nonhydrostatic atmosphere dynamical cores was mainly developed during 2000's–2010's. In these dynamical cores, low-order grid-point methods are often adopted. For example, the Non-hydrostatic ICosahedral Atmospheric Model (NICAM; Tomita and Satoh, 2004; Satoh et al., 2014), the Model for Prediction Across Scales (MPAS; Skamarock et al., 2012), and the ICOsahedral Non-hydrostatic model (ICON; Zängl et al., 2015) are based on either a totally first- or totally second-order scheme. The discretization accuracy has not always been a primary factor in the performance of atmospheric models because physical processes have various uncertain parameters. In situations where the grid spacing is coarser than the gray zone of turbulence, the totally second-order scheme may be appropriate in terms of computational cost and numerical robustness. However, as described above, it is crucial to increase numerical accuracy to precisely conduct the atmospheric LES. Furthermore, even in spatial resolutions coarser than that required by LES, it is undesirable that the shortest wavelength fully resolved by discretization methods, so called the effective resolution, is significantly different from the grid spacing in terms of the physics-dynamics coupling. The low-order spatial scheme typically leads to significant discretization errors at wavelengths shorter than eight grid lengths (Kent et al., 2014). To decrease the gap between the effective resolution and grid scale, it is sensible to use high-order discretization methods in addition to designing better numerical filters for controlling the effective resolution.

Constructing high-order grid-point methods tends to be more complex for spherical geometries than in plane domains with structured grids. To achieve the high-order discretization accuracy horizontally, a conventional straightforward approach is

the spectral transformation method based on the spherical harmonics expansion. This approach provides desirable accuracy for numerical solutions in the wavelength range up to the truncated wave number while avoiding the problem of restrictive timestep near the poles due to the convergence of meridians. However, high-resolution global simulations can suffer from large costs of data communication between all computational nodes in massively parallel supercomputers. On the other hand, some researchers have successfully developed global atmospheric dynamical cores based on high-order grid-point and element-based methods. The essence of the numerical methods can be found in horizontal discretization of the global shallow water equations; For example, Ullrich et al. (2010) for a high-order finite volume method (FVM), while Nair et al. (2005a) and Ullrich (2014) for high-order element-based methods. Ullrich and Jablonowski (2012b) proposed a global nonhydrostatic dynamical core (MCore) based on an FVM with a fourth-order horizontal reconstruction strategy. The Tempest model (Guerra and Ullrich, 2016) uses a high-order spectral element method (SEM) horizontally. In the Nonhydrostatic Unified Model of the Atmosphere (NUMA; Kelly and Giraldo, 2012; Giraldo et al., 2013), which is applicable for both limited-area and global atmospheric simulations, the continuous and discontinuous Galerkin methods are adopted for the spatial discretization. The numerical method prototype used in NUMA is utilized and extended to a global spectral-element dynamical core in the Navy Environmental Prediction System Utilizing a Nonhydrostatic Engine (NEPTUNE) for both horizontal and vertical discretization (e.g., Zaron et al., 2022). SEM is also used for the nonhydrostatic High Order Method Modeling Environment (HOMME-NH; Dennis et al., 2005, 2012; Taylor et al., 2020) included in the Energy Exascale Earth System Model (E3SM), and for the nonhydrostatic dynamical core in the Korean Integrated Model (KIM) system (Hong et al., 2018). The ClimateMachine uses a nodal discontinuous Galerkin method for both horizontal and vertical discretization. The corresponding regional dynamical core is described in Sridhar et al. (2022). In the case of classical high-order element-based methods, it is cumbersome to control the numerical instability caused by the aliasing errors with nonlinear terms (Winters et al., 2018). To overcome this problem, a split form nodal DGM (e.g., Gassner et al., 2016) is a theoretical and computationally efficient approach. A similar method was successfully applied to a global shallow water model in Ricardo et al. (2024) and to a global nonhydrostatic dynamical core for horizontal and vertical discretization in Souza et al. (2023). While conventional dynamical cores adopt a vertical discretization based on a low-order finite difference method or FVM, some studies investigated the potential for the use of high-order vertical discretization (e.g., Guerra and Ullrich, 2016; Yi and Giraldo, 2020; Ishioka et al., 2022). For example, Guerra and Ullrich (2016) introduced an arbitrary-order vertical discretization using a staggered nodal FEM. They reported that high-order vertical discretization improves the representation of vertical dynamics at a relatively low vertical resolution.

By building on progresses from the previous studies showing the applicability of the element-based methods to atmospheric flow simulations, the current study attempted to develop a high-order global dynamical core using a nodal DGM both horizontally and vertically for future global atmospheric simulations with O(10–100 m) grid spacing. For a quasi-uniform spherical grid, a cubed-sphere projection was adopted. Moreover, a terrain-following coordinate was used to treat the topography. Although the numerical methods used in our dynamical core are similar to those used in previous studies that developed global DG dynamical cores such as NUMA and ClimateMachine, we consider that the following points are the unique contributions of the current study: 1) Considering global LES, we formulated SGS eddy viscous and diffusion terms with a Smagorinsky–Lilly type turbulent model in the DGM framework on the cubed-sphere coordinates. A discretization strategy for the scalar

Laplacian operator on the cubed-sphere coordinates with DGM is reported in previous studies (e.g., Nair, 2009). However,

they did not treat the vector Laplacian operator for the vector quantities (for example, momentum). This might be because the rigorous form is so complex that it may not be worth the computational cost required for numerical stabilization. On the other hand, Ullrich (2014) presented a discretization strategy for the vector Laplacian operator with the continuous and discontinuous Galerkin methods. This approach can distinguish the divergence damping and vorticity damping with constant viscous coefficients. Guba et al. (2014) proposed a strategy of hyperviscosity with variable viscous coefficients in SEM where

the vector Laplacian operator is applied to the Cartesian component of the vector fields. However, we did not directly use these approaches to the vector Laplacian when introducing the turbulent model. This is because we need to treat eddy viscous and diffusion coefficients dependent on local wind shear and stratification. In addition, we consider that the vector Laplacian operator applied to the vector component on the cubed-sphere coordinates can be convenient for straightforward distinction between horizontal and vertical directions. Using tensor analysis, we systematically derived the eddy viscosity and diffusion

terms. Then, we represented the corresponding semi-discretized equations using DGM. Subsequently, a quantitative check was performed by conducting an idealized LES experiment of planetary boundary layer turbulence. In particular, we extended a numerical experiment of idealized planetary boundary turbulence used in regional plane models (KT2021 and KT2023) to spherical geometry by slightly changing the initial condition. 2) We modified experimental settings of idealized test cases to demonstrate the numerical convergence with high-order dynamical cores. When using totally second-order dynamical cores,

relatively large discretization errors may occur, which can overshadow the problems of ill-posed experimental settings. Fast numerical convergence achieved using high-order schemes is expected to enable detection of such problems in standard tests. Even when research interests do not include the dynamics, we consider an evaluation framework using high-order dynamical core to be useful. For example, when new physical models are included, the physical performance can be directly evaluated by reducing numerical errors with the dynamical processes. 3) We attempted quantitative evaluations in a series of test cases for

the global dynamical cores to investigate the impact of high-order DGM on the numerical accuracy of atmospheric flow simulations. Although the numerical convergence characteristics of DGM was closely investigated in the case of regional dynamical cores (e.g., Giraldo and Restelli, 2008; Brdar et al., 2013; Blaise et al., 2016), few studies have been conducted to demonstrate it for global nonhydrostatic dynamical cores.

The rest of this paper is organized as follows: In Sect. 2, the governing equations using the general curvilinear coordinates

are formulated. Then, we introduce a cubed-sphere projection and a general vertical coordinate. We represent eddy viscous and diffusion terms associated with the turbulent model in the general curvilinear coordinates. Furthermore, we explain the spatial and temporal discretization adopted for the governing equations. In Sect. 3, we verify the proposed dynamical core through several idealized numerical experiments. Finally, the findings of this study and our future plans are summarized.

## 2 Model Description

 ### 2.1 Governing Equations

As governing equations for dry atmospheric flows, we used the three-dimensional, fully compressible nonhydrostatic equations based on the flux form (e.g., Ullrich and Jablonowski, 2012b). Following Li et al. (2020), a non-orthogonal curvilinear horizontal coordinate $(\xi, \eta)$ was introduced. Subsequently, a general vertical coordinate $\zeta$ was introduced. For the horizontal coordinate transformation, the contravariant form of the metric tensor is represented by $G_h^{ij}$ for $i, j = 1, 2$. A three-dimensional metric tensor with the horizontal coordinate transformation is defined as

$$G^{ij} = \begin{pmatrix} G_h^{11} & G_h^{12} & 0 \\ G_h^{21} & G_h^{22} & 0 \\ 0 & 0 & 1 \end{pmatrix}. \tag{1}$$

The horizontal Jacobian is defined as $\sqrt{G_h} = |G_h^{ij}|^{-\frac{1}{2}}$. For the vertical coordinate transformation, the metric tensor is defined as $G_v^{13} = \partial \zeta / \partial \xi, G_v^{23} = \partial \zeta / \partial \eta$ and the vertical Jacobian is defined as $\sqrt{G_v} = \partial z / \partial \zeta$. The vertical velocity in the transformed vertical coordinate can be written using contravariant components of wind vector $(u^\xi, u^\eta, u^\zeta)$ as

$$\widetilde{u^\zeta} \equiv \frac{d\zeta}{dt} = \frac{1}{\sqrt{G_v}} \left( u^\zeta + \sqrt{G_v} G_v^{13} u^\xi + \sqrt{G_v} G_v^{23} u^\eta \right). \tag{2}$$

The final Jacobian composed of horizontal and vertical coordinate transformations can be represented as $\sqrt{G} = \sqrt{G_h}\sqrt{G_v}$. Hereafter, to briefly describe the formulations, the coordinate variables are sometimes expressed using $(\xi^1, \xi^2, \xi^3) = (\xi, \eta, \zeta)$. In addition, the Einstein summation notation will be applied for repeated indices when representing the geometric relations.

The compact form of the governing equations can be written as

$$\frac{\partial q}{\partial t} + \frac{\partial \left[ f(q) + f_{\text{SGS}}(q, \nabla q) \right]}{\partial \xi} + \frac{\partial \left[ g(q) + g_{\text{SGS}}(q, \nabla q) \right]}{\partial \eta} + \frac{\partial \left[ h(q) + h_{\text{SGS}}(q, \nabla q) \right]}{\partial \zeta}$$
$$= S(q) + S_{\text{SGS}}(q, \nabla q). \tag{3}$$

Here, $q$ is the solution vector defined as

$$q = \left( \sqrt{G}\rho', \sqrt{G}\rho u^\xi, \sqrt{G}\rho u^\eta, \sqrt{G}\rho u^\zeta, \sqrt{G}(\rho\theta)' \right)^T, \tag{4}$$

where $\rho, \theta$ denote the density and potential temperature, respectively. To accurately treat nearly balanced flows, the density $\rho$ and pressure $p$ (thus $\rho\theta$) are decomposed as $q(\xi, \eta, \zeta, t) = q_{\text{hyd}}(\xi, \eta, \zeta) + q'(\xi, \eta, \zeta, t)$, where $q_{\text{hyd}}$ denotes a variable satisfying the hydrostatic balance and $q'$ denotes the deviation. In Eq. (3), $f(q)$, $g(q)$, and $h(q)$ are inviscid fluxes in the $\xi$, $\eta$, and $\zeta$ directions, respectively. The horizontal inviscid fluxes are represented as

$$f(q) = \begin{pmatrix} \sqrt{G}\rho u^\xi \\ \sqrt{G}(\rho u^\xi u^\xi + G_h^{11} p') \\ \sqrt{G}(\rho u^\eta u^\xi + G_h^{21} p') \\ \sqrt{G}\rho u^\zeta u^\xi \\ \sqrt{G}\rho\theta u^\xi \end{pmatrix}, \quad g(q) = \begin{pmatrix} \sqrt{G}\rho u^\eta \\ \sqrt{G}(\rho u^\xi u^\eta + G_h^{12} p') \\ \sqrt{G}(\rho u^\eta u^\eta + G_h^{22} p') \\ \sqrt{G}\rho u^\zeta u^\eta \\ \sqrt{G}\rho\theta u^\eta \end{pmatrix}, \tag{5}$$

and the vertical inviscid fluxes are represented as

$$
\quad \boldsymbol{h}(\boldsymbol{q}) = \begin{pmatrix} \sqrt{G}\rho\widetilde{u^\zeta} \\ \sqrt{G}[\rho u^\xi\widetilde{u^\zeta} + (G_v^{13}G_h^{11} + G_v^{23}G_h^{12})p'] \\ \sqrt{G}[\rho u^\eta\widetilde{u^\zeta} + (G_v^{13}G_h^{21} + G_v^{23}G_h^{22})p'] \\ \sqrt{G}\rho u^\zeta\widetilde{u^\zeta} + \sqrt{G}_h p' \\ \sqrt{G}\rho\theta\widetilde{u^\zeta} \end{pmatrix}. \tag{6}
$$

Furthermore, $\boldsymbol{S}(\boldsymbol{q})$ represents the source terms as

$$
\boldsymbol{S}(\boldsymbol{q}) = \begin{pmatrix} 0 \\ \sqrt{G}(F_H^1 + F_M^1 + F_C^1) \\ \sqrt{G}(F_H^2 + F_M^2 + F_C^2) \\ \sqrt{G}(F_{\mathrm{buo}} + F_C^3) \\ 0 \end{pmatrix}, \tag{7}
$$

where $F_H^i$ for $i = 1, 2$ are the horizontal pressure gradient terms with hydrostatic balance and can be expressed as

$$
F_H^i = -\frac{G_h^{im'}}{\sqrt{G_v}}\left[ \frac{\partial(\sqrt{G_v}p_{\mathrm{hyd}})}{\partial\xi^{m'}} + \frac{\partial(G_v^{m'3}\sqrt{G_v}p_{\mathrm{hyd}})}{\partial\xi^3} \right]. \tag{8}
$$

Here, note that $m' = 1, 2$. $F_M^i = -\Gamma_{ml}^i(\rho u^m u^l + G^{ml}p')$ are the source terms due to the horizontal curvilinear coordinate, where $m, l$ take values of $1, 2, 3$ and $\Gamma_{ml}^i$ is the Christoffel symbol of the second kind, which means the spatial variation of the basis vector. $F_C^i = -G^{im}\epsilon_{jml}2\Omega^m\rho u^l$ are the Coriolis terms, where $\epsilon_{jkl}$ is the three rank Levi–Civita tensor and $\Omega^m$ are the components of angular velocity vector. $F_{\mathrm{buo}} = -\rho'(a/r)^2 g$ is the buoyancy term, where $r$ is the radial coordinate, $a$ is the planetary radius, and $g$ is the standard gravitational acceleration. To close the equation systems, the pressure $p$ is calculated
using the state equation for the ideal gas as

$$
p = P_0\left( \frac{R}{P_0}\rho\theta \right)^{\frac{C_p}{C_v}}, \tag{9}
$$

where $P_0$ is a constant pressure, $R$ is the gas constant, and $C_v$ and $C_p$ are the specific heat at constant volume and constant pressure, respectively. The actual values for these constants are provided in Table 1. In Eq. (3), the terms with subscript SGS are associated with a turbulent model; $\boldsymbol{f}_{\mathrm{SGS}}(\boldsymbol{q}, \nabla\boldsymbol{q})$, $\boldsymbol{g}_{\mathrm{SGS}}(\boldsymbol{q}, \nabla\boldsymbol{q})$, and $\boldsymbol{h}_{\mathrm{SGS}}(\boldsymbol{q}, \nabla\boldsymbol{q})$ are the parameterized eddy fluxes while
$\boldsymbol{S}_{\mathrm{SGS}}(\boldsymbol{q}, \nabla\boldsymbol{q})$ are the source terms with the curvilinear coordinates. The terms associated with the turbulent model are detailed in Sect. 2.2.

As a horizontal curvilinear coordinate, we adopted an equiangular gnomonic cubed-sphere projection (Sadourny, 1972; Ronchi et al., 1996) to map a cube onto a sphere. Compared to a conformal projection (Rančić et al., 1996), we preferred this projection to generate more uniform grids in high spatial resolutions, although the non-orthogonal basis need to be treated. In
each panel of the cube, a local coordinate using the central angles $(\alpha, \beta)$ ($\in [-\pi/4, \pi/4]$) was introduced and related to the

horizontal coordinates $(\xi, \eta)$ by $\xi = \alpha, \eta = \beta$. Based on the derivation with the coordinate transformation in previous studies (e.g., Nair et al., 2005b; Ullrich et al., 2012; Li et al., 2020), the horizontal contravariant metric tensor and the horizontal Jacobian for the equiangular gnomonic cubed-sphere projection can be written as, respectively,

$$G_c^{ij} = \frac{\delta^2}{r^2(1+X^2)(1+Y^2)}\begin{pmatrix} 1+Y^2 & XY \\ XY & 1+X^2 \end{pmatrix}, \quad \sqrt{G_c} = \frac{r^2(1+X^2)(1+Y^2)}{\delta^3}, \tag{10}$$

where $X = \tan\alpha$, $Y = \tan\beta$, $\delta = \sqrt{1+X^2+Y^2}$, and $r$ is the radial coordinate. The Christoffel symbol of the second kind $\Gamma_{ml}^i$ is represented as

$$\Gamma_{ml}^1 = \begin{pmatrix} \dfrac{2XY^2}{\delta^2} & \dfrac{-Y(1+Y^2)}{\delta^2} & \dfrac{\delta_S}{r} \\ \dfrac{-Y(1+Y^2)}{\delta^2} & 0 & 0 \\ \dfrac{\delta_S}{r} & 0 & 0 \end{pmatrix}, \quad \Gamma_{ml}^2 = \begin{pmatrix} 0 & \dfrac{-X(1+X^2)}{\delta^2} & 0 \\ \dfrac{-X(1+X^2)}{\delta^2} & \dfrac{2X^2Y}{\delta^2} & \dfrac{\delta_S}{r} \\ 0 & \dfrac{\delta_S}{r} & 0 \end{pmatrix},$$

$$\Gamma_{ml}^3 = \delta_S \frac{r(1+X^2)(1+Y^2)}{\delta^4}\begin{pmatrix} -(1+X^2) & XY & 0 \\ XY & -(1+Y^2) & 0 \\ 0 & 0 & 0 \end{pmatrix}, \tag{11}$$

where $\delta_S$ is a switch for shallow atmosphere approximation. The components of angular velocity vector included in the Coriolis terms $F_C^i$ are given as

$$\Omega^1 = 0, \quad \Omega^2 = \delta_S \frac{\omega\delta}{r(1+Y^2)}, \quad \Omega^3 = \omega\frac{Y}{\delta}, \quad \text{for the equatorial panels,}$$

$$\Omega^1 = -\delta_S \frac{s\omega X\delta}{r(1+X^2)}, \quad \Omega^2 = -\delta_S \frac{s\omega Y\delta}{r(1+Y^2)}, \quad \Omega^3 = \frac{s\omega}{\delta}, \quad \text{for the polar panels,} \tag{12}$$

where $\omega$ is the angular velocity of the planet and an index $s$ has a value of 1 and -1 for the Northern and Southern polar panels, respectively. In the numerical experiments in Sect. 3, the shallow atmosphere approximation was applied. Then, $r$ and $\delta_S$ are treated as follows: the radial coordinate $r$ in Eqs. (10)–(12) and the buoyancy term in Eq. (6) is replaced by the planetary radius $a$. In Eqs. (11) and (12), the terms with $\delta_S$ are ignored. In addition, the pressure contribution in $F_M^i$ disappears because the relation of $\Gamma_{ml}^i G^{ml} = 0$ is satisfied in the shallow atmosphere approximation.

To treat the topography, the traditional terrain-following coordinate (Phillips, 1957; Gal-Chen and Somerville, 1975) was adopted as a general vertical coordinate. The vertical coordinate transformation can be expressed as

$$\zeta = z_T \frac{z-h}{z_T-h}, \tag{13}$$

where $z$ is the height coordinate, $z_T$ is the top height of computational domain (we assume it is a constant value), and $h$ is the surface height. The corresponding Jacobian and metric tensor can be represented as

$$\sqrt{G_v} = 1 - \frac{h}{z_T}, \quad \sqrt{G_v}G_v^{13} = \left(\frac{\zeta}{z_T}-1\right)\frac{\partial h}{\partial\xi}, \quad \sqrt{G_v}G_v^{23} = \left(\frac{\zeta}{z_T}-1\right)\frac{\partial h}{\partial\eta}, \tag{14}$$

respectively.

## 2.2 Formulation of eddy viscous and diffusion terms in general curvilinear coordinates

Considering global LES in future high-resolution simulations, this subsection describes eddy viscous and diffusion terms in the general curvilinear coordinates. We utilized on a Smagorinsky–Lilly type model (Smagorinsky, 1963; Lilly, 1962) that considered the stratification effect (Brown et al., 1994). This turbulent model was also used in KT2021 and KT2023. As a spatial filter, the Favre-filtering (Favre, 1983) was used. We do not explicitly denote the symbol representing the spatial filter because the filtering approach is essentially the same as that explained in Appendix A of KT2023. The difficulties in the derivation of viscous and diffusion terms are caused by the gradient of vector quantities and the spatial divergence with the non-orthogonal basis because the manipulations grow increasingly complex. However, previous studies that utilized tensor analysis help us provide a systematic derivation (e.g., Ullrich, 2014; Rančić et al., 2017). In the absence of a vertical coordinate transformation, the parameterized fluxes with the turbulent model can be represented in the general curvilinear coordinates as

$$
f_{\text{SGS}}(q, \nabla q) = \begin{pmatrix} 0 \\ -\sqrt{G}\rho\tau^{11} \\ -\sqrt{G}\rho\tau^{12} \\ -\sqrt{G}\rho\tau^{13} \\ -\sqrt{G}\rho\tau^1_* \end{pmatrix}, \quad
g_{\text{SGS}}(q, \nabla q) = \begin{pmatrix} 0 \\ -\sqrt{G}\rho\tau^{21} \\ -\sqrt{G}\rho\tau^{22} \\ -\sqrt{G}\rho\tau^{23} \\ -\sqrt{G}\rho\tau^2_* \end{pmatrix}, \quad
h_{\text{SGS}}(q, \nabla q) = \begin{pmatrix} 0 \\ -\sqrt{G}\rho\tau^{31} \\ -\sqrt{G}\rho\tau^{32} \\ -\sqrt{G}\rho\tau^{33} \\ -\sqrt{G}\rho\tau^3_* \end{pmatrix},
\tag{15}
$$

and the source term can be given by

$$
S_{\text{SGS}}(q, \nabla q) = \begin{pmatrix} 0 \\ -\sqrt{G}\Gamma^1_{ml}\rho\tau^{ml} \\ -\sqrt{G}\Gamma^2_{ml}\rho\tau^{ml} \\ -\sqrt{G}\Gamma^3_{ml}\rho\tau^{ml} \\ 0 \end{pmatrix}.
\tag{16}
$$

In the equations, $\tau^{ij}$ is the contravariant components of parameterized eddy viscous flux tensor ($i = 1, 2, 3$ and $j = 1, 2, 3$) and can be written as

$$
\tau^{ij} = -2\nu_{\text{SGS}}\left(S^{ij} - \frac{G^{ij}}{3}D\right) - \frac{2}{3}G^{ij}K_{\text{SGS}},
\tag{17}
$$

where $S^{ij}$ is the strain velocity tensor, $\nu_{\text{SGS}}$ is the eddy viscosity, $D$ is the divergence of the three-dimensional velocity, and $K_{\text{SGS}}$ is the SGS kinetic energy. The strain velocity tensor is represented as

$$
S^{ij} = \frac{1}{2}\left(G^{im}\frac{\partial u^j_{,m}}{\partial \xi^m} + G^{jm}\frac{\partial u^i_{,m}}{\partial \xi^m}\right),
\tag{18}
$$

using the covariant derivative of the contravariant velocity component

$$
u^i_{,j} = \frac{\partial u^i}{\partial \xi^j} + u^m\Gamma^i_{jm}.
\tag{19}
$$

The eddy viscosity is written as

$$\nu_{\mathrm{SGS}} = C_s \Delta_{\mathrm{SGS}} |S|, \tag{20}$$

where $C_s$, $\Delta_{\mathrm{SGS}}$, and $|S|$ represent the Smagorinsky constant, the filter length, and the norm of strain tensor defined as $\sqrt{2G_{im}G_{jn}S^{ij}S^{mn}}$, respectively. The parameterized eddy diffusive flux can be written as

$$\tau_*^i = -\nu_{\mathrm{SGS}}^* G^{ij} \frac{\partial \theta}{\partial \xi^j}, \tag{21}$$

where $\nu_{\mathrm{SGS}}^*$ is the eddy diffusion coefficient. For further details of the turbulent model, refer to Sect. 2.2 of Nishizawa et al. (2015).

## 2.3 Spatial discretization

We performed the spatial discretization for Eq. (3) based on a nodal DGM (e.g., Hesthaven and Warburton, 2007). In each cubed-sphere panel, the three-dimensional computational domain $\Omega$ was divided using non-overlapping hexahedral elements. To relate the coordinates $(\xi^1, \xi^2, \xi^3) = (\alpha, \beta, \zeta)$ with the local coordinates $\tilde{x} \equiv (\tilde{x}^1, \tilde{x}^2, \tilde{x}^3)$ in a reference element $\Omega_e$, we adopted a linear mapping defined as

$$\tilde{x}^i = 2 \frac{\xi^i - \xi_e^i}{h_e^i}, \tag{22}$$

where $\xi_e^i$ and $h_e^i$ represent the center position and width of the element, respectively, in the $\xi^i$-direction. By equally dividing the $(\alpha, \beta)$ plane, we generated a horizontal mesh including $N_{e,h} \times N_{e,h}$ finite elements. The center horizontal position of $(i', j')$-th element can be written as

$$\alpha_{i'} = -\frac{\pi}{4} + \frac{\pi}{2N_{e,h}}\left(i' - \frac{1}{2}\right), \quad \beta_{j'} = -\frac{\pi}{4} + \frac{\pi}{2N_{e,h}}\left(j' - \frac{1}{2}\right). \tag{23}$$

Using the tensor-product of one-dimensional Lagrange polynomials $l_m(\tilde{x}) = l_{m_1}(\tilde{x}^1)l_{m_2}(\tilde{x}^2)l_{m_3}(\tilde{x}^3)$, a local approximated solution within each element $\Omega_e$ can be represented as

$$q^e|_{\Omega_e}(\tilde{x}, t) = \sum_{m_1=1}^{p+1} \sum_{m_2=1}^{p+1} \sum_{m_3=1}^{p+1} Q_{m_1,m_2,m_3}^e(t) \, l_{m_1}(\tilde{x}^1)l_{m_2}(\tilde{x}^2)l_{m_3}(\tilde{x}^3), \tag{24}$$

In Eq. (24), the coefficients $Q_{m_1,m_2,m_3}^e$ are the unknown degrees of freedom (DOF) and $p$ is the polynomial order. In this study, the Legendre–Gauss–Lobatto (LGL) points were used for interpolation and integration nodes. We defined a representative grid spacing at the equator which approximately corresponds to that in the grid-point methods as

$$\Delta_{h,\mathrm{eq}} = \frac{\pi a}{2N_{e,h}(p+1)}. \tag{25}$$

Similar to the case of the horizontal direction, we defined a representative vertical grid spacing $\Delta_v$. For the uniform vertical element size, $\Delta_v = z_T/(N_{e,v}(p+1))$ where $N_{e,v}$ is the number of vertical elements. Hereafter, we simply refer to $\Delta_{h,\mathrm{eq}}$ and $\Delta_v$ as the horizontal and vertical grid spacing, respectively.

By applying the Galerkin approximation to Eq. (3), a strong form of the semi-discretized equations can be obtained as

$$\frac{d}{dt}\int_{\Omega_e} \boldsymbol{q}^e(\tilde{\boldsymbol{x}},t)\, l_m(\tilde{\boldsymbol{x}})\, J^E\, d\tilde{\boldsymbol{x}} = -\sum_{j=1}^{3}\int_{\Omega_e} \frac{\partial \boldsymbol{F}_j(\boldsymbol{q}^e,\boldsymbol{G})}{\partial \xi^j} l_m(\tilde{\boldsymbol{x}})\, J^E\, d\tilde{\boldsymbol{x}}$$

$$-\int_{\partial\Omega_e} \left[\hat{\boldsymbol{F}}(\boldsymbol{q}^e,\boldsymbol{G}) - \boldsymbol{F}(\boldsymbol{q}^e,\boldsymbol{G})\right]\cdot \boldsymbol{n}\, l_m(\tilde{\boldsymbol{x}})\, J^{\partial E}\, dS$$

$$+\int_{\Omega_e} \left[\boldsymbol{S}(\boldsymbol{q}^e) + \boldsymbol{S}_{\mathrm{SGS}}(\boldsymbol{q}^e,\boldsymbol{G})\right] l_m(\tilde{\boldsymbol{x}})\, J^E\, d\tilde{\boldsymbol{x}}, \tag{26}$$

where $(\boldsymbol{F}_1, \boldsymbol{F}_2, \boldsymbol{F}_3) = (\boldsymbol{f} + \boldsymbol{f}_{\mathrm{SGS}}, \boldsymbol{g} + \boldsymbol{g}_{\mathrm{SGS}}, \boldsymbol{h} + \boldsymbol{h}_{\mathrm{SGS}})$ is the flux vector tensor, $\hat{\boldsymbol{F}}$ is the numerical flux at the element boundary $\partial\Omega_E$, and $\boldsymbol{n}$ is the outward unit vector normal to $\partial\Omega_E$; In the volume and surface integrals, $J^E$ and $J^{\partial E}$ represent the transformation Jacobian with the general curvilinear coordinates and local coordinates within each element. Note that, because of the linear mapping in Eq. (22), the associated geometric factors such as $J^E$ and $J^{\partial E}$ have constant values when the volume and surface integrals are calculated. For the turbulent model, we need to evaluate the eddy viscous flux tensor and diffusion flux, which include a few gradient terms with quantities such as $\chi = (u^\xi, u^\eta, u^\zeta, \theta)$, denoted by $\boldsymbol{G} = (\partial\chi/\partial\xi^1, \partial\chi/\partial\xi^2, \partial\chi/\partial\xi^3)$ in Eq. (26). The gradient discretization in the $\xi^j$-direction is given by

$$\int_{\Omega_e} \rho\, \boldsymbol{G}_j l_m(\tilde{\boldsymbol{x}})\, J^E\, d\tilde{\boldsymbol{x}} = \int_{\Omega_e} \left[\frac{\partial \rho^e \chi^e}{\partial \xi^j} - \chi^e \left(\frac{\partial \rho}{\partial \xi^j}\right)^e\right] l_m(\tilde{\boldsymbol{x}})\, J^E\, d\tilde{\boldsymbol{x}}$$

$$+\int_{\partial\Omega_e} (\widehat{\rho\chi} - \rho^e \chi^e)\, \boldsymbol{n}_{\tilde{x}^j}\cdot \boldsymbol{n}\, l_m(\tilde{\boldsymbol{x}})\, J^{\partial E}\, dS, \tag{27}$$

where $\boldsymbol{n}_{\tilde{x}^j}$ is the unit vector in the $\tilde{x}^j$-direction and the density gradient is calculated by

$$\int_{\Omega_e} \left(\frac{\partial \rho}{\partial \xi^j}\right)^e l_m(\tilde{\boldsymbol{x}})\, J^E\, d\tilde{\boldsymbol{x}} = \int_{\Omega_e} \frac{\partial \rho^e}{\partial \xi^j} l_m(\tilde{\boldsymbol{x}})\, J^E\, d\tilde{\boldsymbol{x}} + \int_{\partial\Omega_e} (\hat{\rho} - \rho^e)\boldsymbol{n}_{\tilde{x}^j}\cdot \boldsymbol{n}\, l_m(\tilde{\boldsymbol{x}})\, J^{\partial E}\, dS. \tag{28}$$

For the numerical flux of the inviscid terms, the Rusanov flux (Rusanov, 1961) was used as a simple choice of the approximated Riemann solvers. Its numerical dissipation is provided based on the maximum absolute eigenvalue of the Jacobian matrix at the left and right sides of the element boundary. Previous studies (e.g., Li et al., 2020) formulated the Rusanov flux taken into account the horizontal and vertical coordinate transformations as

$$\hat{\boldsymbol{F}}_{\mathrm{invis}} = \frac{1}{2}\left\{\left[\boldsymbol{F}_{\mathrm{invis}}(\boldsymbol{q}^+) + \boldsymbol{F}_{\mathrm{invis}}(\boldsymbol{q}^-)\right]\cdot \boldsymbol{n} - \lambda_{\max}\left[\boldsymbol{q}^+ - \boldsymbol{q}^-\right]\right\}, \tag{29}$$

where $\lambda_{\max}$ is the maximum of the absolute value of eigenvalues of the flux Jacobian in the direction $\boldsymbol{n}$, and $\boldsymbol{q}^-$ and $\boldsymbol{q}^+$ represent the interior and exterior values at $\partial\Omega_e$. At the element boundaries in the horizontal directions ($\xi$ and $\eta$), $\lambda_{\max}$ can be represented as

$$\lambda_{\max,\xi} = |u^\xi| + \sqrt{G_c^{11}}c_s, \quad \lambda_{\max,\eta} = |u^\eta| + \sqrt{G_c^{22}}c_s, \tag{30}$$

where $c_s = [(C_p/C_v)RT]^{1/2}$ is the speed of sound wave. For the vertical direction $\zeta$, $\lambda_{\max}$ can be represented as

$$\lambda_{\max,\zeta} = \left|\widetilde{u^\zeta}\right| + \left[1/\sqrt{G_v} + G_v^{13}G_X + G_v^{23}G_Y\right]^{1/2} c_s, \tag{31}$$

where $G_X = G_v^{13}G_c^{11} + G_v^{23}G_c^{12}$ and $G_Y = G_v^{13}G_c^{21} + G_v^{23}G_c^{22}$. The central flux was adopted as the numerical flux of the gradient $G$ and the SGS fluxes ($f_{\mathrm{SGS}}, g_{\mathrm{SGS}}, h_{\mathrm{SGS}}$) with the turbulent model.

When the same nodes are used for interpolation and integration (i.e., collocation), a matrix form of Eqs. (26) and (27) can be obtained as

$$\frac{d\boldsymbol{q}^e}{dt} = -\sum_{j=1}^{3} d_j D_{\tilde{x}^j} \boldsymbol{F}_j(\boldsymbol{q}^e, \boldsymbol{G}) - \sum_{f=1}^{6} s_{\partial\Omega_{e,f}} L_{\partial\Omega_{e,f}} \left[\hat{\boldsymbol{F}}(\boldsymbol{q}^e, \boldsymbol{G}) - \boldsymbol{F}(\boldsymbol{q}^e, \boldsymbol{G})\right] \cdot \boldsymbol{n}$$
$$+ \boldsymbol{S}(\boldsymbol{q}^e) + \boldsymbol{S}_{\mathrm{SGS}}(\boldsymbol{q}^e, \boldsymbol{G}), \tag{32}$$

$$\rho\, G_j = d_j D_{\tilde{x}^j}(\rho^e \chi^e) - \chi^e \left(\frac{\partial\rho}{\partial\xi^j}\right)^e + \sum_{f'=1}^{2} s_{\partial\Omega_{e,f'}} L_{\partial\Omega_{e,f'}} (\widehat{\rho\chi} - \rho^e\chi^e)\, \boldsymbol{n}_{\tilde{x}^j} \cdot \boldsymbol{n}, \tag{33}$$

where $D_{\tilde{x}^j}$ represents the differential matrix for the $\tilde{x}^j$-direction; $L_{\partial\Omega_{e,f}}$ represents the lifting matrix with the surface integral for the $f$-th element surface, and $L_{\partial\Omega_{e,f'}}$ represents the same for the $f'$-th element surface in the gradient operator for the $\tilde{x}^j$-direction. The components of these matrices are given as

$$(D_{\tilde{x}^j})_{m,m'} = M^{-1} \int_{\Omega_e} l_m \frac{\partial l_{m'}}{\partial \tilde{x}_j}\, d\tilde{\boldsymbol{x}}, \quad (L_{\partial\Omega_e,j})_{m,m'} = M^{-1} \int_{\partial\Omega_{e,j}} l_m l_{m'}^{\partial\Omega_{e,j}}\, dS, \tag{34}$$

where $M$ denotes the mass matrix and is given by

$$M_{m,m'} = \int_{\Omega_e} l_m l_{m'}\, d\tilde{\boldsymbol{x}}. \tag{35}$$

The density gradient term is calculated as

$$\left(\frac{\partial\rho}{\partial\xi^j}\right)^e = d_j D_{\tilde{x}^j} \rho^e - \sum_{f'=1}^{2} s_{\partial\Omega_{e,f'}} L_{\partial\Omega_{e,f'}} (\widehat{\rho} - \rho^e)\, \boldsymbol{n}_{\tilde{x}^j} \cdot \boldsymbol{n}. \tag{36}$$

Note that, in Eqs. (32), (33), and (36), $d_j = \partial \tilde{x}^j/\partial\xi^j$ and $s_{\partial\Omega_e,f'} = J_{\partial\Omega_{e,f'}}/J^E$ are constant values in the volume and surface integrals, respectively. We changed the calculation method of mass and lifting matrices depending on temporal discretization; This is detailed in Sect. 2.4.

The balance between the pressure gradient and buoyancy terms should be carefully treated in the discrete momentum equation (e.g., Blaise et al., 2016; Orgis et al., 2017). In the above formulation, because a different discretization space is used between the terms, a numerical imbalance is possible and may cause spurious oscillations, which can destabilize the simulations. To avoid this incompatibility, the vertical polynomial order for the density in the buoyancy term was reduced by one following Blaise et al. (2016).

## 2.4 Temporal discretization

The semi-discretized equations in Eq. (26) can be represented as the following ordinary differential equation (ODE) system

$$\frac{d\boldsymbol{q}}{dt} = \mathcal{S}(\boldsymbol{q}, \nabla\boldsymbol{q}) + \mathcal{F}(\boldsymbol{q}, \nabla\boldsymbol{q}), \tag{37}$$

where $\mathcal{S}(\boldsymbol{q}, \nabla\boldsymbol{q})$ and $\mathcal{F}(\boldsymbol{q}, \nabla\boldsymbol{q})$ represent the tendencies with slow and fast contributions, respectively. This study adopted Runge–Kutta (RK) schemes to solve the ODE system from $t = n\Delta t$ to $t = (n+1)\Delta t$, where $\Delta t$ is the time step and $n$ is a natural number. In this subsection, we describe two approaches for temporal discretization, namely, horizontal explicit and vertical implicit (HEVI) and horizontal explicit and vertical explicit (HEVE) approaches.

We introduce two types of Courant number, which are used to explain timestep settings for the numerical experiments in Sect. 3. For the horizontal advection test, the advective Courant number associated with the horizontal wind is defined as $C_{r,\mathrm{adv}} = U_0 \Delta t / \Delta_{h,\mathrm{eq}}$, where $U_0$ is the representative wind speed. For other numerical experiments, the acoustic Courant number associated with the sound wave propagation is defined as $C_{r,c_s} = c_s \Delta t / \Delta$, where $\Delta$ is the grid spacing; In particular, for the HEVI approach, $\Delta = \Delta_{h,\mathrm{eq}}$.

### 2.4.1 HEVI approach

If the aspect ratio of horizontal grid spacing to its vertical counterpart is large, it is impractical to use fully explicit temporal schemes because the vertically propagating sound waves severely restrict the timestep. A strategy to avoid computational cost in such case is the HEVI approach. The terms corresponding to vertical dynamics with a fast time-scale are evaluated using an implicit temporal scheme, while the remaining terms are evaluated using an explicit temporal scheme. This procedure is regarded as a framework of implicit-explicit (IMEX) time integration scheme (Bao et al., 2015; Gardner et al., 2018). General formulation of IMEX RK scheme (e.g., Ascher et al., 1997) with $v$ stages can be represented as

$$\boldsymbol{q}^{(s)} = \boldsymbol{q}^n + \Delta t \sum_{s'=1}^{s-1} a_{ss'} \mathcal{S}(t + c_{s'}\Delta t, \boldsymbol{q}^{(s')}) + \Delta t \sum_{s'=1}^{s} \tilde{a}_{ss'} \mathcal{F}(t + \tilde{c}_{s'}\Delta t, \boldsymbol{q}^{(s')}) \quad \text{for} \quad s = 1, \ldots, v$$

$$\boldsymbol{q}^{n+1} = \boldsymbol{q}^n + \Delta t \sum_{s=1}^{v} b_s \mathcal{S}(t + c_s\Delta t, \boldsymbol{q}^{(s)}) + \Delta t \sum_{s=1}^{v} \tilde{b}_s \mathcal{F}(t + \tilde{c}_s\Delta t, \boldsymbol{q}^{(s)}), \tag{38}$$

where $a_{ss'}$, $b_s$, and $c_s$ define the explicit temporal integrator, while $\tilde{a}_{ss'}$, $\tilde{b}_s$, and $\tilde{c}_{s'}$ define the implicit temporal integrator; $c_s = \sum_{s'=1}^{s-1} a_{ss'}$ and $\tilde{c}_s = \sum_{s'=1}^{s-1} \tilde{a}_{ss'}$ represents time when slow and fast terms are evaluated, respectively. These coefficients are compactly represented using "double Butcher tableaux", as shown in Table 2. Note that, in the table of the explicit part, $\mathscr{A} = \{a_{ss'}\}$ with $a_{ss'} = 0$ for $s' \geq s$. On the other hand, for the implicit part, $\tilde{\mathscr{A}} = \{\tilde{a}_{ss'}\}$ with $\tilde{a}_{ss'} = 0$ for $s' > s$ in the case of the diagonally implicit RK scheme.

In this study, the terms associated with vertical mass flux, vertical pressure gradient, vertical flux of potential temperature, and buoyancy in Eqs. (3) were treated as fast terms, whereas the other terms were treated as slow terms. To minimize contaminating the spatial accuracy of high-order DGM by temporal errors present in low-order HEVI scheme, a third-order scheme proposed by Kennedy and Carpenter (2003) was adopted; it includes four explicit and three implicit evaluations. The corresponding

double Butcher tableaux are given in Table 2. In the implicit part of each stage, the corresponding nonlinear equation system is solved using Newton's method. In each iteration, the linearized equation system is solved. Obtaining accurate solutions of the nonlinear equation system generally requires numerous iterations. However, this study performed a single iteration in Newton's method (i.e., Rosenbrock approach), significantly reducing the computational cost. Similar approach has been used in previous studies (Ullrich and Jablonowski, 2012a). In the case of the collocation approach, because the horizontal dependency between all nodes within an element vanishes, the vertical implicit evaluation can be parallelly performed at each horizontal node.

For the case of HEVI, the volume and surface integrations in Eqs. (34) and (35) were evaluated using inexact integration with the LGL nodes. Consequently, $M$ and $L_{\partial \Omega_{e,3}}$ became diagonal matrices, which further simplified the matrix structure associated with the vertical spatial operator.

### 2.4.2 HEVE approach

When we consider a horizontal grid spacing with O(10 m) such as in LES, the ratio of horizontal to vertical grid spacing approaches unity. The advantages of HEVI approach decrease. Thus, it is suitable to adopt a fully explicit temporal approach, referred to as HEVE approach. In such cases, RK schemes with a strong stability preserving (SSP) property (Gottlieb et al., 2001) are often used in combination with DGM. Similar to KT2023, this study adopted a ten-stage RK scheme with the fourth-order accuracy proposed by Ketcheson (2008). The corresponding Butcher table is given in Table 3. When using the HEVE approach, entries of the matrices in Eqs. (34) and (35) were directly calculated following Sect. 3.2 in Hesthaven and Warburton (2007).

### 2.5 Modal filtering

For high-order DGM, numerical instability is likely to occur in advection-dominated flows because the numerical dissipations with the upwind numerical fluxes weaken. Furthermore, we adopted a collocation approach due to its computational efficiency, as described in Sect. 2.3. One drawback is that the aliasing errors with evaluations of the nonlinear terms can drive numerical instability. To suppress this numerical instability, a modal filter was used as an additional stabilization mechanism. The filter matrix for the three-dimensional problem can be obtained as

$$\mathscr{F} = V^{3D} C^{3D} V^{3D}, \tag{39}$$

where $V^{3D}$ represents the Vandermode matrix associated with the LGL interpolation nodes (in Eq. (24)) and $C^{3D}$ represents the diagonal cutoff matrix. The entries of $C^{3D}$ are defined as

$$C^{3D}_{(m_1,m_2,m_3),(m_1',m_2',m_3')} = \delta_{m_1,m_1'}\sigma^h_{m_1}\,\delta_{m_2,m_2'}\sigma^h_{m_2}\,\delta_{m_3,m_3'}\sigma^v_{m_3}, \tag{40}$$

where $\sigma^h_i$ and $\sigma^v_i$ represent the decay coefficient for the one-dimensional horizontal and vertical modes $i$, respectively. Based on Hesthaven and Warburton (2007), a typical choice of the coefficient for mode $i$ is provided with an exponential function as

$$\sigma_i = \begin{cases} 1 & \text{if } 0 \leq i \leq p_c \\ \exp\left[-\alpha_m \left(\dfrac{i-p_c}{p-p_c}\right)^{p_m}\right] & \text{if } p_c \leq i \leq p, \end{cases} \tag{41}$$

where $p_c$, $p_m$, and $\alpha_m$ represent the cutoff parameter, the order of the filter, and the non-dimensional decay strength, respectively. In this study, $p_c$ was considered 0. We applied the filter $\mathscr{F}$ to the solution vector $q$ (in Eq. (4)) at the final stage of the RK scheme with a timestep $\Delta t$. Then, the decay time scale for the highest mode can be regarded as approximately equal to $\Delta t / \alpha_m$. We set the order $p_m$, and decay coefficient $\alpha_m$ such that the strength of filter should ensure numerical stability while being as weak as possible. We checked the impact of the modal filter on the convergence rate in Sect. 3.1. In addition, the investigation on how much the modal filters can contaminate the eddy viscosity with the turbulent model and the energy spectra was performed in KT2023.

## 3   Verification of dynamical core

We conducted several tests to verify our dynamical core. These tests are summarized in Table 4. For investigating the behavior of numerical convergence, the number of elements and polynomial order were changed as detailed in Table 5. The dissipation mechanisms used in the numerical experiments are summarized in Table 6. When evaluating numerical errors for the deterministic experiments such as linear advection, gravity wave, mountain wave, and baroclinic wave tests, we used the following error norms as

$$
\begin{aligned}
L_{1,\text{error}} &= \frac{\sum_E \int_{\Omega_E} |\psi(\xi,\eta,\zeta,t) - \psi_{\text{ref}}(\xi,\eta,\zeta,t)| \, \mathrm{d}\boldsymbol{x}}{\sum_E \int_{\Omega_E} \mathrm{d}\boldsymbol{x}}, \\
L_{2,\text{error}} &= \left[ \frac{\sum_E \int_{\Omega_E} [\psi(\xi,\eta,\zeta,t) - \psi_{\text{ref}}(\xi,\eta,\zeta,t)]^2 \, \mathrm{d}\boldsymbol{x}}{\sum_E \int_{\Omega_E} \mathrm{d}\boldsymbol{x}} \right]^{1/2}, \\
L_{\text{inf,error}} &= \max \left[ \psi(\xi,\eta,\zeta,t) - \psi_{\text{ref}}(\xi,\eta,\zeta,t) \right],
\end{aligned}
\tag{42}
$$

where $\psi(\xi,\eta,\zeta,t)$ and $\psi_{\text{ref}}(\xi,\eta,\zeta,t)$ denote the numerical and reference solutions, respectively, and $\sum_E$ represents the summation over all elements. Except for the linear advection test case, the results obtained from a sufficiently high-resolution experiment were used as the reference solution because the exact solution is unknown. In such case, the numerical solution was interpolated into the computational grid with the highest resolution experiment when evaluating the error norms.

For idealized climatological or turbulent flow simulations, such as the Held Suarez and global LES tests, it is difficult to directly evaluate the numerical convergence using the error norms defined in Eq. (42). In the long-termed integration, the chaotic behavior of the nonlinear systems can diverge the numerical solutions. In the turbulent flow simulations, a smaller-scale structure becomes more apparent as the grid spacing decreases until the spatial resolution reaches the physical dissipation scale. Thus, for the test cases, we mainly investigated on the impact of the polynomial order on the shortest wavelength at which the energy spectra began to separate from that in the reference solution.

### 3.1   Linear advection

To verify the spatial discretization with the cubed-sphere geometry, we conducted a two-dimensional linear advection test for a scalar quantity $q$. The experimental setup is similar to test case 1 of Williamson et al. (1992). The longitudinal and latitudinal

components of horizontal wind were prescribed by a solid body rotation as

$$u = u_0(\cos\phi\cos\phi_0 + \sin\phi\cos\lambda\sin\phi_0), \quad v = -u_0\sin\lambda\sin\phi_0, \tag{43}$$

where $\lambda$ and $\phi$ are the longitude and latitude coordinates, respectively, $u_0 = 2\pi a/(12\,[\text{days}])$, and $\phi_0$ denotes the angle between the axis of solid body rotation and the North pole. We considered three values of $\phi_0 = 0, \pi/4, \pi/2$ radians to investigate the impact of singularity with four corners of each panel in the cubed sphere. Although a cosine-bell profile is often given as an initial profile of the advected field, we used a Gaussian profile to confirm the order of accuracy higher than two. The profile is defined as

$$q(\lambda,\phi) = \exp\left(-\frac{d(\lambda,\phi)}{D}\right), \tag{44}$$

where $D$ is the characteristic horizontal scale; $d$ is the great circle distance between a position on the sphere $(\lambda,\phi)$ and the center position of Gaussian profile $(\lambda_c,\phi_c)$, which is calculated by

$$d(\lambda,\phi) = a\arccos\left[\sin\phi_c\sin\phi + \cos\phi_c\cos\phi\cos(\lambda - \lambda_c)\right]. \tag{45}$$

In this experiment, we set $D = a/5$ and $(\lambda_c,\phi_c) = (3\pi/2, 0)$.

To investigate a convergence rate, we changed the horizontal grid spacing $\Delta_{h,\text{eq}}$ from 313 km to 39 km for $p = 1,3,7$, and from 417 km to 52 km for $p = 11$. As a temporal scheme, we adopted a fully explicit fourth-order RK scheme described in Sect. 2.4.2. To focus on the spatial errors, we set sufficiently small timesteps such that $C_{r,\text{adv}} = 7.41 \times 10^{-2}$. In this experiment, the modal filter was not required because the upwind numerical flux provided a sufficient numerical stabilization.

Figure 1 shows the dependence of the $L_1$, $L_2$, and $L_{\text{inf}}$ errors at 12 days on the horizontal resolution. As theoretically expected, we obtained $p+1$-order spatial accuracy for $p = 1,3$, and 7. For $p = 11$ in the high spatial resolutions, the discretization error with the fourth-order temporal scheme, or the round-off error, degrades the convergence rate of 12th-order spatial accuracy. In the figure, the dashed lines represent the error norms in the case of $\phi_0 = \pi/4$ radians when the Gaussian profile passes over the singular points on cubed-sphere mesh. Their magnitudes were similar to those obtained for $\phi_0 = 0, \pi/2$ radians, which are represented by solid and dashed lines, respectively. For $p = 1$, the numerical errors were almost independent of the angle of the rotation axis. The errors for $\phi_0 = \pi/4$ radians can be smaller than those observed for $\phi_0 = 0, \pi/2$ radians (e.g., $p = 3$). The reason has not been confirmed, but we have found similar results in previous studies (Ullrich et al., 2010). In summary, when applying DGM to the advection problem, we consider the influence of singularity with the cubed-sphere coordinate to be quite small.

Because the modal filters are used in other test cases, we checked the impact of the modal filters on the numerical convergence. We conducted the linear advection test in the case of $\phi_0 = 0$ where the order and decay coefficient of the filter changed as $p_m = 64, 32, 16, 8$ and $\alpha_m = 10^{-3}, 10^{-1}, 10^{1}$. Figure 2 shows the impact of the modal filters on the horizontal resolution dependence of the error norms. The results indicate that the filters can degrade the original convergence rate and increase the numerical errors because the modal filter diminishes the high modes in the polynomial expansion. If the case of using high-order modal filters such as $p_m \geq 32$, the degradation of convergence rate was in the range of 1~3 for $p = 7, 11$ even if we set

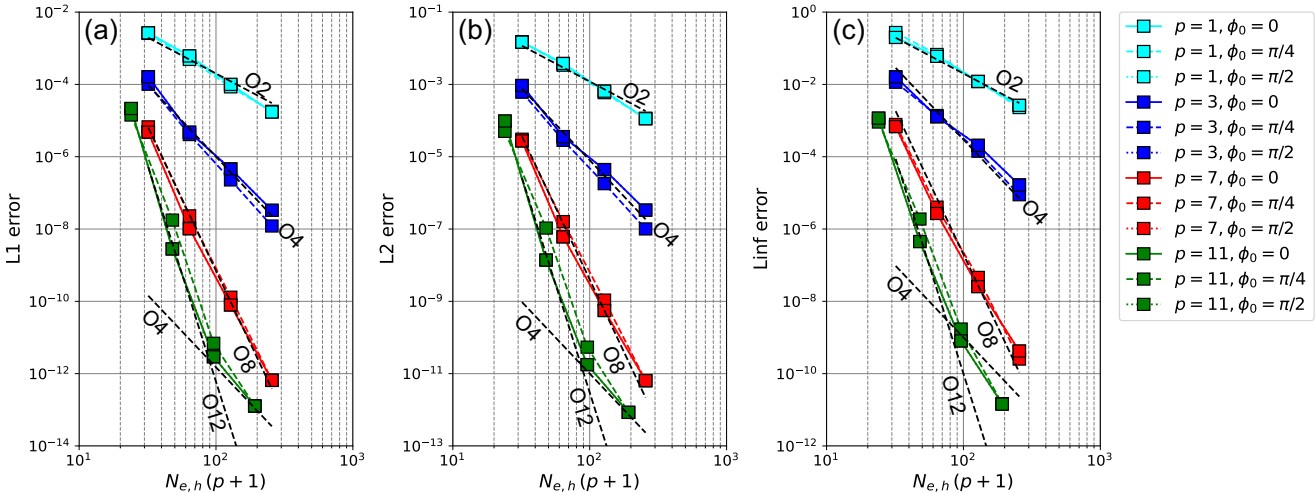

**Figure 1.** Dependence of (a) $L_1$, (b) $L_2$, and (c) $L_{\text{inf}}$ errors at $t = 12$ days on horizontal resolution in a two-dimensional linear advection test using $p = 1$, 3, 7, and 11. The colored solid, dashed, and dotted lines represent the results for $\phi_0 = 0$, $\pi/4$, and $\pi/2$, respectively. A black dashed line labeled "O$n$" indicates the slope with $n$-th order accuracy.

sufficiently large values of $\alpha_m$ such that the highest mode was immediately decayed after one timestep. For $p = 3$, the degrada-
415 tion of convergence rate appeared less obvious. However, it should be noted that the errors without the modal filter were much
larger compared to $p = 7, 11$. Thus, for $p = 3$, the effect of the increased error due to the filters may be more pronounced in the
representation of the flow fields.

## 3.2 Internal gravity wave

To check wave propagation with pressure gradient and buoyancy terms, test cases of gravity wave are often utilized. For
example, Tomita and Satoh (2004) also performed an internal gravity wave test. However, the basic state and initial perturbation
produce vertically high modes and nonlinear terms can develop small flow structures. This is inconvenient for investigating
numerical convergence. On the other hand, the experimental setting based on Baldauf and Brdar (2013), which originally
assumed a two-dimensional computational domain, can focus on a single mode. This study presents a global domain version
of gravity wave test in Baldauf and Brdar (2013). The initial condition was a rest isothermal atmosphere of $T_0 = 300$ K, which
corresponds to a constant Brunt Väisälä frequency of $\sqrt{g^2/(C_p T_0)} \sim 1.8 \times 10^{-2}$ s$^{-1}$. Furthermore, we added a small temperature
perturbation with a Gaussian profile as

$$T' = \Delta T \exp\left(-\frac{d}{D}\right) \sin\left(n_v \pi \frac{z}{z_T}\right) \exp\left(-\frac{g}{2RT_0}z\right), \tag{46}$$

where $\Delta T$ is the amplitude, $D$ is the characteristic horizontal scale, $n_v$ is the index with vertical mode, and $d$ was calculated
using Eq. (45). In this experiment, we set to $z_T = 10$ km, $\Delta T = 0.01$ K, $D = a/5$, $n_v = 1$, and $(\lambda_c, \phi_c) = (0, \pi)$. The Coriolis
force and topography were not considered.

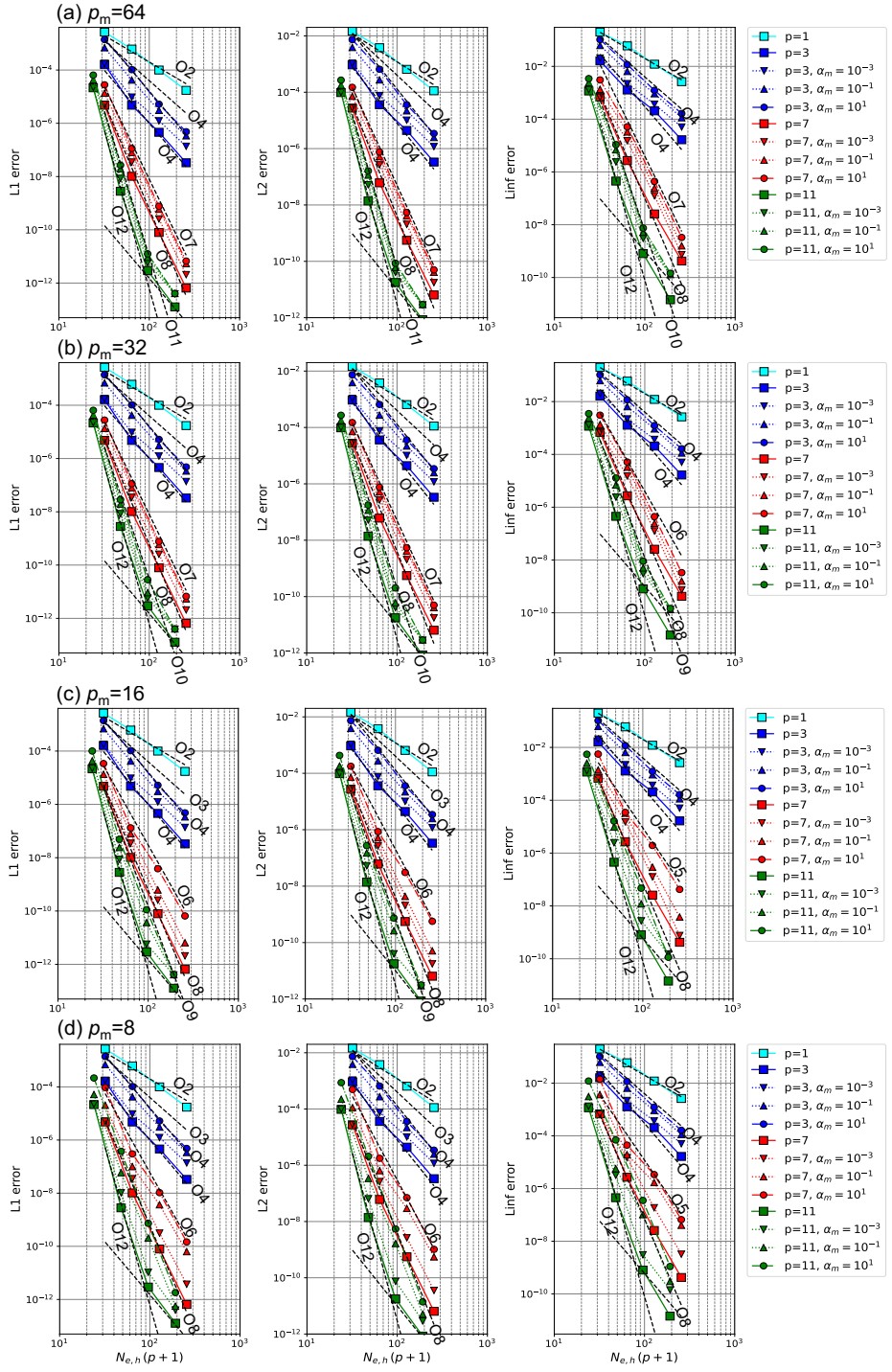

**Figure 2.** The impact of the modal filters on the horizontal resolution dependence of $L_1$, $L_2$, and $L_{\text{inf}}$ errors at $t = 12$ days in a two-dimensional linear advection test for the case $\phi_m = 0$ using $p = 1, 3, 7,$ and $11$: (a) $p_m = 64$, (b) $p_m = 32$, (c) $p_m = 16$, and (d) $p_m = 8$. In each $p_m$, we changed $\alpha_m$ as 0 (without the filter), $10^{-3}$, $10^{-1}$, and $10^1$.

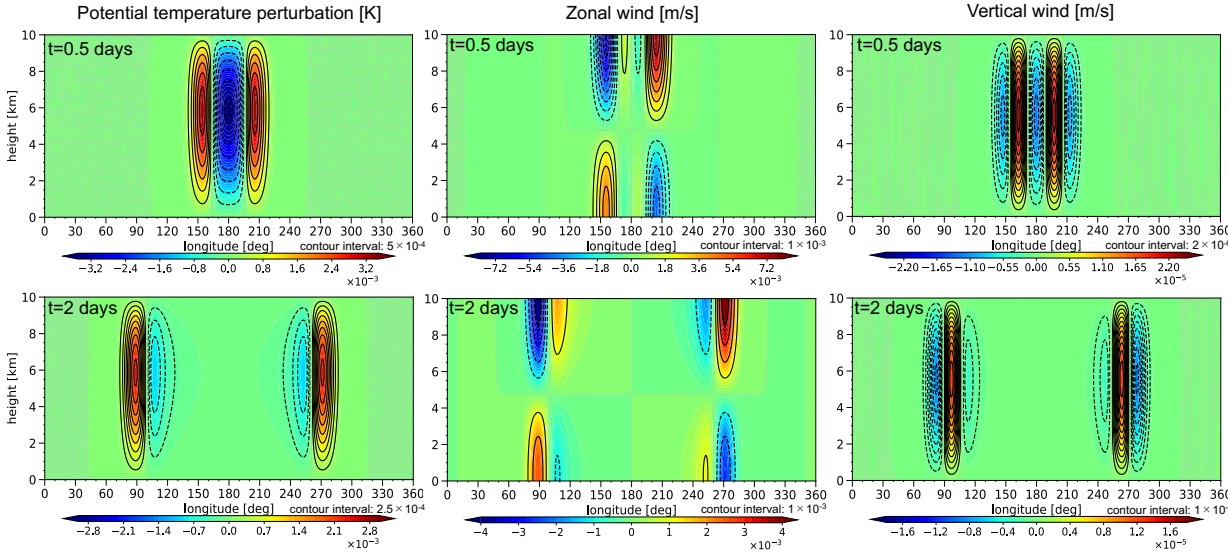

**Figure 3.** The spatial distribution of potential temperature, zonal wind, and vertical wind at the equator after $t = 0.5$ days (upper panel) and $t = 2$ days (lower panel) obtained from a gravity wave test case with $(\Delta_{h,\text{eq}}, \Delta_v) = (78\ \text{km}, 104\ \text{m})$ using $p = 7$.

The horizontal and vertical grid spacing were changed from (313 km, 417 m) to (39 km, 52 m) using $p = 1, 3, 7$. Whereas, for $p = 11$, they were changed from (208 km, 417 m) to (102 km, 104 m). As the temporal scheme, we adopted an IMEX Runge–Kutta scheme with the third-order accuracy, as described in Sect. 2.4.1. For the HEVI scheme, we set the timestep such that $C_{r,c_s} = 1.34 \times 10^{-1}$ for $p = 1, 3, 7$ and $C_{r,c_s} = 1.26 \times 10^{-1}$ for $p = 11$. To investigate the impact of temporal errors, we also conducted additional experiments with smaller timesteps for $p = 7$ and $p = 11$ where the above Courant number was reduced by factors of $1/2$ and $1/4$. In the absence of a modal filter, the self-convergence of numerical solutions was investigated. The reference solution was obtained from a high-resolution experiment where horizontal and vertical grid spacing were (20 km, 26 m) with $p = 7$ and $C_{r,c_s} = 6.30 \times 10^{-2}$.

To present the temporal evolution of gravity wave, Figure 3 shows the spatial distribution of potential temperature, zonal wind, and vertical wind after $t = 0.5$ days and $t = 2$ days. Based on this result, the horizontal phase speed was estimated to be approximately 58 m s$^{-1}$. This result corresponds well to the linear theoretical value under the hydrostatic approximation, $Nz_T/(\pi n_v) \sim 57$ m s$^{-1}$.

Figure 4 shows the dependence of error norms on spatial resolutions for the density perturbation $\rho'$, horizontal wind $u^\xi$, vertical wind $w$, and perturbation of potential temperature weighted density $(\rho\theta)'$. For relatively low-order $p$, such as $p = 1$ and $p = 3$, almost $p+1$-order accuracy was observed for the four variables in sufficiently high spatial resolutions. However, due to the fast wave modes, temporal errors for the third-order HEVI scheme can dominate over the spatial errors in the cases of larger $p$ and higher resolutions. This behavior was evident for the error norms of all variables except for the horizontal wind. For the case of $p = 7$ with $C_{r,c_s} = 1.34 \times 10^{-1}$, the convergence rate had approximately third-order slopes for $\rho'$ and $(\rho\theta)'$, and it had

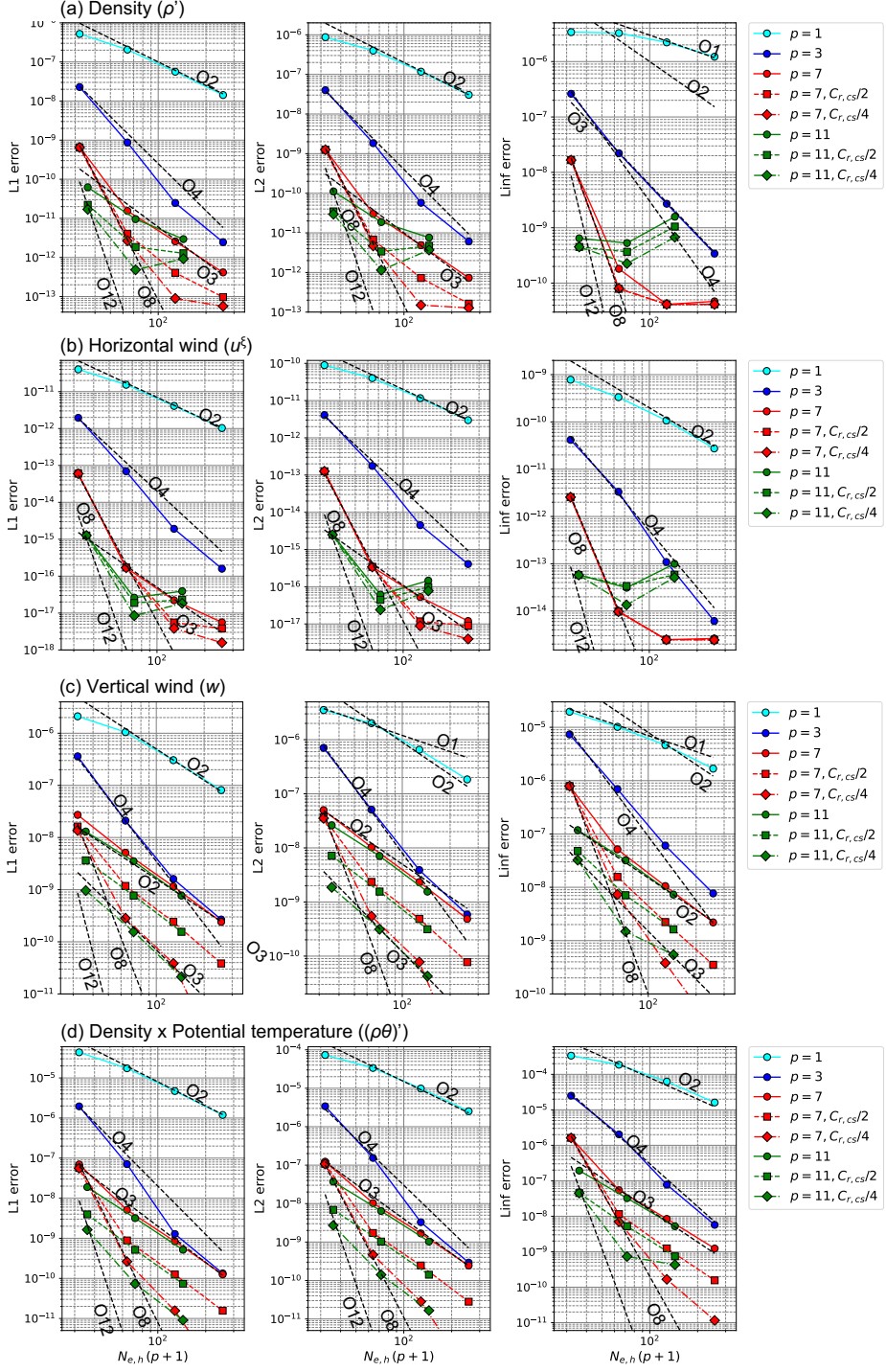

**Figure 4.** Dependence of $L_1$, $L_2$, and $L_{\text{inf}}$ errors on spatial resolution for (a) density perturbation ($\rho'$), (b) horizontal wind ($u^\xi$), (c) vertical wind ($w$), and (d) perturbation of potential temperature weighted density ($(\rho\theta)'$) after $t = 2$ days in a gravity wave test case.

between second- and third-order slope for $w$. As the timestep decreased, the convergence rate for $p = 7$ approached $p+1$-order. Even for the cases of small Courant number, due to an increase in round-off errors, the reduction in the error norms for $\rho'$ and $u^\xi$ stopped as the grid spacing decreased. For $p = 11$, this problem of round-off errors was worse. Stagnating of error reduction appeared in the spatial resolutions coarser than that in $p = 7$ and the errors increased with the spatial resolution. Note that the influence of round-off errors might be overestimated because the amplitude of initial perturbation was significantly small and no modal filter was used in this experiment. Thus, the problem is considered to be not critical in practical simulations that include the modal filtering or turbulent schemes.

## 3.3 Mountain wave

Adopting the basic terrain-following coordinate introduced in Sect. 2.1 together with low-order schemes is well known to produce large numerical errors with pressure gradient terms and to develop spurious flows (e.g., Zängl, 2012). However, such issues can be avoided using high-order DGM. To check the numerical behavior of the basic terrain-following coordinate in high-order DGM, we performed a mountain wave test on a reduced planet radius based on Klemp et al. (2015) (referred to as KSP2015) and the test case 2-1 in a Dynamical core model intercomparison project (DCMIP) test case document (Ullrich et al., 2012). Here, the planetary radius was set to $a/X_r$, where $X_r = 166.7$ is the scaling factor. In this experiment, the rotation was not considered. KSP2015 considered a topography profile in the form of

$$h^{\mathrm{KSP2015}}(\lambda, \phi) = h_0 \exp\left(-\frac{\tilde{d}^2}{d_0^2}\right) \cos^2\left(\pi \frac{\tilde{d}}{d_1}\right) \cos\phi, \tag{47}$$

where $\tilde{d} = a/X_r(\lambda - \lambda_c)$ (here, $\lambda_c = \pi$), $d_0 = 5000$ m, and $d_1 = 4000$ m. The maximum height of mountain $h_0$ was set to 25 m. In this topography profile, the mountain wave structure along the equator is comparable to the results with two-dimensional Schär type mountain (Schär et al., 2002). On the other hand, from the perspective of investigating the numerical convergence, it is undesirable for the zonal scale of topography to decrease with the latitudes and eventually become zero at the poles. To ensure that the minimum horizontal scale is sufficiently resolved in high resolution simulations, we eliminated the undulation of the mountain at the high latitudes using a tapering function as

$$h(\lambda, \phi) = h^{\mathrm{KSP2015}}(\lambda, \phi) \frac{1}{2}\left[1 + \tanh\left(\frac{|\phi| - \pi/3}{8\pi/180}\right)\right]. \tag{48}$$

As initial condition, we assumed a rest isothermal atmosphere of 300 K. KSP2015 considered an impulsive start where a zonal wind in solid body rotation ($u = U_0 \cos\phi$ where $u_0 = 20$ m s$^{-1}$) and the corresponding balanced state were initially given. However, such impulsive start produces initial shocks with small spatial scales, which complicates the discussion on the numerical convergence. To mitigate the influence of impulsive start, we gradually accelerated the wind using relaxation terms with the time scale of 60 s. Further details are provided in Appendix B1.

The horizontal and vertical grid spacing changed from (625 m, 500 m) to (156 m, 125 m) using $p = 3, 7, 11$. The model top was set to 30 km. As the temporal scheme, we used a fully explicit fourth-order RK scheme described in Sect. 2.4.2. For the HEVE scheme, we set the timesteps such that $C_{r,c_s} = 2.63 \times 10^{-1}$. The reflection of waves at the model top was suppressed by introducing a sponge layer at $z > 15$ km, where the vertical element size linearly increased with the altitude. Moreover,

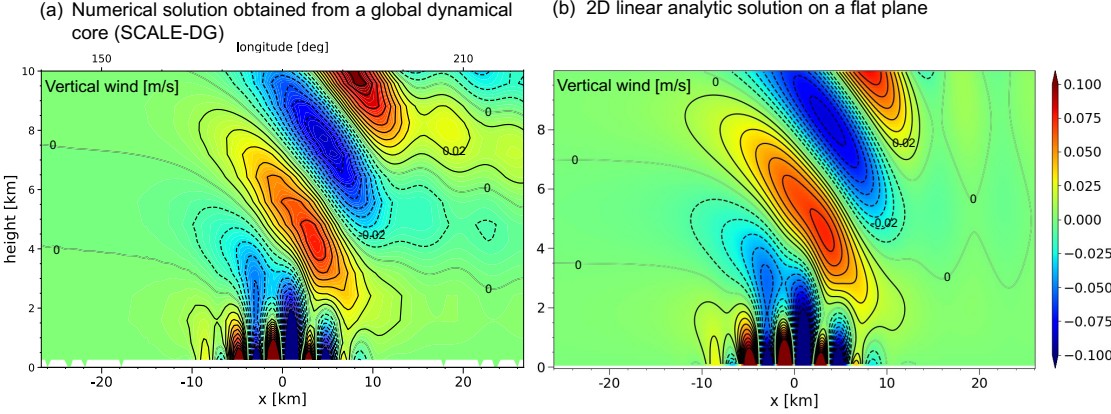

**Figure 5.** The spatial distribution of vertical wind at the equator obtained from a mountain wave test case with a Schär-like mountain: (a) Numerical solution at $t = 2$ hours obtained from $(\Delta_{h,\mathrm{eq}}, \Delta_v) = (625\text{ m}, 500\text{ m})$ using $p = 7$, (b) Two-dimensional linear analytic solution on a flat plane (shown for comparison).

a lateral sponge layer was placed to reduce the disruption of targeting mountain wave structure by initial shocks globally propagating. For the details on the sponge layer, refer to Appendix B2. The reference solution was obtained from a high-resolution experiment where the horizontal and vertical grid spacing were (78 m, 62.5 m) with $p = 7$. In this test case, to ensure numerical stability, we used a weak modal filter, which is summarized in Table 7.

Figure 5(a) shows the spatial distribution of vertical wind after 2 hours. For comparison, a linear analytic solution on a flat plane in the two-dimensional Cartesian coordinates is shown in Fig. 5(b) (The derivation can be found in Appendix A of KSP2015). Because the characteristic wavelength of mountain scaled by the Scorer parameter is $d_0 N/U_0 \sim O(1)$, this setting corresponds to a nonhydrostatic regime of mountain wave. In such regime, the waves with small-scale wavelengths are trapped near the surface, while large-scale waves propagate upward. The obtained wind pattern well reproduced the results shown in Fig. 2(a) of KSP2015. On the other hand, the numerical solution and the linear analytic solution on a flat plane was slightly different. For example, the vertical wavelength of the large-scale waves in Fig. 5(a) was shorter compared to that in Fig. 5(b). Based on the consideration using our regional dynamical core, the difference might be caused by the spherical experimental setup. Thus, we expect this difference to decreases as the planetary radius increases while the spatial scale of the mountain remains unchanged.

Figure 6 shows the dependence of error norms on spatial resolution for the density perturbation $\rho'$, horizontal wind $u^\xi$, vertical wind $w$, and perturbation of potential temperature weighted density $(\rho\theta)'$. A comparison performed at the fixed DOF shows that the numerical errors decreased with the increase in polynomial order, although the convergence rate was smaller than $p + 1$-order accuracy. For example, the slope of $L_2$ error norm was approximately 3/4 of that with $p + 1$-order accuracy. Based on additional experiments with the corresponding two-dimensional setup, the sub-optimal convergence can be related to several factors such as the modal filter and the spatial discretization for Jacobian cofactors ($\sqrt{G_v}G_v^{13}$ and $\sqrt{G_v}G_v^{23}$). For further details, refer to Appendix B3. Because the modal filters shave off the high modes in the polynomial expansion, the

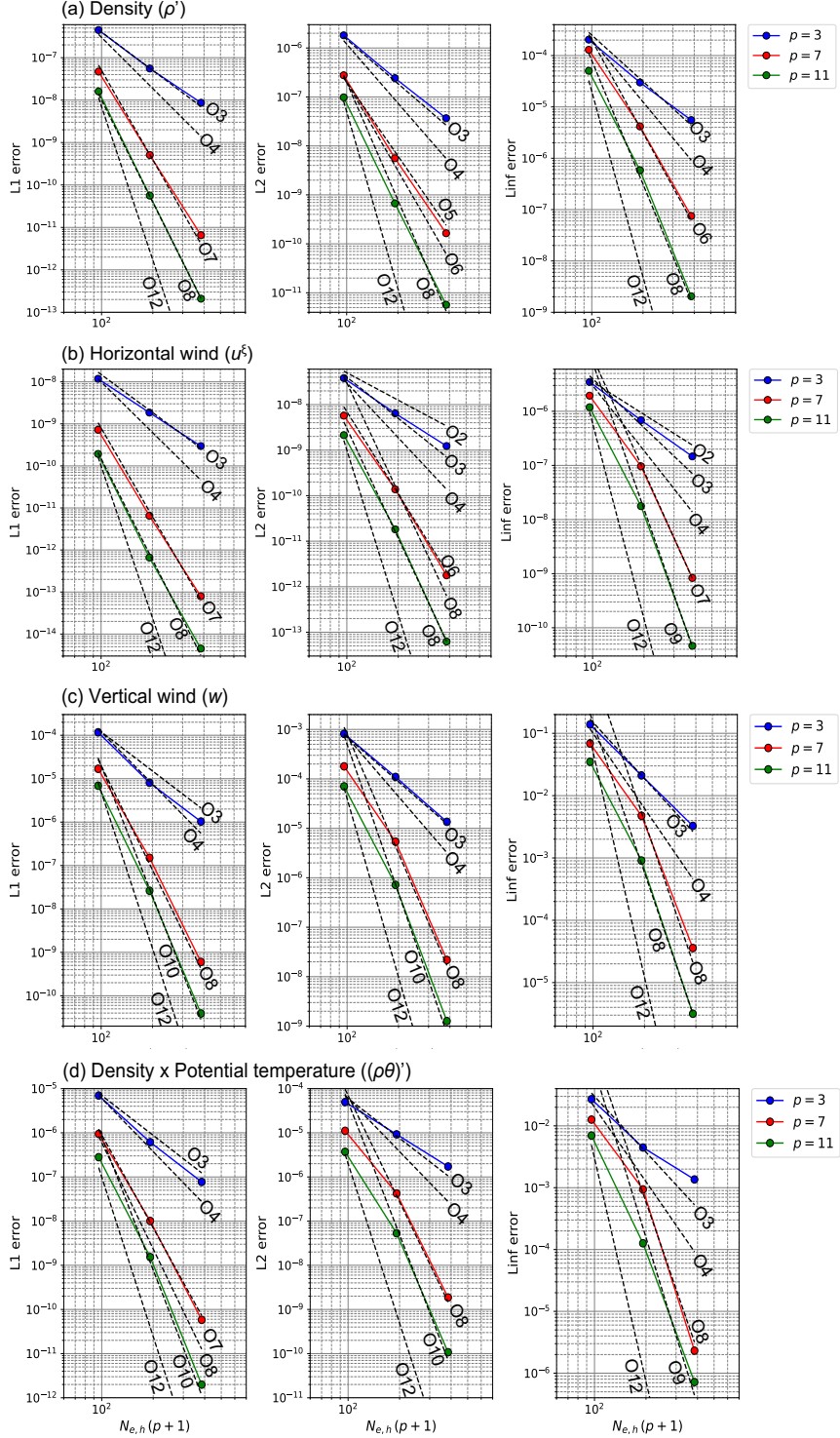

**Figure 6.** Dependence of $L_1$, $L_2$, and $L_{\mathrm{inf}}$ errors on spatial resolution for (a) perturbation of density ($\rho'$), (b) horizontal wind ($u^\xi$), (c) vertical wind ($w$), and (d) perturbation of potential temperature weighted density ($(\rho\theta)'$) after 2 hours in a mountain wave test case.

convergence rate can be degraded. When requiring a convergence rate with a certain order of the accuracy, we need to increase the polynomial order according to the filter intensity.

## 3.4 Baroclinic instability

Baroclinic instability is a typical phenomenon in the mid-latitudes. It includes small-scale structures such as front and filament formations. We conducted an idealized numerical experiment based on Jablonowski and Williamson (2006) (referred to as JW2006). In the aspect of numerical method, dynamical cores must accurately represent the wave growth process. In addition, it is necessary to treat the developing small-scale flow structures while ensuring numerical stability. For the experimental setup, the initial zonally symmetric fields were expressed using the analytic expressions of a steady-state solution of the adiabatic

inviscid primitive equations. To trigger baroclinic instability, a perturbation with a Gaussian profile was added to the zonal wind in the Northern hemisphere. For further details on parameter values, refer to Sect. 2(a) of JW2006.

     We investigated on the dependence of numerical solutions on the horizontal resolution as performed in JW2006. The horizontal grid spacing $\Delta_{h,\mathrm{eq}}$ changed from 250 km to 32 km with a fixed total vertical DOF of 24 for $p = 3$ and 7, whereas it changed from 208 km to 52 km with a fixed total vertical DOF of 36 for $p = 11$. We used a stretched vertical grid based on Eq.

(102) in Ullrich et al. (2012). The stretching parameter was set such that the vertical grid spacing $\Delta_v$ near the surface took the values of 305 m, 523 m, and 426 m for $p = 3, 7$, and 11, respectively. The stretching is further detailed in Appendix C. For the third-order HEVI scheme, we set the timesteps such that $C_{r,c_s} = 1.68 \times 10^{-1}$ for $p = 3, 7$ and $C_{r,c_s} = 1.26 \times 10^{-1}$ for $p = 11$. Furthermore, the modal filter was utilized to maintain the numerical stability. Its parameters are summarized in Table 8. When calculating the $L_2$ error of surface pressure, we used the results obtained from the corresponding highest resolution experiment

for each $p$ as the reference solution to directly discuss the behavior of numerical convergence associated with the horizontal spatial or temporal accuracy.

     Figure 7 shows the temporal evolution of baroclinic wave for the case of $\Delta_{h,\mathrm{eq}} = 63$ km using $p = 7$. The obtained horizontal distributions of surface pressure and temperature at 850 hPa were similar to those reported in the previous studies. For example, see Fig. 5 in JW06, which was obtained from the FV dynamical core (Lin and Rood, 1996, 1997). The wave grew very slowly

for 4 days. After that, the highs and lows deepened significantly and the wave began to break at the 8-th day. Figure 8 shows dependence of the surface pressure and temperature at 850 hPa (after 9 days) on the horizontal spatial resolution for $p = 7$. The same figure obtained from the FV dynamical core can be seen in Fig. 6 of JW06. Our dynamical core provided reasonably accurate numerical solutions for experiments performed at high spatial resolution. These solutions were comparable to the reference solutions reported in the previous studies. In addition, the effective resolution was apparently higher than that of the

low-order global dynamical core. For example, in the marginally resolved simulation setting, $\Delta_{h,\mathrm{eq}} \sim 250$ km, the amplitude and phase errors were small.

     For a quantitative evaluation of the horizontal resolution dependence, Figure 9 shows the temporal evolution of $L_2$ error norm of the surface pressure for $p = 3, 7$, and 11. In the figure, the gray shade represents an uncertainty range of reference solutions estimated by various dynamical cores in JW2006. In our evaluation strategy, we successfully captured the numerical

convergence with horizontal discretization and temporal errors. This is because the vertical spatial errors have similar values

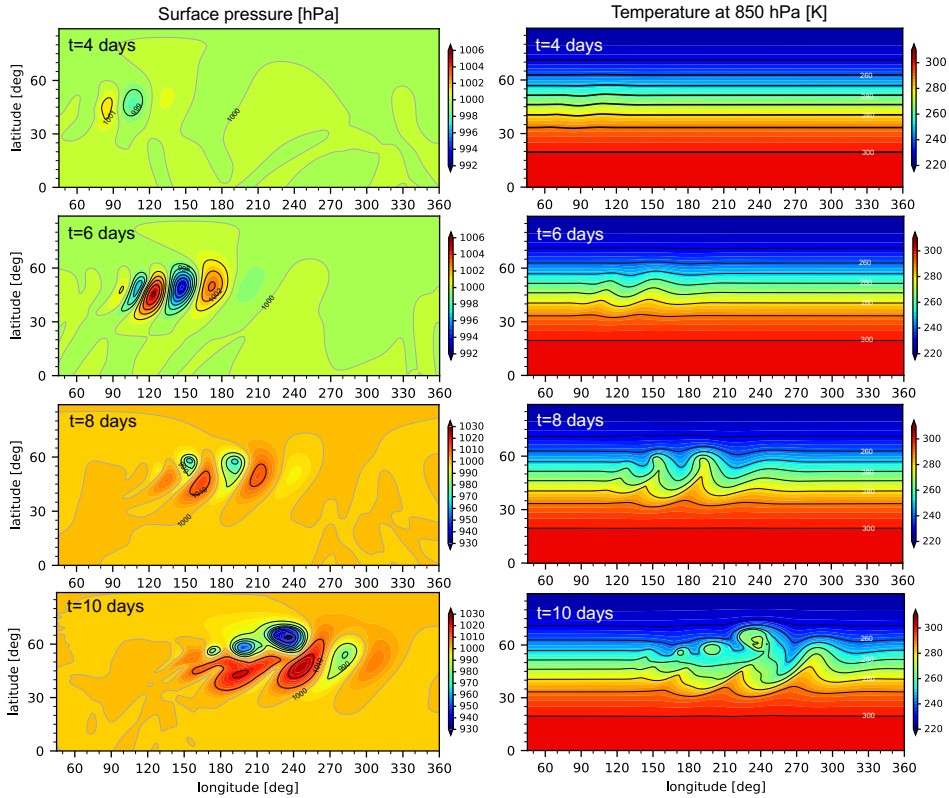

**Figure 7.** Spatial distribution of the surface pressure and temperature at 850 hPa after $t = 4, 6, 8,$ and 10 days in a baroclinic instability test. We present the results obtained from the experiment using $p = 7$ with $\Delta_{h,\text{eq}} = 63$ km and vertical DOF of 24.

among different horizontal resolution cases with the same $p$ and these errors virtually cancel out when the $L_2$ error is evaluated. Until about 6 days (except the initial adjustment stage), the $L_2$ error decreased with the horizontal resolution. The magnitude was significantly small compared to that reported in previous studies. For example, in the horizontal grid spacing of 50 km (0.5 degrees), the $L_2$ error was $1 \times 10^{-2}$ for the FV dynamical core and $5 \times 10^{-3}$ for Mcore (Ullrich and Jablonowski, 2012b). After 6

540  days when the baroclinic wave started to develop significantly, the $L_2$ errors rapidly grew and the difference between horizontal resolutions decreased. For $p = 3$, the feature of numerical convergence at the mature stage was similar to that obtained from MCore. In summary, the $L_2$ errors for $\Delta_{h,\text{eq}} < 250$ km are within the uncertainty range suggested by JW2006. Thus, we consider that the numerical solutions obtained from the proposed model are reasonable.

### 3.5  Held–Suarez test

As a long-term idealized benchmark for real climate simulations, we conducted the Held–Suarez test (Held and Suarez, 1994), which used a prescribed forcing that mimics complex physics parameterization. The boundary-layer friction was represented in a form of Rayleigh damping. The diabatic heating/cooling effect was represented using a Newtonian relaxation term to a

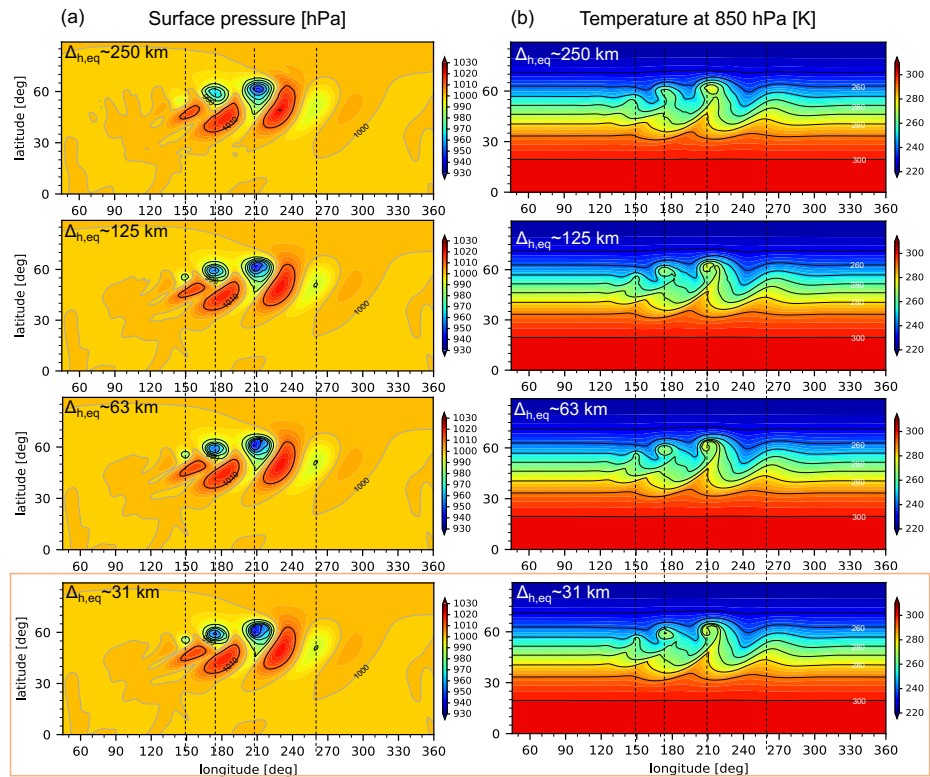

**Figure 8.** Horizontal resolution dependence of (a) surface pressure and (b) temperature at 850 hPa after $t = 9$ days in a baroclinic instability test. We present the results obtained from the experiments using $p = 7$ with the vertical DOF of 24.

prescribed temperature in radiative equilibrium $T_e$. For further details on these terms, see p.1826 in Held and Suarez (1994). In this study, a rest atmospheric field in hydrostatic balance with $T_e$ was given as the initial condition.

To investigate the spatial resolution dependence, the horizontal grid spacing $\Delta_{h,\mathrm{eq}}$ and vertical DOF changed from (208 km, 32) to (52 km, 128) for $p = 3, 7$, and from (208 km, 36) to (52 km, 144) for $p = 11$. The vertical grid spacing was stretched using the strategy in Appendix C. For example, when the vertical DOF is 32, the vertical grid spacing near the surface becomes approximately 350 m. In the cases of $\Delta_{h,\mathrm{eq}} = 208, 104$, and 52 km, the temporal integration was performed for 1200 days; The first 200-days data was discarded during the statistical analysis. For high resolution cases of $\Delta_{h,\mathrm{eq}} = 52$ km and 26 km, the

results after the spin-up period with coarser resolutions were used as the initial data, and temporal integration was conducted for 1000 days. As the temporal scheme, we adopted the third-order HEVI scheme with the acoustic Courant number of $C_{r,c_s} = 1.26 \times 10^{-1}$ for $p = 3, 7$ and $C_{r,c_s} = 7.56 \times 10^{-2}$ for $p = 11$. Moreover, we used the modal filters summarized in Table 9. Note that the large decay coefficients were set to stabilize long temporal integrations with nonlinear flow processes. The reference solution was obtained from a high-resolution experiment where $\Delta_{h,\mathrm{eq}}$ and vertical DOF were (26 km, 256) with $p = 7$.

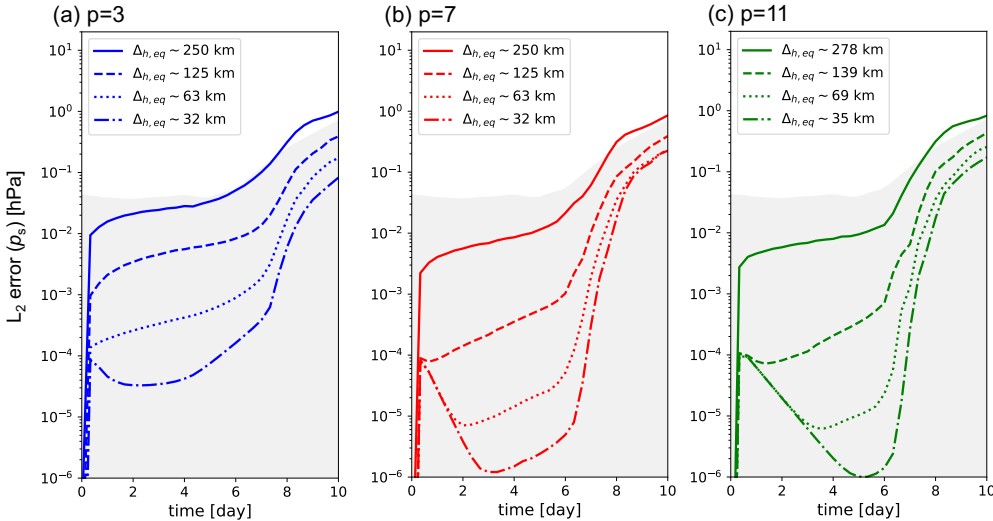

**Figure 9.** Dependence of the $L_2$ error norm for surface pressure on horizontal spatial resolution in a baroclinic instability test using (a) $p = 3$, (b) $p = 7$, and (c) $p = 11$. Note that the reference solution for each $p$ is the result from the corresponding highest resolution experiment.

Figure 10 shows the zonally and temporally averaged atmospheric fields in a statistical equilibrium state for $\Delta_{h,\mathrm{eq}} = 208$ km using $p = 7$. The obtained pattern and strength of general circulations were similar with the results obtained by using nearly the same horizontal spatial resolution used in previous studies (e.g., Wan et al., 2008). For a single westerly jet in each hemisphere, a maximum velocity of 32 m s$^{-1}$ was obtained at $p = 250$ hPa. Easterlies existed in equatorial and polar lower atmosphere and near the model top at low latitude. As shown in Fig. 10(c)-(f), the baroclinic wave activity in the proposed DG model was similar to that reported in previous studies. At $p = 250$ hPa in the mid-latitudes, the magnitude of eddy momentum flux reached approximately 70 m$^2$/s$^2$. The maximum of poleward eddy heat flux was located at $p = 850$ hPa in the midlatitude and its value reached approximately 22 K m s$^{-1}$. The eddy kinetic energy and temperature variance reached maximum values of approximately 430 m$^2$/s$^2$ at $p = 250$ hPa and 45 K$^2$ at $p = 800$ hPa in the midlatitude, respectively.

As discussed in a previous study (Wan et al., 2008), these eddy quantities such as the eddy kinetic energy and temperature variance are sensitive to the spatial resolution. As shown in Figs. 11(a), (c), the absolute peak values increased with the spatial resolution and began to converge when the horizontal grid spacing was less than 50 km. The convergence of peak values with $p = 7, 11$ was faster than that in the case of $p = 3$. For comparison, the corresponding peak values indicated from previous studies are shown by the colored boxes in the figure. The obtained trend of spatial resolution dependence for $p = 3$ was similar to that reported by studies using conventional low-order grid point methods (Tomita and Satoh, 2004; Wan et al., 2013). On the other hand, the peak values from $p = 7, 11$ at the horizontal grid spacing of approximately 200 km were similar to the results obtained by using the horizontal spectral method (Wan et al., 2008).

Figure 12 shows kinetic energy spectra of horizontal winds at $p = 850$ hPa and $p = 250$ hPa. As reported in previous studies (e.g., Malardel and Wedi, 2016; Tolstykh et al., 2017), the obtained spectra had the $n^{-3}$ slope at the spherical harmonic degrees

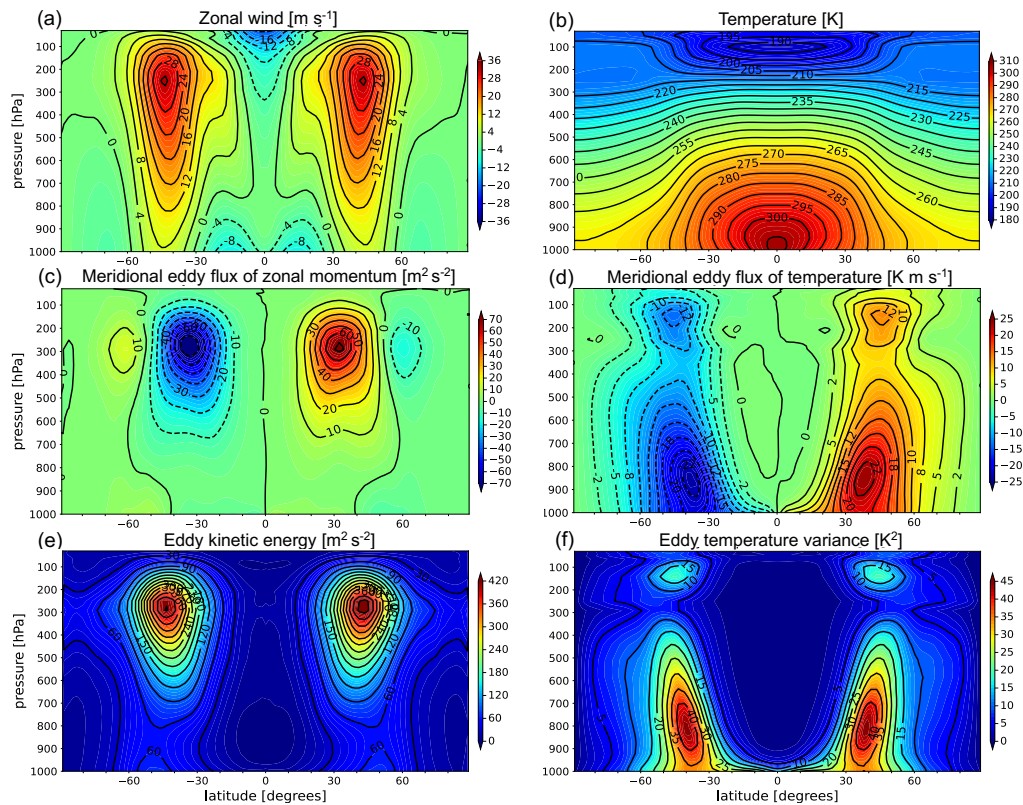

**Figure 10.** Zonally and temporally averaged atmospheric fields in a statistical equilibrium state: (a) zonal wind, (b) temperature, (c) meridional eddy flux of zonal momentum, (d) meridional eddy flux of temperature, (e) eddy kinetic energy, and (f) eddy temperature variance obtained from a Held–Suarez test with $\Delta_{h,\mathrm{eq}} = 208$ km using $p = 7$. As is typically done in previous studies, when taking the zonal and temporal average, we used the 1000-days data after the spin-up calculation.

between $10 \sim 100$. The steeper slope compared to $-3$ reflects the influence of numerical dissipation mechanism with the upwind
numerical flux and the modal filter. For the cases of $p = 7, 11$, the obtained energy spectra well followed that for the reference experiment at the wavelengths longer than eight grid lengths. In the spatial resolution dependence of peak values with the eddy quantities shown in Fig. 11, there was no significant difference between $p = 7$ and $p = 11$, whereas the improvement of effective resolution by higher polynomial order of $p = 11$ can be observed in the energy spectra as the grid spacing decreases. For $p = 3$, the energy spectra overlapped with that of the reference experiment at a wavelength range longer than $10\sim20$ grid
lengths. Furthermore, for $\Delta_{h,\mathrm{eq}} = 208$ km with $p = 3$, the entire spectra were smaller than the reference solution. Thus, using strong modal filters to ensure numerical stability for long-term integration has a significant effect on the spectra and effective resolutions when using $p \leq 3$. These results indicate that there is room for improving our treatment of the nonlinear terms to weaken the modal filters.

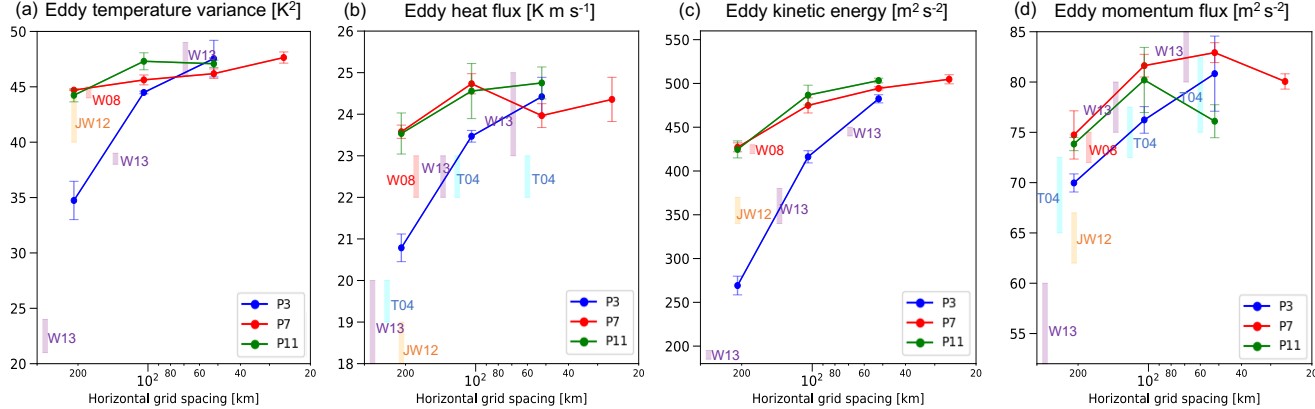

**Figure 11.** Dependence of absolute peak values on the spatial resolution: (a) eddy temperature variance, (b) eddy heat flux, (c) eddy kinetic energy, and (d) eddy momentum flux. The time averaging period is the same as that in Fig. 11. In each panel, the difference between the Northern and Southern hemispheres is represented using the error bars. The colored boxes labeled by T04, W08, UJ12, and W13 denote the corresponding peak values indicated by the results reported in Tomita and Satoh (2004), Wan et al. (2008), Ullrich et al. (2012), and Wan et al. (2013), respectively. Because the peak values were estimated from the contour figures, note that the uncertainty is large, and its range is roughly represented by the box height.

## 3.6 Planetary boundary layer turbulence experiment on a small planet

As a first step toward future global LES with O(10 m) grid spacing, we performed a global extension of the LES experiment of idealized planetary boundary turbulence in Nishizawa et al. (2015), KT2021, and KT2023. Currently, it is not feasible to conduct a global LES for a planetary size of Earth using a uniform spatial resolution of O(10 m). To save the required computational resources, the planetary radius was set to 3.4 km. Although this value is significantly different from that in realistic planets such as Earth and the effect of spherical geometry may affect the convection structure, we consider that this

test is useful to verify the turbulent model described in Sect. 2.2. We focused on the case of applying the shallow atmosphere approximation because we expected the results to be comparable to those reported in our previous studies. This approximation is obviously unsuitable for discussing physical aspect in this experimental setting. For the case without approximation, refer to Appendix D. The experimental setup is as follows: The altitude of model top was set to 3 km. There were no rotation and topography. Initially, we set a stable stratification with a vertical gradient of potential temperature of 4 K/km and added random

perturbations with an amplitude of 1 K. Because it is difficult to consider a uniform wind in the global situation, there was no initial motion in contrast to that in our previous studies. To drive thermal convections, a constant heat flux of 200 W m$^{-2}$ was imposed at the surface. To focus on the turbulent model, radiation and moist processes were not considered. In the turbulent model, we set the filter length to double that of the local grid spacing, which follows our previous studies. A reflection of waves at the model top was prevented using a sponge layer, where the vertical wind was decayed by the Rayleigh damping.

The $e$-folding time varied as the half cosine function from zero at $z = 2$ km to 10 s at the model top.

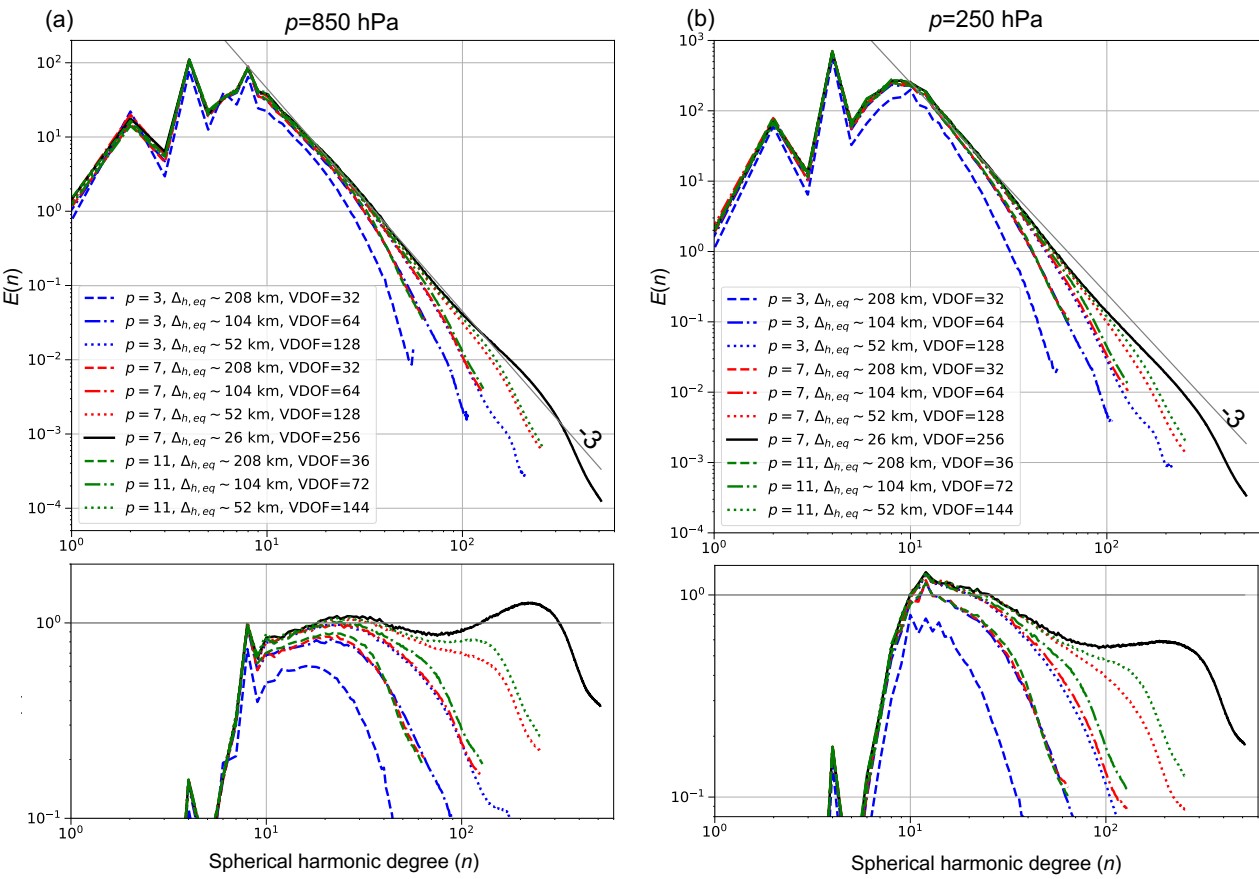

**Figure 12.** Energy spectra of horizontal wind in a statistical equilibrium state at (a) $p = 850$ hPa and (b) $p = 250$ hPa in a Held–Suarez test. As explained in the legend, the difference between spatial resolutions is represented by line types, and the line color indicates the polynomial order. The results from the reference experiment are shown by solid black lines. Lower panels represent the compensated spectra, which is proportion to $E(n)n^3$. The temporal average was calculated over 1000 days after the spin-up period; In the highest resolution case ($\Delta_{h,\mathrm{eq}} = 26$ km), it was performed over 300 days.

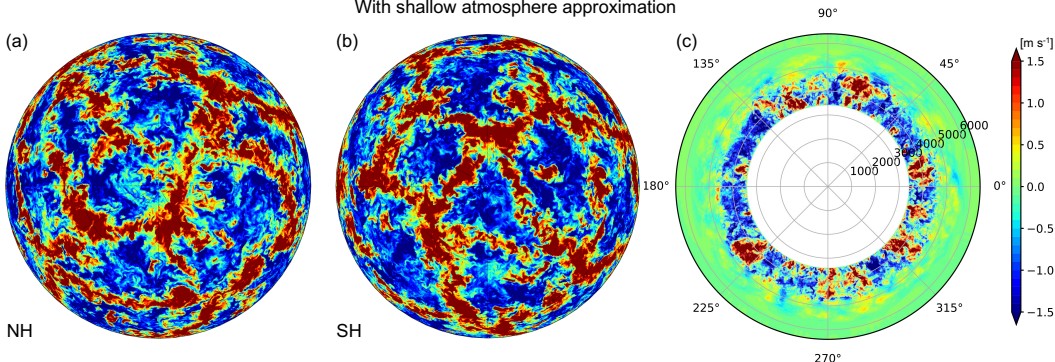

**Figure 13.** The horizontal distribution of vertical wind at $z = 500$ m after $t = 4$ hours when the shallow atmosphere approximation is applied in the LES of an idealized planetary boundary layer turbulence for the case of $\Delta_{h,\mathrm{eq}} = 10$ m using $p = 7$: (a) Northern hemisphere (NH), (b) Southern hemisphere (SH), and (c) their corresponding cross-sections along the equator.

To check the impact of polynomial order on the energy spectra as in KT2023, we changed $p$ as $3, 4$, and $7$ while setting the horizontal and vertical grid spacing to be approximately 10 m. Numerical stability was ensured by using a modal filter with parameters $p_{m,h} = p_{m,v} = 32$ and $\alpha_{m,h} = \alpha_{m,v} = 10^{-3}$. As the temporal scheme, a fully explicit fourth-order RK scheme described in Sect. 2.4.2 was used for the inviscid terms, whereas the forward Euler scheme was adopted for the SGS terms. We set the timestep such that $C_{r,c_s} = 4.38 \times 10^{-1}$. The integration time was 4 hours for the case of $p = 7$. To reduce the computational cost, the output at 3 hours was used as the initial condition of the other experiments for which the integration time was 1 hour.

Figure 13 shows the horizontal distribution for vertical wind at $z = 500$ m and cross-section along the equator after $t = 4$ hours in the case of $p = 7$. The convective cells had polygonal structures with a horizontal scale of approximately 2-3 km. The height of PBL reached between 1–1.5 km. To present the vertical structure of PBL, Figure 14 shows the vertical distribution of potential temperature, turbulent transport of heat and momentum, and skewness of vertical wind for $p = 7$. In these panels, the gray shade represents the results obtained from KT2023 using the plane regional model. The results obtained in this study were well similar to those reported in KT2023.

Figure 15 shows the kinetic energy spectra of three-dimensional wind at $z = 500$ m, which was temporally averaged during the last 30 minutes. The features of the obtained energy spectra were similar to those reported in KT2023. At longer wavelengths than eight grid lengths, the slope of spectra was approximately $-5/3$. On the other hand, the slope of spectra at the shorter wavelength range deepened due to the turbulent model, numerical dissipation with the upwind numerical flux, and the modal filter. KT2023 indicated that $p > 3$ is required that the effect of numerical diffusion term is sufficiently small compared to that of the SGS eddy viscosity term at the wavelength longer than eight grid length. This is true for global LES as shown in Fig. 15(b).

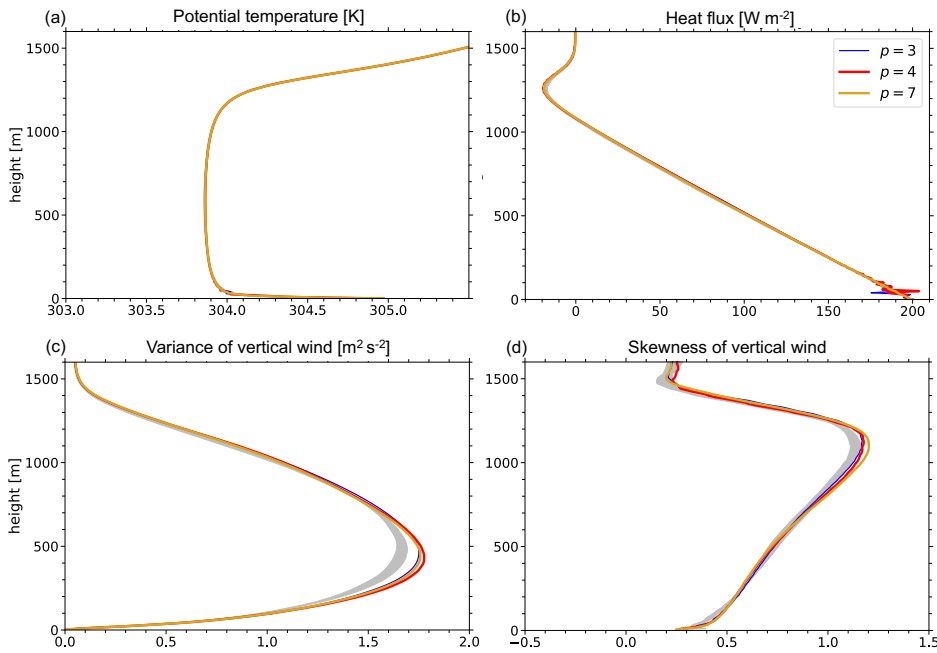

**Figure 14.** The vertical structure of PBL temporally averaged during the last 30 minutes: (a) potential temperature, (b) resolved eddy heat flux plus SGS heat flux, (c) variance of vertical wind, and (d) skewness of vertical wind for $p = 3, 4$, and 7. The gray shade represents the results obtained from KT2023 using the plane model.

## 4 Conclusions

For conducting future high-resolution atmospheric simulations precisely, our previous study (KT2021) indicated that conventional low-order discretization methods used in the state-of-the-art global nonhydrostatic dynamical cores have a problem of numerical errors because it is possible to contaminate the effect of physical parameterization schemes. To overcome this issue, we developed a global nonhydrostatic atmospheric dynamical core of dry atmosphere using the discontinuous Galerkin method (DGM) as the spatial discretization because DGM has several advantages over grid-point methods, including the simple strategy for high-order discretization and the high floating-point operations per second (FLOPS) count in recent parallel supercomputers. Furthermore, considering global LES, we formulated a Smagorinsky–Lilly type turbulent model in the cubed sphere coordinates and discretized it in the DGM framework. To verify the proposed global dynamical core, several numerical experiments, from the linear advection test to the Held–Suarez test, were conducted. To demonstrate the high-order numerical convergence, the experimental setup of existing test cases were slightly modified. In addition, an idealized test case was proposed to check the behavior of global dynamical cores including the turbulent model. Thorough the numerical experiments with various polynomial orders ($p$) and spatial resolutions, we discussed the impact of high-order spatial discretization on the quality of numerical solutions in the atmospheric flow simulations.

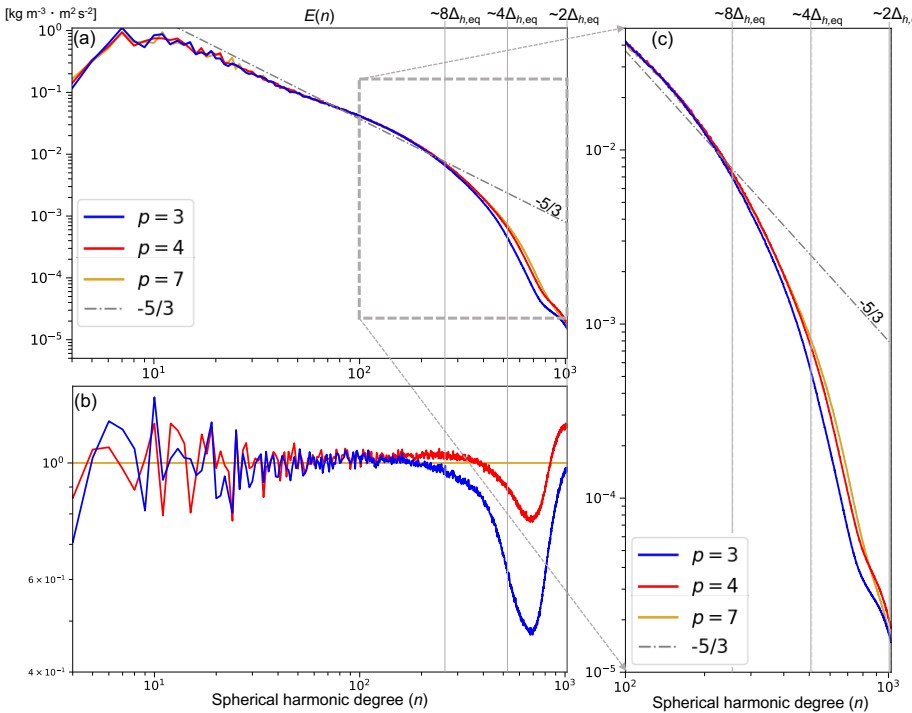

**Figure 15.** (a) Density-weighted energy spectra $E(n)$ of three-dimensional wind at the height of 500 m for $p = 3, 4$, and 7. The dash-dotted gray line represents $aE(n)$ where $a = 8.0 \times 10^1$. (b) Spectra normalized by the result of $p = 7$. (c) Partial expanded view of energy spectra in the short wavelength range.

For the deterministic test cases, such as the linear advection and gravity wave tests, $p+1$-order spatial accuracy was confirmed until the temporal discretization and round-off errors became significant compared to the spatial errors. In the gravity wave test, it was observed that the temporal errors with the third-order HEVI scheme can contaminate the convergence rate of high-order spatial discretization even when using the horizontal acoustic Courant number of O(0.1). To investigate the numerical performance of terrain-following coordinate with DGM, we conducted a mountain wave test case based on that used in Klemp et al. (2015). However, we made some modifications to investigate high-order numerical convergence. When comparing the results for a fixed DOF, the advantage of large polynomial order was apparent in terms of the fast numerical convergence, although the resultant convergence rate was slightly smaller than the optimal order associated with the spatial discretization. The results of the baroclinic instability test showed that, when $p \geq 3$ and $\Delta_{h,\text{eq}} < 240$ km, the obtained $L_2$ error norms of surface pressure entered the uncertainty range indicated by the previous studies. We confirmed the rapid numerical convergence over the second-order accuracy until the mature stage was reached. Subsequently, the sharp gradient with the front structure developed and the waves began to break, which made it difficult to identify the numerical convergence with the high-order schemes.

For test cases in which small-scale turbulent structures developed, such as the Held–Suarez test and the LES experiment of PBL turbulence, we mainly focused on the energy spectra in terms of the effective resolution. In the Held–Suarez test, where the turbulence model was not used, the extent of dissipation effect with the numerical flux and modal filters was clearly visible. Based on the comparison with the energy spectra for the reference experiment, we confirmed that the effective resolution was improved as the polynomial order increased. When we used high-order modal filters with large decay coefficients to ensure numerical stability during long temporal integration, the effective resolution was estimated to be between 10~20 grid lengths for $p = 3$ and eight grid lengths for $p = 7, 11$. To enhance the effective resolutions by weakening the modal filters, we consider that entropy stable DGM adopted in Souza et al. (2023) is a promising method, although this topic is beyond the scope of current study. To check the behavior of turbulent model included in global dynamical cores, we proposed an idealized test case of global LES considering with a small planetary radius, which is an extension of the experimental setup used in KT2021 and KT2023 with regional plane models. In our numerical experiments with the shallow atmosphere approximation, the convective cell pattern and vertical structures of PBL well reproduced the results of the regional plane model. We confirmed that the obtained energy spectra obeyed the Kolmogorov spectra of turbulence at the wavelength range longer than eight grid lengths when $p > 3$ was used together with the Rusanov numerical flux and a scale-selective modal filter. This result was consistent with the numerical criteria discussed in KT2023.

This study demonstrated the applicability of high-order DGM to global atmospheric dynamical cores via a series of numerical experiments; However, several tasks required to conduct realistic atmospheric simulations were not performed. To treat the effect of topography in LES, we must also consider the vertical coordinate transformation in the SGS terms of turbulent model. Such formulation can be achieved using the chain rule, as performed in the differential terms with inviscid fluxes. A related issue is the treatment of topography with steep slopes in high-order strategies. To investigate whether DGM-based dynamical core is robust for realistic topography, a Held-Suarez experiment with realistic topography may be appropriate. Such work is expected to yield deep understandings about the impact of effective resolutions of topography on large-scale flows when high-order DGM is used. Furthermore, a severe timestep restriction for explicit temporal schemes is one of the unresolved issues in high-order DGM. We expect that the computational overheads would be ignored in several cases; A coarser spatial resolution can be used due to the high-order numerical convergence or the small communication cost in DGM is taken advantage of. However, to accelerate DG dynamical cores in all situations, developing sophisticated temporal treatments is an important future work. Finally, it is indispensable to perform a coupling between the physics (such as moist and radiation processes) and DGM-based dynamics. Recent studies begin to discuss the potential difficulties with the element-based methods. For example, in the context of SEM, Herrington et al. (2019) indicated that a straightforward evaluation for physics tendencies at irregular nodes within the element causes a grid imprinting along the element boundaries. To solve this problem, they introduced a physics grid with a quasi-equal volume coarser than the node intervals with the dynamics when calculating the physics tendencies. While taking care of the advantages associated with the effective resolutions of high-order dynamical cores, we will explore how to treat the coupling of physics and dynamics in the DGM framework. For actual operational runs including physical processes and other factors, such as realistic topography, the degree of numerical filters depends on situations and cannot be generalized now. It is an important issue to produce a kind of criterion for the numerical stabilization.

**Table 1.** Values of parameters.

| Symbol | Description | Value |
|---|---|---|
| $C_p$ | Specific heat for constant pressure of dry air | $1.0046 \times 10^4$ J K$^{-1}$ kg$^{-1}$ |
| $C_v$ | Specific heat for constant volume of dry air | $7.1760 \times 10^3$ J K$^{-1}$ kg$^{-1}$ |
| $R$ | Gas constant | $C_p - C_v$ |
| $P_0$ | Reference value of pressure | 1000 hPa |
| $a$ | Planetary radius of planet | $6.3712 \times 10^3$ km |
| $X_r$ | Factor of reduced planetary radius | 166.7 |
| $\omega$ | Angular velocity of planet | $7.2920 \times 10^{-5}$ s$^{-1}$ |
| $g$ | Standard gravitational acceleration | 9.8066 m/s$^2$ |

**Table 2.** Double Butcher table for a third-order IMEX RK scheme proposed by Kennedy and Carpenter (2003).

| $c_s$ | $a_{ss'}$ | | | |
|---|---|---|---|---|
| 0 | 0 | 0 | 0 | 0 |
| $\dfrac{1767732205903}{2027836641118}$ | $\dfrac{1767732205903}{2027836641118}$ | 0 | 0 | 0 |
| $\dfrac{3}{5}$ | $\dfrac{5535828885825}{10492691773637}$ | $\dfrac{788022342437}{10882634858940}$ | 0 | 0 |
| 1 | $\dfrac{6485989280629}{16251701735622}$ | $-\dfrac{4246266847089}{9704473918619}$ | $\dfrac{10755448449292}{10357097424841}$ | 0 |
| $b_s$ | $\dfrac{1471266399579}{7840856788654}$ | $-\dfrac{4482444167858}{7529755066697}$ | $\dfrac{11266239266428}{11593286722821}$ | $\dfrac{1767732205903}{4055673282236}$ |

| $\tilde{c}_s$ | $\tilde{a}_{ss'}$ | | | |
|---|---|---|---|---|
| 0 | 0 | 0 | 0 | 0 |
| $\dfrac{1767732205903}{2027836641118}$ | $\dfrac{1767732205903}{4055673282236}$ | $\dfrac{1767732205903}{4055673282236}$ | 0 | 0 |
| $\dfrac{3}{5}$ | $\dfrac{2746238789719}{10658868560708}$ | $-\dfrac{640167445237}{6845629431997}$ | $\dfrac{1767732205903}{4055673282236}$ | 0 |
| 1 | $\dfrac{1471266399579}{7840856788654}$ | $-\dfrac{4482444167858}{7529755066697}$ | $\dfrac{1767732205903}{11593286722821}$ | $\dfrac{1767732205903}{4055673282236}$ |
| $\tilde{b}_s$ | $\dfrac{1471266399579}{7840856788654}$ | $-\dfrac{4482444167858}{7529755066697}$ | $\dfrac{11266239266428}{11593286722821}$ | $\dfrac{1767732205903}{4055673282236}$ |

*Code and data availability.* Source codes of SCALE-DG v0.8.0 and setting files for numerical experiments are available at the Zenodo repository (https://doi.org/10.5281/zenodo.10901697), where we have provided scripts to create figures in this paper. They are provided as open source under the MIT license. SCALE library v.5.5.1 which is a key dependent software of SCALE-DG is available at the Zenodo repository (https://doi.org/10.5281/zenodo.10952494), and is subject to the BSD-2-Clause license. Due to large data size, the obtained results from the numerical experiments are saved in the local storage at RIKEN R-CCS.

**Table 3.** Butcher table for a fourth-order fully explicit RK scheme with ten stages porposed by Ketcheson (2008).

| $c_s$ | $a_{ss'}$ | | | | | | | | |
|---|---|---|---|---|---|---|---|---|---|
| $0$ | | | | | | | | | |
| $\frac{1}{6}$ | $\frac{1}{6}$ | | | | | | | | |
| $\frac{1}{3}$ | $\frac{1}{6}$ | $\frac{1}{6}$ | | | | | | | |
| $\frac{1}{2}$ | $\frac{1}{6}$ | $\frac{1}{6}$ | $\frac{1}{6}$ | | | | | | |
| $\frac{2}{3}$ | $\frac{1}{6}$ | $\frac{1}{6}$ | $\frac{1}{6}$ | $\frac{1}{6}$ | | | | | |
| $\frac{1}{3}$ | $\frac{1}{15}$ | $\frac{1}{15}$ | $\frac{1}{15}$ | $\frac{1}{15}$ | $\frac{1}{15}$ | | | | |
| $\frac{1}{2}$ | $\frac{1}{15}$ | $\frac{1}{15}$ | $\frac{1}{15}$ | $\frac{1}{15}$ | $\frac{1}{15}$ | $\frac{1}{6}$ | | | |
| $\frac{2}{3}$ | $\frac{1}{15}$ | $\frac{1}{15}$ | $\frac{1}{15}$ | $\frac{1}{15}$ | $\frac{1}{15}$ | $\frac{1}{6}$ | $\frac{1}{6}$ | | |
| $\frac{5}{6}$ | $\frac{1}{15}$ | $\frac{1}{15}$ | $\frac{1}{15}$ | $\frac{1}{15}$ | $\frac{1}{15}$ | $\frac{1}{6}$ | $\frac{1}{6}$ | $\frac{1}{6}$ | |
| $1$ | $\frac{1}{15}$ | $\frac{1}{15}$ | $\frac{1}{15}$ | $\frac{1}{15}$ | $\frac{1}{15}$ | $\frac{1}{6}$ | $\frac{1}{6}$ | $\frac{1}{6}$ | $\frac{1}{6}$ |
| $b_s$ | $\frac{1}{10}$ | $\frac{1}{10}$ | $\frac{1}{10}$ | $\frac{1}{10}$ | $\frac{1}{10}$ | $\frac{1}{10}$ | $\frac{1}{10}$ | $\frac{1}{10}$ | $\frac{1}{10}$ |

**Table 4.** Summary of numerical experiments for validating the proposed dynamical core.

| Test case | Focus |
|---|---|
| Linear advection | Validation of the cubed-sphere projection, Convergence rate with advection, Impact of modal filters on numerical convergence |
| Internal gravity wave | Validation of the pressure gradient and buoyancy terms, Convergence rate with wave propagation |
| Mountain wave | Validation of the terrain-following coordinate, Convergence rate with vertical coordinate transformation |
| Baroclinic instability | Numerical robustness in developing small-scale flow structures, Numerical convergence discussed in previous studies |
| Held–Suarez test | Numerical robustness in climatic simulations with long-term integrations, Numerical convergence and effective resolutions |
| Global LES in a small planet | Validation of the turbulent model formulated in cubed-sphere coordinate, Effective resolutions on energy spectra, Consistency of numerical criterion indicated in KT2023 |

## Appendix A: Additional linear advection test

In Sect. 3.1, by conducting a linear advection test with a smooth profile in a solid-body rotation flow, we tested the spatial discretization with the cubed-sphere geometry and checked the convergence rate with the high-order DGM. In this section, we performed a linear advection test using the Gaussian hills and slotted cylinder profiles in a deformation flow, which is the

**Table 5.** Summary of the polynomial order $p$, the number of elements, and the resulting equatorial resolution $\Delta_{h,\mathrm{eq}}$ in the numerical experiments. For the number of elements, we denote the number of horizontally one-dimensional elements on a panel of the cubed-sphere as $N_{e,h}$ and the number of vertical elements as $N_{e,v}$.

| Test case | $p$ | $(N_{e,h}, N_{e,v})$ | $\Delta_{h,\mathrm{eq}}$ |
|---|---|---|---|
| Linear advection | 1 | $(16,-), (32,-), (64,-), (128,-)$ | 313, 156, 78, 39 km |
| | 3 | $(8,-), (16,-), (32,-), (64,-)$ | 313, 156, 78, 39 km |
| | 7 | $(4,-), (8,-), (16,-), (32,-)$ | 313, 156, 78, 39 km |
| | 11 | $(2,-), (4,-), (8,-), (16,-)$ | 417, 208, 104, 52 km |
| Internal gravity wave | 1 | $(16,12), (32,24), (64,48), (128,96)$ | 313, 156, 78, 39 km |
| | 3 | $(8,6), (16,12), (32,24), (64,48)$ | 313, 156, 78, 39 km |
| | 7 | $(4,3), (8,6), (16,12), (32,24), (64,48)$ | 313, 156, 78, 39 km |
| | 11 | $(3,2), (6,4), (12,8)$ | 278, 139, 69 km |
| Mountain wave | 3 | $(24,12), (48,20), (96,36)$ | 625, 313, 156 m |
| | 7 | $(12,6), (24,12), (48,20), (96,36)$ | 625, 313, 156, 78 m |
| | 11 | $(8,5), (16,8), (32,14)$ | 625, 313, 156 m |
| Baroclinic wave | 3 | $(10,8), (20,8), (40,8), (80,8)$ | 250, 125, 63, 31 km |
| | 7 | $(5,4), (10,4), (20,4), (40,4), (80,4)$ | 250, 125, 63, 31, 16 km |
| | 11 | $(3,3), (6,3), (12,3), (24,3)$ | 278, 139, 69, 35 km |
| Held–Suarez test | 3 | $(12,8), (24,16), (48,32)$ | 208, 104, 52 km |
| | 7 | $(6,4), (12,8), (24,16), (48,32)$ | 208, 104, 52, 26 km |
| | 11 | $(4,3), (8,6), (16,12)$ | 208, 104, 52 km |
| Global LES in a small planet | 3 | $(128,64)$ | 10 m |
| | 4 | $(100,52)$ | 11 m |
| | 7 | $(64,32), (12,8), (24,16), (48,32)$ | 10 m |

Case 4 presented in Nair and Lauritzen (2010) (hereinafter, referred to as NL2010). The experimental setup with the spatial resolution, polynomial order, and timestep was similar to that described in Sect. 3.1. To compare the errors reported in Guba et al. (2014) (hereinafter, referred to as G2014), we normalized the errors following Appendix C of NL2010.

Figure A1 shows the dependence of error norms on the horizontal resolution when the Gaussian hills were given as the initial condition of the tracer. Because of the infinitely smooth profile, we obtained $p+1$-order accuracy for $p = 1, 3, 7,$ and 11. The behavior of numerical convergence and the magnitude of errors were comparable to those in G2014 (see the values of $l_1$, $l_2$, $l_\infty$ for "Gauss." with the hyperviscosity (but without the limiter) in Tables 1-4 and Fig. 4 of G2014). In our results, the $L_{\mathrm{inf}}$ error was larger than the unity for $p = 11$ in the coarsest spatial resolution. This reflects a numerical instability with the aliasing errors near the static stagnation point of the deformation flow. It occurred when the static stagnation point was located at the element boundaries and numerical dissipation was not sufficient. By introducing a very weak modal filter with $\alpha_{m,h} = 2.5 \times 10^{-2}$ and $p_{m,h} = 64$, we can control the numerical instability as shown by the green dashed line in Fig. A1.

**Table 6.** Summary of dissipation mechanism in the numerical experiments. In the table, ○ means it is included, while − means it is not included. For the modal filtering in the mountain wave, baroclinic wave, and Held–Suarez tests, "scale-selective" means a high-order filter with $p_m \geq 8$ and "strong" means a large decay coefficient of $\alpha_m \geq O(1)$. The parameters of the filters are detailed in Tables 7, 8, and 9.

| Test case | Implicit diffusion with Rusanov flux | Explicit diffusion with modal filtering | Turbulence parameterization |
|---|---|---|---|
| Linear advection | ○ | − | − |
| Impact of modal filters in linear advection | ○ | $p_{m,h} = 64, 32, 16, 8$ and $\alpha_{m,h} = 10^{-3}, 10^{-1}, 10^{1}$ | − |
| Internal gravity wave | ○ | − | − |
| Mountain wave | ○ | scale-selective, weak (see Table. 7) | − |
| Baroclinic wave | ○ | scale-selective, weak for $p = 3$, scale-selective, strong for $p = 7, 11$ (see Table. 8) | − |
| Held–Suarez test | ○ | scale-selective, weak for $p = 3$, scale-selective, strong for $p = 7, 11$ (see Table. 9) | − |
| Global LES in a small planet | ○ | $p_{m,h} = p_{m,v} = 32, \alpha_{m,h} = \alpha_{m,v} = 10^{-3}$ | ○ |

**Table 7.** Modal filter orders and decay coefficients used in the mountain wave test.

| | $p_{m,h}, p_{m,v}$ | $\alpha_{m,h}, \alpha_{m,v}$ |
|---|---|---|
| $p = 3$ | 32 | $1.0 \times 10^{-2}$ |
| $p = 7$ ($\Delta_{h,\text{eq}} = 625$ m) | 64 | $1.0 \times 10^{-2}$ |
| $p = 7$ ($\Delta_{h,\text{eq}} = 313, 156, 78$ m) | 64 | $5.0 \times 10^{-3}$ |
| $p = 11$ ($\Delta_{h,\text{eq}} = 625$ m) | 64 | $1.0 \times 10^{-2}$ |
| $p = 11$ ($\Delta_{h,\text{eq}} = 313$ m) | 64 | $7.5 \times 10^{-2}$ |
| $p = 11$ ($\Delta_{h,\text{eq}} = 156$ m) | 64 | $5.0 \times 10^{-3}$ |

**Table 8.** Modal filter orders and decay coefficients used in the baroclinic instability test. Because the vertical resolution was fixed when increasing the horizontal resolution, the decay coefficient for the vertical filter $\alpha_{m,v}$ was reduced in proportion to the timestep.

| | $p_{m,h}$ | $\alpha_{m,h}$ | $p_{m,v}$ | $\alpha_{m,v}$ (for $\Delta_{h,\text{eq}} = 250$ km) |
|---|---|---|---|---|
| $p = 3$ | 16 | $2.0 \times 10^{-1}$ | 12 | $8.0 \times 10^{-1}$ |
| $p = 7$ | 16 | $2.0 \times 10^{0}$ | 16 | $1.2 \times 10^{0}$ |
| $p = 11$ | 24 | $1.6 \times 10^{1}$ | 24 | $1.6 \times 10^{1}$ |

**Table 9.** Modal filter orders and decay coefficients used in the Held–Suarez test after 250-days spin-up experiments. Note that we temporarily increased $\alpha_{m,h}$ to $6.0\times10^0$ for 20 days during the 1000-day integration for the case of $\Delta_{h,\mathrm{eq}}=26$ km using $p=7$.

| | $p_{m,h}$ | $\alpha_{m,h}$ | $p_{m,v}$ | $\alpha_{m,v}$ |
|---|---|---|---|---|
| $p=3$ | 8 | $1.0\times10^{-1}$ | 8 | $5.0\times10^{-2}$ |
| $p=7$ ($\Delta_{h,\mathrm{eq}}=208,104$ km) | 16 | $4.0\times10^0$ | 16 | $4.0\times10^0$ |
| $p=7$ ($\Delta_{h,\mathrm{eq}}=52,26^*$ km) | 16 | $5.0\times10^0$ | 16 | $5.0\times10^0$ |
| $p=11$ | 16 | $4.0\times10^0$ | 16 | $4.0\times10^0$ |

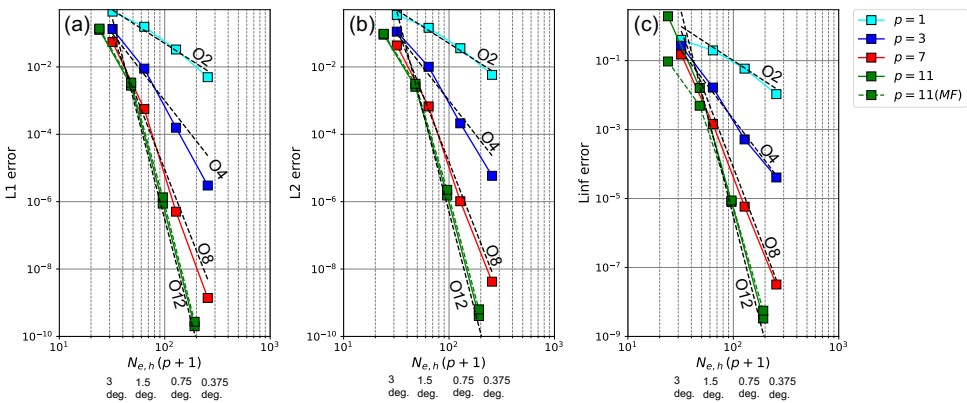

**Figure A1.** Dependence of (a) $L_1$, (b) $L_2$, and (c) $L_{\inf}$ errors at $t=12$ days on the horizontal resolution in the Case 4 in NL2010 with the Gaussian hills using $p=1,3,7,$ and 11. The green dashed line represents the case of applying a modal filter for $p=11$.

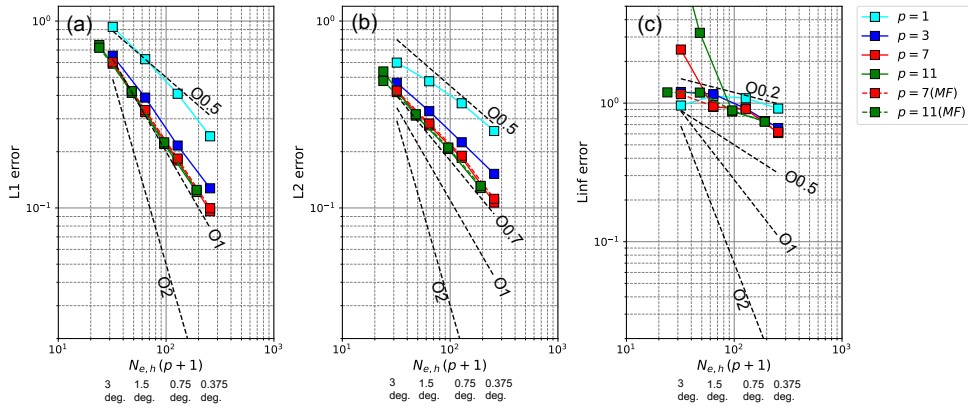

**Figure A2.** Dependence of (a) $L_1$, (b) $L_2$, and (c) $L_{\inf}$ errors at $t=12$ days on the horizontal resolution in the Case 4 in NL2010 with the slotted cylinders using $p=1,3,7,$ and 11. The dashed lines represents the cases of applying a modal filter for $p=7,11$.

Figure A2 shows the dependence of error norms on the horizontal resolution when the slotted cylinders were given as the initial tracer profile. Because of the $C0$ discontinuous field, we cannot expect the convergence rate higher than the first-order accuracy. Even when using the high polynomial orders $p > 3$, we obtained at most the first order for the $L_1$ error norm. For the $L_{inf}$ error, the convergence rate was near the zero order. The behavior of slow numerical convergence and the magnitude of numerical errors were similar to those reported in G2014 (see the error norms for "Cyl." with the hyperviscosity (but without the limiter) in Tables 1-4 of G2014). As seen in the Gaussian hills case, due to the numerical instability near the static stagnation point, we observed very large $L_{inf}$ error values for the cases of $p = 7, 11$ with the coarse spatial resolution. The modal filter used for the Gaussian hills case can suppress the numerical instability as shown by the dashed lines in Fig. A2.

## Appendix B: Additional information for mountain wave test

In this section, we detail the spin-up strategy and sponge layer, which were used in the mountain wave test described in Sect. 3.3. In addition, we consider reasons why the obtained convergence rate was slightly less than the optimal order accuracy.

### B1 Spin-up strategy

To mitigate the influence of impulsive start on numerical solutions, we gradually accelerated the wind as performed in previous studies with regional experimental setup (e.g., Durran, 1986; Sachsperger et al., 2016). The initial condition was a rest isothermal atmosphere and was represented as

$$u^\xi = 0, \; u^\zeta = 0, \; u^\eta = 0,$$
$$p = P_0 \exp\left(-\frac{gz}{RT_0}\right),$$
$$\rho = \frac{P_0}{RT_0} \exp\left(-\frac{gz}{RT_0}\right),$$

(B1)

where $P_0 = 10^5$ Pa and $T_0 = 300$ K. To accelerate a zonal wind, we added the relaxation terms in the right-hand side of governing equations as

$$\frac{\partial \sqrt{G}\rho'}{\partial t} = \cdots - \alpha_f \sqrt{G}\rho',$$
$$\frac{\partial \sqrt{G}\rho u^\xi}{\partial t} = \cdots - \alpha_f \sqrt{G}\left(\rho u^\xi - \rho U^\xi\right),$$
$$\frac{\partial \sqrt{G}\rho u^\eta}{\partial t} = \cdots - \alpha_f \sqrt{G}\left(\rho u^\eta - \rho U^\eta\right),$$
$$\frac{\partial \sqrt{G}\rho u^\zeta}{\partial t} = \cdots - \alpha_f \sqrt{G}\left(\rho u^\zeta - \rho U^\zeta\right),$$
$$\frac{\partial \sqrt{G}(\rho\theta)'}{\partial t} = \cdots - \alpha_f \sqrt{G}(\rho\theta)',$$

(B2)

where $(U^\xi, U^\eta, U^\zeta)$ are the vector components of prescribed wind and $\alpha_f$ is a time-dependent coefficient with Rayleigh forcing terms, which is provided in this subsection. Note that we set the hydrostatic balance part of pressure and density as

$$p_{\text{hyd}} = P_0 \exp\left(-\frac{u_{\text{eq}}}{2RT_0}\sin^2\phi - \frac{gz}{RT_0}\right), \quad \rho_{\text{hyd}} = \frac{p_{\text{hyd}}}{RT_0}, \tag{B3}$$

which satisfies a dynamically balanced state associated with a zonal wind in solid rotation, $u_{\text{eq}}\cos\phi$. Then, the perturbation at the initial time is given by $p' = p - p_{\text{hyd}}$, $\rho' = \rho - \rho_{\text{hyd}}$.

As the horizontal component of prescribed wind, we considered a zonal wind in solid body rotation where $u_{\text{eq}} = 20$ m s$^{-1}$. The corresponding $(U^\xi, U^\eta)$ can be calculated by considering the coordinate conversion between the cubed-sphere and geographic coordinates. To improve the inconsistency with no-flux boundary condition at the surface, the vertical component was added in the form of

$$U^\zeta = -\sqrt{G}_v (G_v^{13} U^\xi + G_v^{23} U^\eta) \exp\left(-\frac{\zeta}{H_f}\right), \tag{B4}$$

where $H_f$ was set to 2 km in this study. This modification also reduces the influence of initial shock. On the other hand, the coefficient with the forcing terms was given as $\alpha_f(t) = w(t)\tau_f^{-1}$, where

$$
\begin{aligned}
w &= 1 & \text{for } 0 \le t \le t_1, \\
w &= \frac{1}{2}\left[1 - \cos\left(\pi\frac{t-t_1}{t_2-t_1}\right)\right] & \text{for } t_1 \le t \le t_2, \\
w &= 0 & \text{for } t \ge t_2,
\end{aligned}
\tag{B5}
$$

and $\tau_f$ is the forcing time scale. In this study, these parameters were set as $\tau_f = 60$ s, $t_1 = 200$ s, and $t_2 = 1800$ s.

## B2  Sponge layer

To suppress a reflection of waves at the model top, we introduced a sponge layer at upper computational domain. In addition, to reduce the disruption of our targeting structure of mountain wave due to the global propagation of initial shocks, a lateral sponge layer was placed. As in Eq. (B2), linear damping terms were added to the governing equations as follows:

$$
\begin{aligned}
\frac{\partial \sqrt{G}\rho'}{\partial t} &= \cdots - \alpha_s \sqrt{G}\rho', \\
\frac{\partial \sqrt{G}\rho u^\xi}{\partial t} &= \cdots - \alpha_s \sqrt{G}\left(\rho u^\xi - U^\xi\right), \\
\frac{\partial \sqrt{G}\rho u^\eta}{\partial t} &= \cdots - \alpha_s \sqrt{G}\left(\rho u^\eta - \rho U^\eta\right), \\
\frac{\partial \sqrt{G}\rho u^\zeta}{\partial t} &= \cdots - \alpha_s \sqrt{G}\left(\rho u^\zeta - \rho U^\zeta\right), \\
\frac{\partial \sqrt{G}(\rho\theta)'}{\partial t} &= \cdots - \alpha_s \sqrt{G}(\rho\theta)'.
\end{aligned}
\tag{B6}
$$

The decay coefficient was given as $\alpha_s = (1 - w(t))(\alpha_{s,h} + \alpha_{s,v})$ where $\alpha_{s,h}$ and $\alpha_{s,v}$ are the coefficients for lateral and upper sponge layers, respectively. To avoid the sponge layer interfering with the initial forcing in Eq. (B2), as the initial forcing

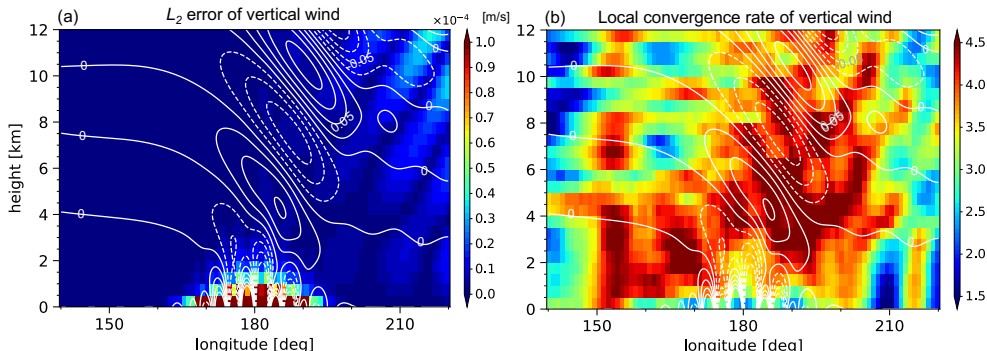

**Figure B1.** After 2 hours in a mountain wave test case with global model, spatial distribution of (a) $L_2$ error and (b) local convergence rate for the vertical wind at the equator. For the $L_2$ error, we show the result obtained from the experiment where the horizontal and vertical grid spacing $(\Delta_h, \Delta_v)$ are set to 156 m and 125 m, respectively, using $p = 3$. The cell color in the figure corresponds to the average values within the finite element. When evaluating the local convergence rate, we used the element average of $L_2$ error obtained from two experiments: a coarse resolution experiment ($\Delta_h = 625$ m, $\Delta_v = 500$ m) and the highest resolution experiment for $p = 3$ ($\Delta_h = 156$ m, $\Delta_v = 125$ m). To see the large-scale structure of local convergence rate, moving average was taken across the five elements horizontally. In each figure, the white lines represent the vertical wind in the highest resolution experiment for $p = 3$.

weakens, we gradually activated the sponge layer using the coefficient $(1 - w(t))$. The coefficient for the upper sponge layer was given as

$$\alpha_{s,v} = \frac{\tau_{s,v}^{-1}}{2} \left\{ \frac{1}{2} \left[ 1 + \tanh\left( \frac{z - (z_T + z_{\mathrm{sp}})/2}{\delta_{\mathrm{sp},v}(z_T - z_{\mathrm{sp}})} \right) \right] \right\}, \tag{B7}$$

whereas, for the lateral sponge layer,

$$\alpha_{s,h} = \frac{\tau_{s,h}^{-1}}{2} \left\{ \left[ 1 - \tanh\left( \frac{\lambda - \pi/4}{\delta_{\mathrm{sp},h}\pi/2} \right) \right] + \left[ 1 + \tanh\left( \frac{\lambda - 7\pi/4}{\delta_{\mathrm{sp},h}\pi/2} \right) \right] \right\} \cdot \frac{1}{2} \left[ 1 + \tanh\left( \frac{|\phi| - \pi/3}{8\pi/180} \right) \right], \tag{B8}$$

where $z_T$ is the height of model top, and $\tau^{s,v}$ and $\tau^{s,h}$ are the decay time scales corresponding to the upper and lateral sponge layers. Note that the coefficient for the lateral sponge layer is multiplied by a tapering function in the latitudinal direction to avoid an infinite zonal scale near the poles, as performed in Eq. (48). In this study, we set $z_{\mathrm{sp}} = 15$ km, $\delta_{\mathrm{sp},v} = \delta_{\mathrm{sp},h} = 0.16$, and $\tau_{s,v} = \tau_{s,h} = 100$ s.

### B3 Investigation on the degradation of the optimal numerical convergence

In Fig. 6, the convergence rate obtained from the mountain test case was slightly smaller than that achieved for $p + 1$-order accuracy. We consider the reasons behind this result to be as follows. First, to evaluate the differentials with the Jacobian cofactors ($\sqrt{G_v}G_v^{13}$ and $\sqrt{G_v}G_v^{23}$), we used same discretization operator, as described in Sect. 2.3. This strategy is beneficial to simply satisfy the geometric conservation law identity in the discretized equations. However, because the calculated geometric factors have the order $p$, it is possible to degrade the optimal convergence. Figures B1 (a), (b) show the spatial distribution of

numerical errors for vertical wind and the local convergence rate, respectively, for $p = 3$. The numerical error was large near the surface where the mountain exists. Furthermore, the relatively slow convergence rate appeared. The rate near the surface was between two and three, while it approached the value of four at locations apart from the surface. Second, the modal filter can reduce the convergence rate during the long-term temporal integrations even if we adopted a high-order modal filter with a relatively small decay coefficient.

To increase the certainty of our speculations, we conducted additional numerical experiments. To simplify the investigations and save the computational resources, we treated the corresponding two-dimensional experimental setup. With respect to the Jacobian cofactors, we considered two cases: (i) the case where it is numerically given by using the same discretization operator mentioned in Sect. 2.3, and (ii) the case when it is given by analytically evaluating the spatial derivatives at the node. In addition, to discuss the impact of modal filters on the convergence rate, we considered the case of no modal filter for $p = 3$ because we found that the 2-hours temporal integration without filters can be performed only for $p = 3$. As performed with the global model case, we conducted several numerical experiments changing the spatial resolutions and polynomial orders. To evaluate the error norms, we used the results from the reference experiments with $p = 7$, where horizontal and vertical grid spacing were $\Delta_h = 78$ m and $\Delta_v = 62.5$ m ($z < 15$ km), respectively.

Figures B2(a), (b) show the spatial distribution of numerical errors for vertical wind and the local convergence rate obtained from the two-dimensional experiments with $p = 3$. As shown in the global experiment (see Fig. B1), the convergence rate near the mountain achieved only the third-order accuracy in the cases of numerically calculated Jacobian cofactor. On the other hand, when the analytical Jacobian cofactor was used, the numerical errors near the mountain decreased and the convergence rate approached the fourth-order accuracy. Thus, the calculation strategy of Jacobian cofactor is one of the reasons for sub-optimal convergence.

Figure B3 shows that the dependence of $L_1$, $L_2$, and $L_{\text{inf}}$ errors on the spatial resolution. First, we focus on the results with $p = 3$. When the metric cofactors were analytically evaluated and the modal filter was not used, the fourth-order accuracy was observed except for the density. In case of numerically calculated Jacobian cofactor, the convergence rate of $L_2$ and $L_{\text{inf}}$ errors were characterized by the sub-optimal order because the Jacobian cofactors have only $p$th-order accuracy, as mentioned above. Such behavior was observed for horizontal wind, vertical wind, and the perturbation of potential temperature weighted density based on the comparison between (i) and (ii) cases. On the other hand, as indicated by the blue and cyan lines, the order reduction with the modal filters was obvious for the horizontal wind, while for other variables, there was little influence. This may be because the filters act on not only the perturbation part of horizontal wind but also on the mean flow part. For higher order cases ($p = 7, 11$), the filters are unavoidable for ensuring numerical stability in classical DGM. Then, the convergence rate can be limited by the modal filters, and the analytical Jacobian cofactor would have little impact. Even for the case (ii), $L_2$ and $L_{\text{inf}}$ errors of horizontal and vertical wind had the convergence rate slightly less than the optimal order. As for the density, note that the third-order accuracy was obtained for $p = 3$ even when using the analytical Jacobian cofactor and removing the modal filter. It remains unclear why the density error does not decrease in the optimal order. We may need to pursue how to discretely deal with the hydrostatic balance (e.g., Li and Xing, 2018) and investigate the boundary errors with no-normal flux condition near the surface.

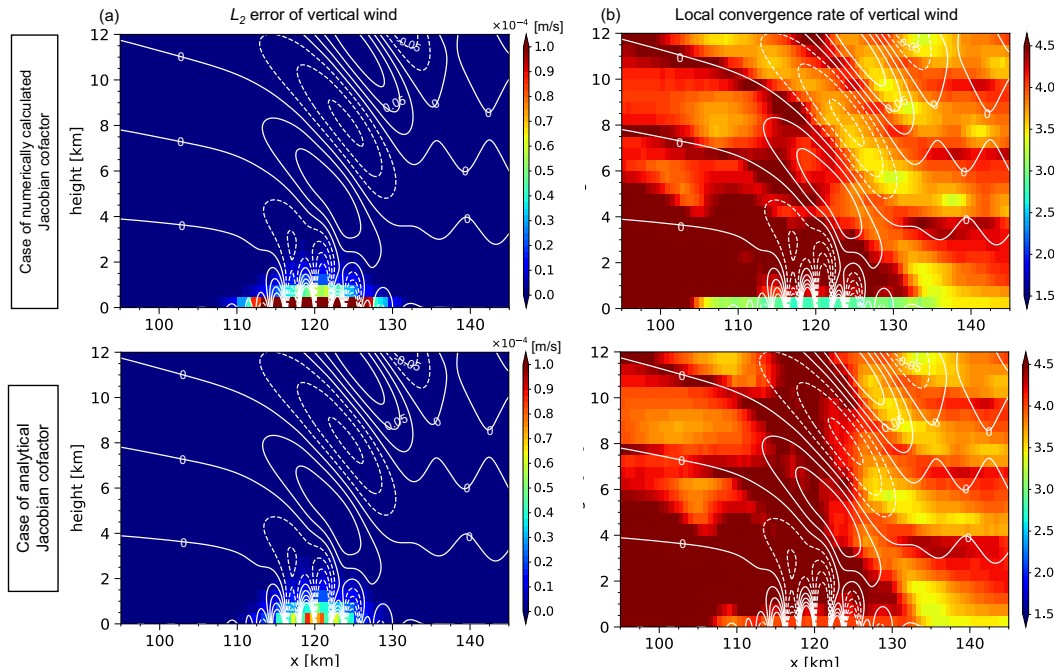

**Figure B2.** After 2 hours in a two-dimensional mountain wave test case, spatial distribution of (a) $L_2$ error and (b) local convergence rate for the vertical wind in the cases of numerically calculated Jacobian cofactor (upper panels) and analytical Jacobian cofactor (lower panels) for $p = 3$. In the $L_2$ error, we show the results obtained from the experiments with $\Delta_h = 156$ m and $\Delta_v = 125$ m. When evaluating the local convergence rate, we used the results obtained from two experiments: a coarse resolution experiment ($\Delta_h = 312$ m, $\Delta_v = 250$ m) and the highest resolution experiment for $p = 3$ ($\Delta_h = 39$ m, $\Delta_v = 31.25$ m). In each figure, the white lines represent the vertical wind in the reference experiment.

## Appendix C: Vertical grid stretching in the baroclinic wave and Held–Suarez tests

In the baroclinic wave and Held–Suarez tests, a function form for stretching the vertical element size is similar with that in Ullrich and Jablonowski (2012b). We calculated the vertical coordinate $\zeta$ at the top element boundary of $k'$-th element as

$$\zeta_{k'+\frac{1}{2}} = z_T \frac{1}{\sqrt{b+1}-1}\left[\sqrt{b\left(\frac{k'}{N_{e,v}}\right)^2 + 1} - 1\right] \tag{C1}$$

where $b$ is a positive parameter and $N_{e,z}$ is the number of elements in the vertical direction. As $b$ decreases, the vertical element size near the surface reduces compared to the upper domain of the model. We set $b = 20$ for $p = 3, 7$ and $b = 5$ for $p = 11$ in the baroclinic wave test, while $b = 3$ for $p = 3, 7$ and $b = 5$ for $p = 11$ in the Held-Suarez test.

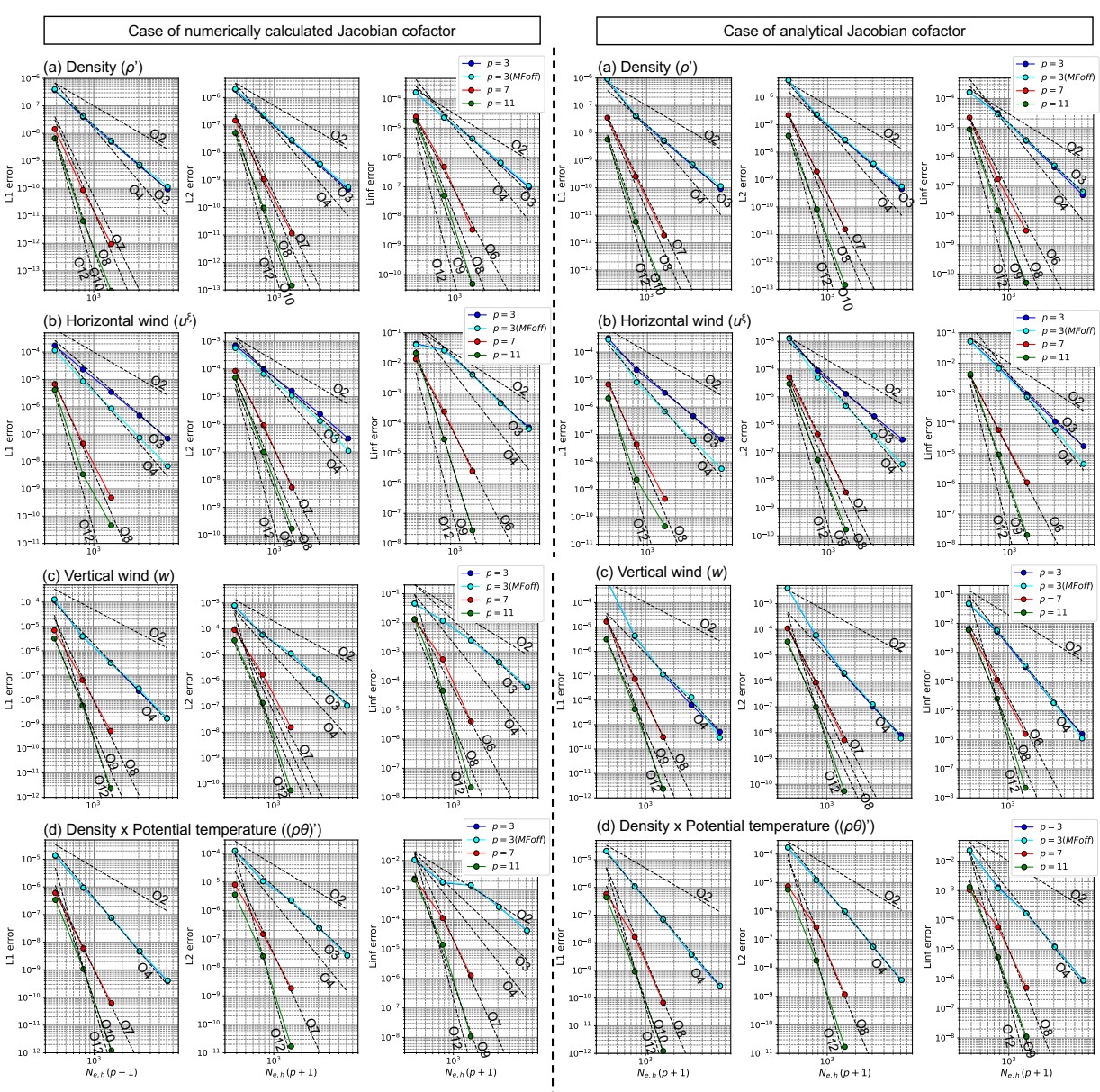

**Figure B3.** Dependence of $L_1$, $L_2$, and $L_{\text{inf}}$ errors on spatial resolution for (a) density perturbation ($\rho'$), (b) horizontal wind ($u^\xi$), (c) vertical wind ($w$), and (d) perturbation of potential temperature weighted density ($(\rho\theta)'$) after 2 hours in a mountain wave test case with the two-dimensional experimental setup. Note that the cyan lines represent the results for the case $p = 3$ without the modal filter (MFoff).

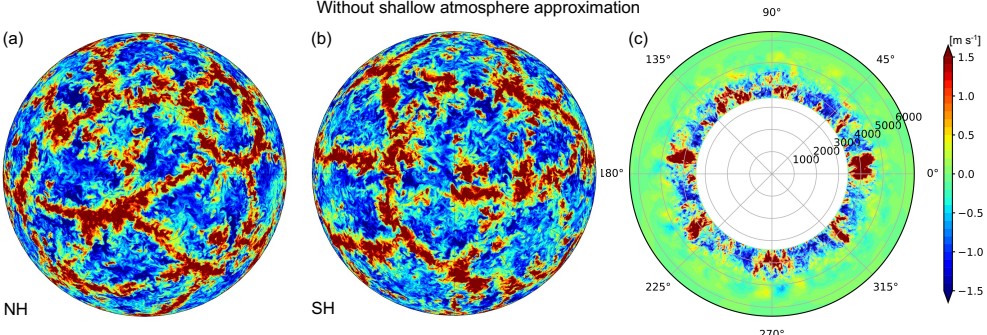

**Figure D1.** Same as Fig. 13, but these results were obtained without shallow atmosphere approximation. The horizontal distribution for vertical wind at $z = 500$ m after $t = 4$ hours: (a) Northern hemisphere (NH), (b) Southern hemisphere (SH), and (c) the corresponding cross-sections along the equator.

## Appendix D: The effect of not using shallow atmosphere approximation on global PBL turbulence experiment

In Sect. 3.6, we showed the results of PBL turbulence experiment with shallow atmosphere approximation. By applying this approximation, the obtained results become comparable with those reported in KT2021 and KT2023 which used the plane regional model. However, because the planetary radius is set to be 3.4 km, this approximation is not suitable for discussing physical aspects such as the impact of sphere geometry on the convective cells. This section shows the results when the shallow atmosphere approximation is not applied.

Figures D1 and D2 show the horizontal distributions of convective cells and vertical structure of PBL when the shallow atmosphere approximation is not applied. In Fig. D2, all results with the shallow atmosphere approximation are represented by the gray shade. An open cell pattern with the characteristic horizontal scale of a few kilometers was observed irrespective of whether we apply the shallow atmosphere approximation. On the other hand, the winds became weaker and the PBL was shallower compared to that in the shallow atmosphere approximation. We consider that such changes are consistent with the effect of spherical geometry because horizontal area increases with the altitude.

Figure D3 shows the energy spectra when the shallow atmosphere approximation is not applied. The major feature of energy spectra in the inertial subrange remained mostly unchanged. For example, the wavelength range obeyed -5/3 power law and the relation of effective resolution with polynomial order.

*Author contributions.* All authors have contributed to writing this paper. YK mainly performed the general development of the simulation programs based on DGM and conducted the numerical experiments. HT guided this project considering future high-resolution atmospheric simulations and high performance computers. He also contributed to improve the structure of this paper and discuss the simulation results.

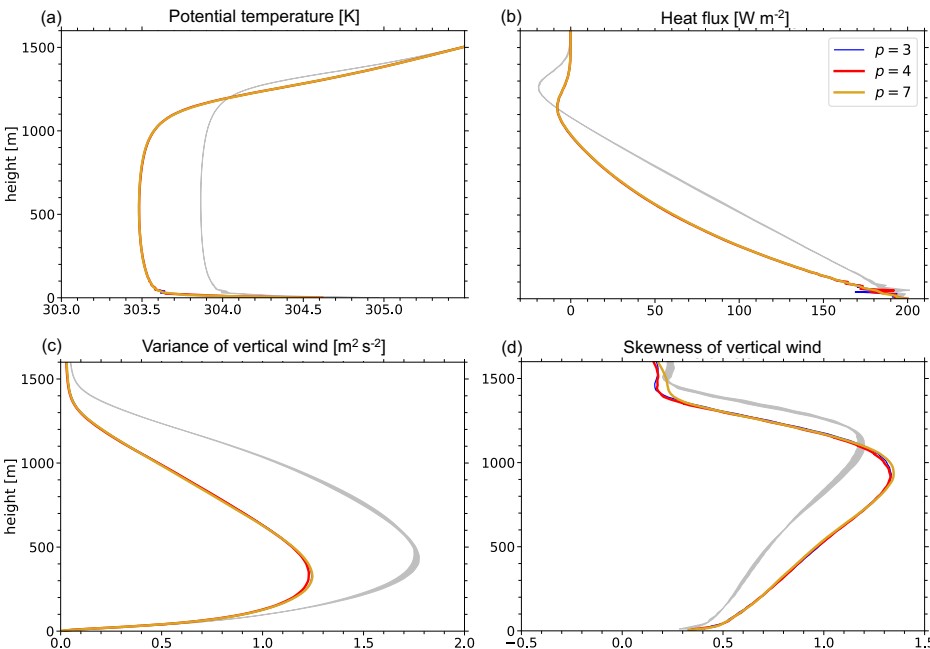

**Figure D2.** Same as Fig. 14, but figures show the vertical structure of PBL temporally averaged during the last 30 minutes without shallow atmosphere approximation: (a) potential temperature, (b) resolved eddy heat flux plus SGS heat flux, (c) variance of vertical wind, and (d) skewness of vertical wind. In these panels, the blue, red, and yellow lines represent the results for $p = 3, 4$, and 7, respectively. We indicate the corresponding results with the shallow atmosphere approximation, shown in Fig. 14, by the gray shade.

*Competing interests.* The authors have no competing interests to declare.

*Acknowledgements.* This research was supported by the MEXT KAKENHI (grant number: JP20H05731), Moonshot R&D (grant number: JPMJMS2286), the Foundation for Computational Science (FOCUS) Establishing Supercomputing Center of Excellence, JST AIP (grant number: JPMJCR19U2), JICA and JST SATREPS (grant Number: JPMJSA2109), and JSPS Core-to-Core Program (grant number: JPJSCCA20220001). The experiments in this study were performed using the supercomputer Fugaku at RIKEN (Project ID: ra000005, 840    hp200271, and hp230278) and the Wisteria supercomputer at the University of Tokyo. We thank Dr. Seiya Nishizawa, Dr. Hiroaki Miura, and Dr. Keiichi Ishioka and three reviewers (Dr. Valeria Barra and anonymous reviewers) for their valuable comments and suggestions. We would like to thank Editage (www.editage.jp) for the English language editing.

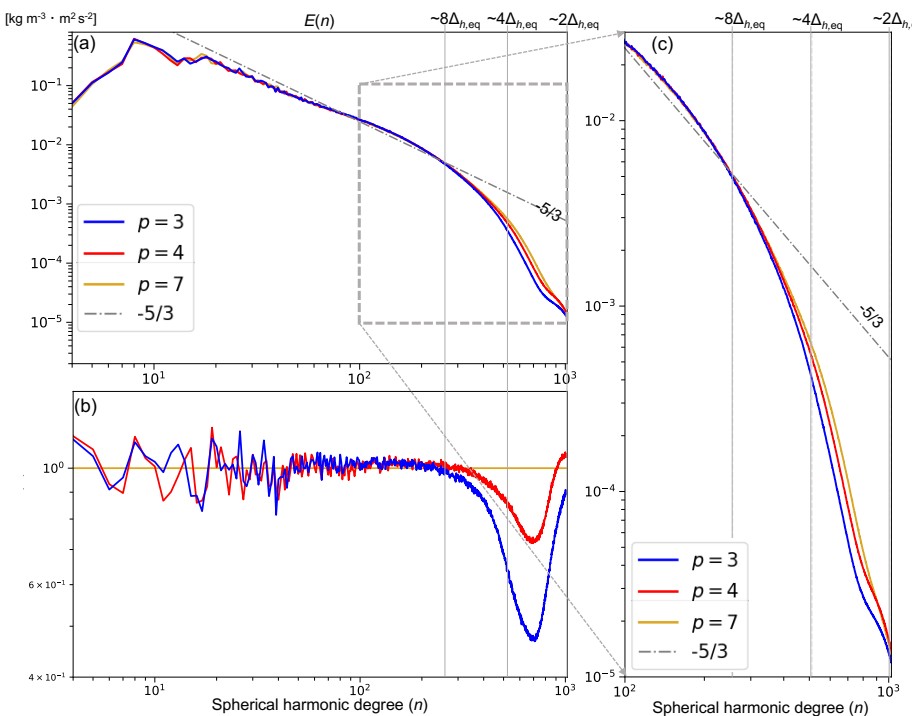

**Figure D3.** Same as Fig. 15, but figures show the results without shallow atmosphere approximation: (a) Density-weighted energy spectra $E(n)$ of three-dimensional wind at a height of 500 m for $p = 3, 4$, and 7. The dash-dotted gray line represents $aE(n)$ where $a = 8.0 \times 10^1$. (b) Spectra normalized by the result of $p = 7$. (c) Partial expanded view of energy spectra in the short wavelength range.

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
