# Peer review of "Development of high-order global dynamical core using discontinuous Galerkin method for atmospheric LES and proposal of test cases: SCALE-DG v0.8.0"

_EGUsphere, 2024_

## Referee Comment (RC1)

General comments:

The paper presents a numerical dynamical core (*dycore*) for global non-hydrostatic atmospheric simulations. The numerical discretization uses the high-order discontinuous Galerkin method (DGM) both horizontally and vertically and targets global atmospheric simulations in the setting of large-eddy simulation (LES), with grid spacing of O(10–100 m). The paper presents several numerical experiments to verify the numerical framework adopted. The problems are scientifically important and the work seems to have been carried out with care. The scientific significance of the work and novelty compared to other DG dycores are the aspects that concern me the most. The grammatical correctness of the English language is also another aspect that would require further revisions.

I am in general in favor of acceptance to GMD, provided the authors can convincingly respond to the comments.

Specific comments:

1) My main concern is with the aim and motivation of the work. Authors state that "Recently developed supercomputers have enabled us to conduct high-resolution global atmospheric simulations using a sub-kilometer horizontal grid spacing", without commenting or expanding on whether this should really be undertaken as a scientific endeavor. Just because we can, it does not mean that we should. The authors do not seem to outweigh the pros and cons of conducting such numerical simulations, especially in light of the carbon footprint and computational costs associated with said sub-kilometer scale simulations.

2) In the literature review, the authors seem to miss to mention the The Nonhydrostatic Unified Model of the Atmosphere (NUMA), which also successfully used DGM

3) The authors mention multiple times that they conduct classical numerical experiments to *validate* their numerical model. However, they seem to confuse Validation with Verification. In Numerical Analysis, the concepts of Verification and Validation (V&V) can be oversimplified in a succinct manner by saying that "verification is solving the equations right" (verifying the numerics) and "validation is solving the right equations" (verifying the physics - often done by comparing model results with actual data given by observations).

4) The authors seem to have chosen favorable examples/results and have not sufficiently provided explanations on reasons behind some degraded results, such as when a less than theoretical convergence rate was achieved. Also, the use of filters to overcome numerical instabilities was not always comprehensively justified and their effect on the quality of the results was not extensively elaborated on.

5) Section 3, line 331: Can the authors elaborate a little more on why they consider "difficult" to directly evaluate the numerical convergence in those cases?

6) In Sec 3.1 for the Linear Advection experiment, I wonder if the authors also verified their solver using a slotted cylinder example. This is often used in the literature because the sharp features of the geometry would particularly challenge the solver. I would appreciate if the authors would conduct such numerical experiments and would compare their convergence rates with the results reported in the literature, e.g., Guba et al.

"*Optimization-based limiters for the spectral element method*" (2014)
https://doi.org/10.1016/j.jcp.2014.02.029 looking at the results without limiters. In the
same section, regarding the numerical results in Figure 1, the authors have not
sufficiently explained why the case with $\alpha$ = 0, i.e., no singularity in the coordinates on
the cubed-sphere corners, in almost all cases presents larger numerical errors.

7) Sec 3.3, line 443: authors mention the modal filter as one potential reason for the
degraded sub-optimal convergence. Shouldn't it help instead? Can they elaborate on
this further?

8) Section 3.5, Caption of Figure 11: Can the authors explain why they presented numerical
results for the highest resolution case with a temporal average over only 300 days as
opposed to 1000 days for the other cases? Was it too computationally expensive to
perform the highest resolution simulation over 1000 days, or the model presented
difficulties over 300 days, such as it suffered from numerical instabilities/crashes?

Technical corrections (the referee will use italic font for addition to the quoted text where
appropriate):

1. Line 5: "the impact of high-order DGM on atmospheric flows was investigated". I
would rephrase this with another sentence along the lines of: "the impact of
high-order DGM on the quality or accuracy of the numerical simulations of
atmospheric flows was investigated"
2. Line 16: "In *the* near future"
3. Line 18: "Then, large-eddy simulation (LES) is a promising strategy, *since in* LES"
4. Line 33-34: Rephrase "In the context of DGM, KT2023 investigated the problem
with the order of accuracy necessary for LES"
5. Line 35: Add plural for generic or non specific countable nouns in English, i.e.,
"modal filters are used", or add an article if you want to use singular nouns
6. Line 36-37: Authors mention "2000-2010" but then they survey literature
belonging to the following decade
7. Line 64: Please introduce the FDM acronym before using it
8. Line 73: "the impact of high-order DGM on the atmospheric flows". I would
rephrase this, similar to the Abstract sentence.
9. Line 71-72: "We focused" and then "We attempt". Please check grammar
consistency of temporal tenses throughout the text
10. Line 86: "required" -> "requiring"?
11. Line 155: "angular velocity of *the* planet"
12. Line 156: "In *the* numerical experiments".
13. Line 172: "is essentially *the* same as"
14. Line 175: "In the absence of *a* vertical"
15. Line 183: "D is the divergence of *the* three-dimensional velocity"
16. Line 194: "For further details *on the* turbulence model"
17. Line 227: "For the numerical flux of *the* inviscid terms"
18. Line 229: reword "considered"
19. Line 230: "transformations*, and* is formulated as"

20. Line 270: "restrict the time step" (remove "to")
21. Line 280: "in *the* case of the diagonally implicit RK scheme"
22. Line 288: "To obtain the solutions of *the* nonlinear equation system"
23. Line 291: "In the case of *the* collocation approach"
24. Line 301: "When using *the* HEVI approach"
25. Line 302: "entries of *the* matrices"
26. Line 304: Rephrase with: "For high-order methods, numerical instability is likely to occur in advection-dominated flows, because the discrete advection operator is oscillatory."
27. Line 310: Remove "represents" or "is"
28. Line 317: "the order of *the* filter"
29. Line 318: "at the final stage of *the* RK scheme"
30. Line 320. Rename Section 3 "*Verification* of *the* dynamical core"
31. Line 322: "we mainly *focused* on the impact of *the* polynomial order on *the* effective". There are several missing articles throughout the text. I stopped correcting all of them after some point. The authors should more carefully proof-read for English correctness.
32. Line 339: I know $\alpha$ is used in the literature to denote the angle between the axis of the solid body rotation and the North pole. However, the authors should be careful because they also previously used $(\alpha, \beta, \zeta)$ for the local coordinates on the cubed-sphere.
33. Line 379: "error*s*"
34. Line 381: "*a* modal filter" and "*was* investigated"
35. Line 392: "except *for* the horizontal wind"
36. Line 427: "details *on* the sponge layer"
37. Line 446-447 avoid repetition of "includes" and "included" in the same sentence by using a synonym
38. Line 455: "stretch*ed*"
39. Line 473: "evaluation *of the* horizontal resolution"
40. Line 501: "by using *similar* spatial resolution"
41. Section A3: reword the section title "Investigation on the degradation of the optimal numerical convergence"
42. Figure A2: remove bold text in caption
43. Figure A3: remove bold text in caption
44. Line 741: I am sorry, but even in the acknowledgement sentence in which the authors thank the company they used for the English editing, there is a grammatical error "We would like to thank Editage for *the* English language editing"

---

## Author Comment (AC1)

**Response to reviewer (egusphere-2024-1477 manuscript)**

Dear Dr. Valeria Barra

Thank you for your careful review. We appreciate your thoughtful comments to improve our paper. We copied your comments in the blue text and have provided our responses in the black text. We have revised the manuscript according to your suggestions. Our point-by-point responses to the reviewer's comments are provided below. We hope that these improvements satisfactorily address the issues pointed out by you.

**General comments:**

The paper presents a numerical dynamical core (dycore) for global non-hydrostatic atmospheric simulations. The numerical discretization uses the high-order discontinuous Galerkin method (DGM) both horizontally and vertically and targets global atmospheric simulations in the setting of large-eddy simulation (LES), with grid spacing of O(10–100 m). The paper presents several numerical experiments to verify the numerical framework adopted. The problems are scientifically important and the work seems to have been carried out with care. The scientific significance of the work and novelty compared to other DG dycores are the aspects that concern me the most. The grammatical correctness of the English language is also another aspect that would require further revisions.

We are grateful that you noticed the importance of the problems treated in our study. Before answering the specific comments, please allow us to refer to the scientific significance of the work and novelty compared to previous studies that developed DG dynamical cores. As other reviewer pointed out, the spatial discretization strategy used in SCALE-DG follows a nodal DGM (e.g., Hesthaven and Warbuton, 2007), which is mostly similar with that used in ClimateMachine and NUMA. Although this study has little novelty in the context of numerical methods of DGM, we consider that the following points to be our unique contributions:

1) Introduction of a turbulent model to a global DG dynamical core on cubed-sphere coordinates:

   To construct a global LES model, we formulated SGS eddy viscous and diffusion terms with a Smagorinsky-Lilly type turbulent model on cubed-sphere geometry in the DGM framework. Several previous studies (Ullrich, 2014; Guba et al., 2014) presented strategies for the vector Laplacian operator in element-based global shallow water models on the cubed-sphere coordinates. For our purpose of introducing the turbulent model, we treated the Laplacian

operator acting on the component of vector fields in the cubed-sphere coordinates and the eddy viscosity dependent on local flow fields. Furthermore, we introduced the turbulence model to our DG dynamical core and verified its behavior by conducting an LES experiment of idealized planetary boundary layer turbulence.

2) Modification of test cases for high-order dynamical core:

We modified existing test cases to investigate the numerical convergence associated with high-order dynamical cores. When using the totally second-order dynamical cores, due to relatively large discretization errors, the problem of ill-posed experimental setting might not be essential. However, we modified the experimental setup to evaluate numerical features, such as the convergence rate, of high-order dynamical cores.

3) Evaluation of numerical convergence with global dynamical core based on DGM:

By conducting several standard tests, we quantitatively evaluated the numerical convergence of a global nonhydrostatic dynamical core based on DGM and indicated the high-order convergence rate. Although such investigations for regional DG dynamical cores can be found (e.g., Giraldo and Restelli, 2008; Bardar et al., 2013; Blaise et al., 2016), few studies are available on global DG dynamical cores.

[References]

- Blaise, S., Lambrechts, J., Deleersnijder, E. (2016): A stabilization for three-dimensional discontinuous Galerkin discretizations applied to nonhydrostatic atmospheric simulations. *International Journal for Numerical Methods in Fluids*, *81*(9), 558-585. https://doi.org/10.1002/fld.4197
- Brdar, S., Baldauf, M., Dedner, A., Klöfkorn, R. (2013): Comparison of dynamical cores for NWP models: comparison of COSMO and Dune. *Theoretical and Computational Fluid Dynamics*, *27*, 453-472. https://doi.org/10.1007/s00162-012-0264-z
- Giraldo, F. X., Restelli, M. (2008): A study of spectral element and discontinuous Galerkin methods for the Navier–Stokes equations in nonhydrostatic mesoscale atmospheric modeling: Equation sets and test cases. *Journal of Computational Physics*, *227*(8), 3849-3877. https://doi.org/10.1016/j.jcp.2007.12.009
- Guba, O., Taylor, M. A., Ullrich, P. A., Overfelt, J. R., Levy, M. N. (2014). The spectral element method (SEM) on variable-resolution grids: Evaluating grid sensitivity and resolution-aware numerical viscosity. *Geoscientific Model Development*, *7*(6), 2803-2816. https://doi.org/10.5194/gmd-7-2803-2014

- Ullrich, P. A. (2014): A global finite-element shallow-water model supporting continuous and discontinuous elements. *Geoscientific Model Development*, *7*(6), 3017-3035. https://doi.org/10.5194/gmd-7-3017-2014

**Specific comments:**

1. My main concern is with the aim and motivation of the work. Authors state that "Recently developed supercomputers have enabled us to conduct high-resolution global atmospheric simulations using a sub-kilometer horizontal grid spacing", without commenting or expanding on whether this should really be undertaken as a scientific endeavor. Just because we can, it does not mean that we should. The authors do not seem to outweigh the pros and cons of conducting such numerical simulations, especially in light of the carbon footprint and computational costs associated with said sub-kilometer scale simulations.

We agree that it is crucial to consider whether conducting high-resolution global atmospheric simulations with a sub-kilometer horizontal grid spacing has additional scientific value.

The accurate representation of convections is an essential part in global climate simulations. Miyamoto et al. (2013) conducted the global simulation with the horizontal grid spacing of 870 m where individual deep convections are explicitly represented using multiple grids and the cumulus parametrization was not used. In such simulation, it is expected to reduce uncertainties associated with the cumulus parametrization. The simulation achieved by using the K computer not only demonstrated the computational feasibility of global sub-meter resolution simulations using recent supercomputers, but also provided important scientific findings. For example, their grid-refinement experiments gave important knowledge about the numerical convergence of statistical properties of deep convections. In addition, by further analyzing the datasets obtained from the 870 m simulation, Miyamoto et al. (2015) elucidated the convection differences in various atmospheric disturbances including the Madden-Julia oscillations, tropical cyclones, mid-latitudinal low-pressure disturbances, and synoptic-scale front.

However, in the high-resolution simulation of Miyamoto et al. (2013), the spatial resolution is insufficient to explicitly represent the turbulent flows in boundary layers and the low-level clouds such as shallow cumuli and we need to rely on the parametrizations. Although such parametrization is crucial to reproduce the global radiation budget in the realistic Earth's atmosphere, they are a source of uncertainty in climate simulations. To decrease the uncertainty and accurately model the climate system, we cannot avoid increasing the spatial resolutions and replacing the parametrization by a strategy with less uncertain parameters, such as LES compared to RANS. In terms of conducting LES precisely, Kawai and Tomita (2021, 2023) claimed that high-order dynamical cores are needed.

The required computational resources are a serious issue as you pointed out. Although global simulations with horizontal grid spacing of 100~200 m appears feasible in exascale computers, this spatial resolution is still in the gray zone with the turbulence. For a "true" global LES with uniformly $O(10\text{ m})$ grid spacing, we will have to wait for the development of zeta-scale computers. To reduce the computational cost in the future global LES, it is necessary to consider sophisticated techniques such as a dynamically adaptive mesh refinement (e.g., Blaise and St-Cyr, 2012) and a super-parameterization (Grabowski, 2016).

Based on your comment, we have modified the statements in the first paragraph of Sect. 1 as "Recently developed supercomputers have enabled us to conduct high-resolution global atmospheric simulations using a sub-kilometer horizontal grid spacing. For example, Miyamoto et al. (2013) conducted a global simulation at a horizontal grid spacing of 870 m and discussed the numerical convergence of statistical properties of deep moist convections. In the near future, this continuous development of computer technology is expected to enable us to perform global simulations using $O(10\text{–}100\text{ m})$ grid spacing (Satoh et al., 2019), which begin to explicitly represent turbulence in the inertial sub-range. Then, large-eddy simulation (LES) is a promising strategy, since in LES, the turbulence in a spatial scale larger than a spatial filter is explicitly calculated, whereas the effect of turbulence in a smaller spatial scale is parameterized using eddy viscosity and diffusion terms. By explicitly representing the large-scale eddies in boundary layers and the low-level clouds such as shallow cumuli, we expect to reduce a source of uncertainty associated with the parameterizations and improve representation of global radiation budget in a realistic Earth's atmosphere."

[References]

- Blaise, S., & St-Cyr, A. (2012): A dynamic hp-adaptive discontinuous Galerkin method for shallow-water flows on the sphere with application to a global tsunami simulation. *Monthly Weather Review*, *140*(3), 978-996. https://doi.org/10.1175/MWR-D-11-00038.1
- Grabowski, W. W. (2016): Towards global large eddy simulation: Super-parameterization revisited. *Journal of the Meteorological Society of Japan. Ser. II*, *94*(4), 327-344. https://doi.org/10.2151/jmsj.2016-017
- Miyamoto, Y., Kajikawa, Y., Yoshida, R., Yamaura, T., Yashiro, H., & Tomita, H. (2013): Deep moist atmospheric convection in a subkilometer global simulation. *Geophysical Research Letters*, 40(18), 4922-4926. https://doi.org/10.1002/grl.50944
- Miyamoto, Y., Yoshida, R., Yamaura, T., Yashiro, H., Tomita, H., & Kajikawa, Y. (2015): Does convection vary in different cloud disturbances?. *Atmospheric Science Letters*, *16*(3), 305-309. https://doi.org/10.1002/asl2.558

2. In the literature review, the authors seem to miss to mention the The Nonhydrostatic Unified Model of the Atmosphere (NUMA), which also successfully used DGM

Thank you for your suggestion. We agree that we should refer to the NUMA as a nonhydrostatic global dynamical core based on the high-order element-based method. In lines 68-70, we have added a sentence as

"In the Nonhydrostatic Unified Model of the Atmosphere (NUMA; Kelly and Giraldo, 2012; Giraldo et al., 2013), which is applicable for both limited-area and global atmospheric simulations, the continuous and discontinuous Galerkin methods are adopted for the spatial discretization."

[References]
- Kelly, J. F., & Giraldo, F. X. (2012): Continuous and discontinuous Galerkin methods for a scalable three-dimensional nonhydrostatic atmospheric model: Limited-area mode. *Journal of Computational Physics*, 231(24), 7988-8008. https://doi.org/10.1016/j.jcp.2012.04.042
- Giraldo, F. X., Kelly, J. F., & Constantinescu, E. M. (2013): Implicit-explicit formulations of a three-dimensional nonhydrostatic unified model of the atmosphere (NUMA). *SIAM Journal on Scientific Computing*, 35(5), B1162-B1194. https://doi.org/10.1137/120876034

3. The authors mention multiple times that they conduct classical numerical experiments to validate their numerical model. However, they seem to confuse Validation with Verification. In Numerical Analysis, the concepts of Verification and Validation (V&V) can be oversimplified in a succinct manner by saying that "verification is solving the equations right" (verifying the numerics) and "validation is solving the right equations" (verifying the physics - often done by comparing model results with actual data given by observations).

Thank you for pointing out our confusion between "validation" and "verification" and explaining the difference in the context of the numerical analysis. To obtain a correct understanding, we also checked the meaning of "V&V" in several Japanese documents. We think that "verification" is appropriate for representing the investigation of model performance using test cases for the dynamical cores. In the revised manuscript, we have replaced "validation" by "verification".

4. The authors seem to have chosen favorable examples/results and have not sufficiently provided explanations on reasons behind some degraded results, such as when a less than theoretical convergence rate was achieved. Also, the use of filters to overcome numerical

instabilities was not always comprehensively justified and their effect on the quality of the results was not extensively elaborated on.

[Figure]

Fig. R1: Impact of the order $p_m$ and the coefficient $\alpha_m$ in the modal filter on the numerical convergence in a linear advection test: (a) $p_m$=64, (b) $p_m$=32, (c) $p_m$=16, and (d) $p_m$=8. In each $p_m$, we changed $\alpha_m$ as 0 (without the filter), $10^{-3}$, $10^{-1}$, and $10^1$. Please note that the results for $\alpha_m$=0 are identical to those obtained for $\varphi_0$=0 in Fig. 1 of our paper.

Through several deterministic tests including the linear advection, gravity wave, and mountain wave tests, our purpose was to check the numerical convergence associated with high-order DGM. Thus, we focused on showing the convergence rate higher than that achieved by the second-order accuracy in the standard test cases. On the other hand, we agree that it is fair to present a situation where there is less benefit of high-order methods. Based on Comment 6, we have added the results for an advection test where a discontinuous tracer profile is advected by a deformation flow. Please see our response to Comment 6.

As for the explanation of the degraded results, it is preferable to elaborate the impact of modal filters on the convergence rate, as suggested by you. We had investigated how much the modal filters contaminate the eddy viscosity with the turbulent model and the energy spectra in Kawai and Tomita (2023). Here, we focused on the issue regarding the low convergence rate. We conducted a linear advection test described in Sect. 3.1 using the modal filters defined in Eq. (41) of the revised manuscript. Figure R1 shows the impact of order ($p_\mathrm{m}$) and the decay coefficient ($\alpha_\mathrm{m}$) in the modal filter on the numerical convergence. The results indicate that the filters can degrade the original convergence rate and increase the numerical errors because the filter diminishes several high modes in the polynomial expansion. If we use high-order modal filters such as $p_m \geq 32$, the degradation of convergence rate was in the range of 1~3 for $p=7$, 11 even when we set sufficiently large values of $\alpha_\mathrm{m}$ such that the highest mode was immediately decayed after one timestep. For $p=3$, the degradation of convergence rate appeared less obvious. However, we note that the errors without the modal filter were much larger compared to those observed in the case of $p=7$, 11. Thus, for $p=3$, the effect of the increased error due to the filters may be more pronounced in the representation of the flow fields. We have mentioned this investigation into the lines 406-415 and have added Fig. R1 as Fig. 2 in the revised manuscript.

5. Section 3, line 331: Can the authors elaborate a little more on why they consider "difficult" to directly evaluate the numerical convergence in those cases?

We would like to mean that the chaotic behavior of the nonlinear systems can diverge the numerical solutions in the long-termed climatological tests such as the Held-Suarez test. In turbulent simulations such as the boundary layer turbulence, smaller-scale structure of the flows appears as the grid spacing decreases (until the spatial resolution reaches the dissipation scale associated with the kinematic viscosity). Thus, it is difficult to evaluate the quality of the numerical solutions using the $L_1$, $L_2$, and $L_\mathrm{inf}$ error norms, which were appropriate for the deterministic tests such as the linear advection, gravity wave, and mountain wave tests.

In lines 371-375 of the revised manuscript, we have modified the corresponding statement as "… it is difficult to directly evaluate the numerical convergence using the error norms defined

in Eq. (42). In the long-termed integration, the chaotic behavior of the nonlinear systems can diverge the numerical solutions. In the turbulent flow simulations, a smaller scale structure becomes more apparent as the grid spacing decreases until the spatial resolution reaches the physical dissipation scale."

6.  In Sec 3.1 for the Linear Advection experiment, I wonder if the authors also verified their solver using a slotted cylinder example. This is often used in the literature because the sharp features of the geometry would particularly challenge the solver. I would appreciate if the authors would conduct such numerical experiments and would compare their convergence rates with the results reported in the literature, e.g., Guba et al. "Optimization-based limiters for the spectral element method" (2014) https://doi.org/10.1016/j.jcp.2014.02.029 looking at the results without limiters. In the same section, regarding the numerical results in Figure 1, the authors have not sufficiently explained why the case with $\alpha = 0$, i.e., no singularity in the coordinates on the cubed-sphere corners, in almost all cases presents larger numerical errors.

We would like to inform you that the purpose in Sect. 3.1 is to test the spatial discretization with the cubed-sphere geometry and check the convergence rate with high-order DGM in the smooth solution. We agree you that the linear advection tests with a discontinuous profile provide important information about the robustness of numerical schemes. For readers with such interests, we have conducted an advection test using the Gaussian hills and slotted cylinder profiles in a deformation flow, which is presented as Case 4 in Nair and Lauritzen (2010). In contrast to the case of the solid body rotation flow, the tracer profile is significantly deformed during one period.

[Results of Case 4 in Nair and Lauritzen (2010) as suggested by you]

Figure. R2 shows the dependence of error norms on the horizontal resolution when the Gaussian hills were given as the initial condition of the tracer. To compare the errors of Guba et al. (2014), hereinafter referred to as G2014, we normalized the errors following Appendix C of Nair and Lauritzen (2010). Because of the infinitely smooth profile, we obtained ($p$+1)-order accuracy for $p$=1, 3, 7, and 11 using the upwind numerical flux without the modal filter. The behavior of $p$+1-order numerical convergence and the magnitude of errors are comparable to G2014 (see the values of $l_1$, $l_2$, $l_\infty$ for "Gauss." with the hyperviscosity (but without the limiter) in Tables 1-4 and Fig. 4 of G2014). In our results, the $L_{\text{inf}}$ error was larger than the unity for $p$=11 in the coarsest spatial resolution. This reflects a numerical instability with the aliasing errors near the static stagnation point of the deformation flow. It occurred when the static stagnation

[Figure]

Fig. R2: Dependence of (a) $L_1$, (b) $L_2$, and (c) $L_{\text{inf}}$ errors after t=12 days on the horizontal resolution in the Case 4 in Nair and Lauritzen (2010) with the Gaussian hills. The green dashed line represents the case of applying a modal filter for $p$=11.

[Figure]

Fig. R3: Dependence of (a) $L_1$, (b) $L_2$, and (c) $L_{\text{inf}}$ errors after $t$=12 days on the horizontal resolution in the Case 4 in Nair and Lauritzen (2010) with the slotted cylinders. The green dashed line represents the case of applying a modal filter for $p$=11.

point was located at the element boundaries and numerical dissipation was not sufficient. By introducing a very weak modal filter with $\alpha_{m,h} = 2.5 \times 10^{-2}$, $p_{m,h} = 64$, we can control the numerical instability (see the green dashed line in the $L_{\text{inf}}$ error of Fig. R2).

Figure R3 shows the dependence of error norms on the horizontal resolution when the slotted cylinders were given as the initial tracer profile. Because of the $C^0$ discontinuous field, we cannot expect the convergence rate to be higher than that in the case of first-order accuracy.

Even when using the high polynomial orders ($p>3$), we obtained at most the first order for the $L_1$ error norm. For the $L_{\text{inf}}$ error, the convergence rate was near the zero order. The behavior of slow numerical convergence and the magnitude of numerical errors were similar to those reported in G2014 (see the error norms for "Cyl." with the hyperviscosity (but without the limiter) in Tables 1-4 of G2014). As seen in the case of Gaussian hills, due to the numerical instability near the static stagnation point, we found very large $L_{\text{inf}}$ errors for the cases of $p=7$, 11 with the coarse spatial resolution. By adopting the modal filter used in the case of Gaussian hills, the numerical instability was suppressed. The corresponding results are represented by the dashed lines in Fig. R3.

[Figure]

Fig. R4: Fig.1 in the previous manuscript which represented the dependence of (a) $L_1$, (b) $L_2$, and (c) $L_{\text{inf}}$ errors after $t=12$ days on the horizontal resolution in a two-dimensional linear advection. However, we incorrectly set the axis angles with the solid-body rotation flow for the cases of $\varphi_0=\pi/4$, $\pi/2$.

[Figure]

Fig. R5: The modified version of Fig. R4 where we correctly set the axis angle with the solid-body rotation flow.

Based on your suggestion, we presented the above mentioned results in Appendix A of the revised manuscript.

[Results of linear advection experiment discussed in Sect. 3.1]

Next, we refer to your comment regarding Fig. 1, which is presented as Fig. R4 in this document. First, we apologize that the axis angles with the solid-body rotation flow were incorrectly set for the cases of $\varphi_0=\pi/4$, $\pi/2$. The results obtained from the modified experiment are shown in Fig. R5. The numerical errors for $\varphi_0=\pi/2$ were identical to those obtained for $\varphi_0=0$. The change is intuitively plausible because, for $\varphi_0=0$, $\pi/2$, there is no essential difference in the movement of advected quantities in the cubed-sphere geometry. For $\varphi_0=0$, $\pi/2$, the numerical errors were still larger compared to $\varphi_0=\pi/4$ when $p=3$ was used. Note that the error norms shown in Fig. R5 are the values at a certain time. Figure R6 shows the temporal series and the moving average of $L_2$ error norm for the case of $\Delta_{h,eq}=78$ km for $p=1$, 3, and 7 and $\Delta_{h,eq}=104$ km for $p=11$. As you mentioned, the moving average indicates that the numerical errors for $\varphi_0=0$, $\pi/2$ tend to be larger than those obtained for $\varphi_0=\pi/4$. Similar results are seen in previous studies (e.g., see Table 2 of Ullrich et al. (2010)).

Although we investigated the temporal evolution and spatial distribution of the errors, we have not yet confirmed the reason. As one possible reason, the change in the grid aspect ratio on the tracer path is small for $\varphi_0=\pi/4$ compared to $\varphi_0=0$, $\pi/2$ (see the case of gnomonic projection in Fig. 3 of Rančić et al. (2017)). However, because this is a matter of speculation, we would like to keep the description in the manuscript to the fact that the corner singularity with the cubed-sphere coordinates is well handled without any numerical instability in the DGM framework. In lines 402-403 of the revised manuscript, we have modified the corresponding statement as "The errors for $\varphi_0=\pi/4$ radians can be smaller than those observed for $\varphi_0=0$, $\pi/2$ radians (e.g.,

[Figure]

Fig. R6: (a) Time series of the $L_2$ error obtained from $\Delta_{h,eq}=78$ km for $p=1,3$, and 7 and $\Delta_{h,eq}=104$ km for $p=11$ in a two-dimensional linear advection. (b) the moving average for 6 days. Note that the solid and dashed lines represent the cases of $\varphi_0=0$, $\pi/4$, respectively.

*p*=3). The reason has not been confirmed, but we have found similar results in previous studies (Ullrich et al., 2010)."

[References]

- Guba, O., Taylor, M., & St-Cyr, A. (2014): Optimization-based limiters for the spectral element method. *Journal of Computational Physics*, 267, 176-195.
  https://doi.org/10.1016/j.jcp.2014.02.029
- Nair, R. D. & Lauritzen, P. H. (2010): A class of deformational flow test cases for linear transport problems on the sphere. *Journal of computational physics*, *229*(23), 8868-8887.
  https://doi.org/10.1016/j.jcp.2010.08.014
- Rančić, M., Purser, R. J., Jović, D., Vasic, R., & Black, T. (2017): A nonhydrostatic multiscale model on the uniform Jacobian cubed sphere. *Monthly Weather Review*, *145*(3), 1083-1105.
  https://doi.org/10.1175/MWR-D-16-0178.1
- Ullrich, Paul A., Christiane Jablonowski, & Bram Van Leer. (2010): High-order finite-volume methods for the shallow-water equations on the sphere. *Journal of Computational Physics*, 229, 6104–6134.
  https://doi.org/10.1016/j.jcp.2010.04.044

7. Sec 3.3, line 443: authors mention the modal filter as one potential reason for the degraded sub-optimal convergence. Shouldn't it help instead? Can they elaborate on this further?

As we described in the reply to Comment 4, the modal filters contribute to prevent the numerical instability, while also contaminating the numerical accuracy in calculating the smooth flow fields. The essential cause is that the filters shave off the high modes in the elementwise polynomial expansion. In Appendix A3 of previous manuscript, we indicated the degraded suboptimal order due to the modal filters. When requiring a convergence rate with a certain order of the accuracy, we need to increase the polynomial order according to the intensity of the filters.

Based on your comment, we have added statements as

"Because the modal filters shave off the high modes in the polynomial expansion, the convergence rate can be degraded. When requiring a convergence rate with a certain order accuracy, we need to increase the polynomial order according to the filter intensity."

in lines 499-501 of the revised manuscript.

8. Section 3.5, Caption of Figure 11: Can the authors explain why they presented numerical results for the highest resolution case with a temporal average over only 300 days as opposed to 1000 days for the other cases? Was it too computationally expensive to perform the highest resolution simulation over 1000 days, or the model presented difficulties over 300 days, such

as it suffered from numerical instabilities/crashes?

The initial reason for the integration time in the highest resolution experiment case shorter than that in the other cases is due to the computational resources available to us. We agree that it is better to show the results based on the simulation over the 1000 days. Thus, we extended the temporal integration of highest resolution experiment case.

Unfortunately, to continue the stable temporal integration, we need to temporarily increase a horizontal coefficient of the modal filter by 20 % for 20 days during the 1000-day integration. In the revised manuscript, the fact has been mentioned in the caption of Table 9. In addition, using the obtained results, we have replaced the corresponding figures (Figs. 11 and 12). Because no qualitative change was observed, we have retained our claim in the manuscript.

**Technical corrections:**

First, the authors are deeply grateful for many suggestions regarding our grammatical issues.

1. Line 5: "the impact of high-order DGM on atmospheric flows was investigated". I would rephrase this with another sentence along the lines of: "the impact of high-order DGM on the quality or accuracy of the numerical simulations of atmospheric flows was investigated"

   Based on your comment, we have modified the corresponding sentence as
   "… the impact of high-order DGM on the accuracy of the numerical simulations of atmospheric flows was investigated."
   in line 5 of the revised manuscript.

2. Line 16: "In the near future"

   We have modified this in line 18 of the revised manuscript.

3. Line 18: "Then, large-eddy simulation (LES) is a promising strategy, since in LES"

   We have modified this in line 21 of the revised manuscript.

4. Line 33-34: Rephrase "In the context of DGM, KT2023 investigated the problem with the order of accuracy necessary for LES"

   Thank you for pointing out an unclear statement. We have revised this statement as
   "KT2023 extended the discussion presented in KT2021 to the DGM framework and investigated a polynomial order necessary for precisely conducting LES."
   in lines 40-41 of the revised manuscript.

5. Line 35: Add plural for generic or non specific countable nouns in English, i.e., "modal filters

are used", or add an article if you want to use singular nouns

Considering the intent here, we have modified "filter" to the plural in line 42 of the revised manuscript.

6. Line 36-37: Authors mention "2000-2010" but then they survey literature belonging to the following decade

We apologize for the error in the wording of the age. We should write it as "2000's-2010's". We have modified it in lines 43-44 of the revised manuscript.

7. Line 64: Please introduce the FDM acronym before using it

Thank you for pointing it out. The term "finite difference method" appears only once, so we have decided not to use the abbreviation in the revised manuscript.

8. Line 73: "the impact of high-order DGM on the atmospheric flows". I would rephrase this, similar to the Abstract sentence.

Based on your suggestion, we have revised the corresponding statements as "... the impact of high-order DGM on the numerical accuracy of atmospheric flow simulations" in lines 113-114 of the revised manuscript.

9. Line 71-72: "We focused" and then "We attempt". Please check grammar consistency of temporal tenses throughout the text

Thank you for pointing out the tense inconsistency in the sentences. The past tense should be used here, and we have modified it in line 112 of the revised manuscript. In addition, we have reconsidered the temporal tense throughout the text.

10. Line 86: "required" -> "requiring"?

Thank you for your suggestion. We apologized that "for" was missing as other reviewer pointed out. We have corrected it in line 96 of the revised manuscript.

11. Line 155: "angular velocity of the planet"

We have modified it in line 181 of the revised manuscript.

12. Line 156: "In the numerical experiments".

We have added an indefinite article it in line 182 of the revised manuscript.

13. Line 172: "is essentially the same as"

We have added an indefinite article in line 198 of the revised manuscript.

14. Line 175: "In the absence of a vertical"

We have modified it in line 201 of the revised manuscript.

15. Line 183: "D is the divergence of the three-dimensional velocity"

We have added an indefinite article in line 209 of the revised manuscript.

16. Line 194: "For further details on the turbulence model"

We have added an indefinite article in line 219 of the revised manuscript.

17. Line 227: "For the numerical flux of the inviscid terms"

We have added an indefinite article in line 256 of the revised manuscript.

18. Line 229: reword "considered"

We have replaced this word by "taken into account" in line 259 of the revised manuscript.

19. Line 230: "transformations, and is formulated as"

Thank you very much for your comment and we have reconsidered this sentence. In lines 258-259 of the revised manuscript, we have modified it as

"Previous studies (e.g., Li et al., 2020) formulated the Rusanov flux taken into account the horizontal and vertical coordinate transformation as .."

20. Line 270: "restrict the time step" (remove "to")

We have removed to "to" in line 304 of the revised manuscript.

21. Line 280: "in the case of the diagonally implicit RK scheme"

We have added an indefinite article in line 315 of the revised manuscript.

22. Line 288: "To obtain the solutions of the nonlinear equation system"

We have added an indefinite article in line 322 of the revised manuscript.

23. Line 291: "In the case of the collocation approach"

We have added an indefinite article in line 325 of the revised manuscript.

24. Line 301: "When using the HEVI approach"

We have added an indefinite article in line 335 of the revised manuscript.

25. Line 302: "entries of the matrices"

    We have added an indefinite article in line 336 of the revised manuscript.

26. Line 304: Rephrase with: "For high-order methods, numerical instability is likely to occur in advection-dominated flows, because the discrete advection operator is oscillatory."

    We have reconsidered the statement. In line 339-340 of the revised manuscript, we have modified it as

    "For high-order DGM, numerical instability likely to occur in advection-dominated flows because the numerical dissipations with the upwind numerical fluxes weaken."

    Although you suggested the statement "because the discrete advection operator is oscillatory", we would like to mention that the inherent numerical dissipations with the upwind numerical fluxes are weak at a wavelength range longer than 2~4 grid lengths when a large polynomial order is used. Thank you very much for your suggestion.

27. Line 310: Remove "represents" or "is"

    We apologize the careless mistake. We have modified it in line 345 of the revised manuscript.

28. Line 317: "the order of the filter"

    We have added an indefinite article in line 351 of the revised manuscript.

29. Line 318: "at the final stage of the RK scheme"

    We have added an indefinite article in line 352 of the revised manuscript.

30. Line 320. Rename Section 3 "Verification of the dynamical core"

    We have used "verification" in the section title.

31. Line 322: "we mainly focused on the impact of the polynomial order on the effective". There are several missing articles throughout the text. I stopped correcting all of them after some point. The authors should more carefully proof-read for English correctness.

    We have added an indefinite article in line 375 of the revised manuscript. Thank you for suggesting our manuscript need further proof-read for English correctness. In the revised manuscript, we have reconsidered the use of articles. Then, the modifications have been checked by Editage.

32. Line 339: I know $\alpha$ is used in the literature to denote the angle between the axis of the solid body rotation and the North pole. However, the authors should be careful because they also previously used $(\alpha, \beta, \zeta)$ for the local coordinates on the cubed-sphere.

 Thank you for pointing out we should change the symbol for denoting the angle between the axis of the solid body rotation and the North pole. In the revised manuscript, to denote it, we have used $\varphi_0$. In addition, we have replaced the symbol of the latitude by $\varphi$.

33. Line 379: "errors"

 We have modified it in line 432 of the revised manuscript.

34. Line 381: "a modal filter" and "was investigated"

 We have modified them in line 434 of the revised manuscript.

35. Line 392: "except for the horizontal wind"

 We have added "for" in line 445 of the revised manuscript.

36. Line 427: "details on the sponge layer"

 We have modified it in line 480 of the revised manuscript.

37. Line 446-447 avoid repetition of "includes" and "included" in the same sentence by using a synonym

 Thank you for pointing out the repetition. We have modified the corresponding statement as "It includes small-scale structures such as front and filament formations."
 in lines 503-504 of the revised manuscript.

38. Line 455: "stretched"

 We have modified it in line 512 of the revised manuscript.

39. Line 473: "evaluation of the horizontal resolution"

 We have modified it in line 530 of the revised manuscript.

40. Line 501: "by using similar spatial resolution"

 Thank you for your suggestion and other reviewer also commented this statement. We have modified this sentence as "nearly the same horizontal spatial resolution" in lines 559-560 of the revised manuscript.

41. Section A3: reword the section title "Investigation on the degradation of the optimal numerical convergence"

   Thank you very much for indicating a better representation. We have modified the section title in the revised manuscript.

42. Figure A2: remove bold text in caption

   We have changed it to a normal style in the revised manuscript.

43. Figure A3: remove bold text in caption

   We have changed it to a normal style in the revised manuscript.

44. Line 741: I am sorry, but even in the acknowledgement sentence in which the authors thank the company they used for the English editing, there is a grammatical error "We would like to thank Editage for the English language editing"

   We apologize the grammatical issues in the manuscript. In the revised manuscript, we have made further effort for English correctness and the proofreading by Editage has been done again. We greatly appreciate your comments.

**Further notable modifications made in the revised manuscript**

- In the Held-Suarez test, we adjusted the vertical element size for the case of $\Delta_{h,eq}$=280 km using $p$=7 for the vertical stretching to be consistent to all cases using $p$=7 and corrected the vertical grids for the case of $\Delta_{h,eq}$=52 km using $p$=3. In addition, we extended the temporal integration of the highest resolution case to 1000 days. In Figs. 10, 11, and 12 of the revised manuscript, we have presented the results of the modified experiments. Fortunately, because there were no essential changes, we did not change the claims made in the previous manuscript.

- We added a new appendix, Appendix C. Here, we have described the vertical stretching used in the baroclinic wave and Held-Suarez tests.

---

## Author Comment (AC2)

**Response to reviewer2 (egusphere-2024-1477 manuscript)**

Thank you for your careful review. We appreciate your thoughtful comments to improve our paper. We copied your comments in the blue text and have provided our responses in the black text. We have revised the manuscript according to your suggestions. Our point-by-point responses to the reviewers' comments are provided below. We hope that these improvements satisfactorily address the issues pointed out by you.

This paper presents a global nonhydrostatic dry atmospheric dynamical core discretised using high order discontinous Galerkin methods. The motivation for using high order methods comes from the authors' previous work which showed that when using an LES turbulence model, the order of accuracy of the spatial discretisation needs to be sufficiently high. This also motivates the use of DG methods as they avoid the large computational stencil required for high order grid point methods. The new model is tested using a range of well known 3D problems, with some changes made to avoid the development of small scales in order to make convergence analysis possible.

The work presented here is of interest and has been carefully done, however it is not clear exactly what is new about this model and how it relates to previous DG / high-order discretisations. In some places additional clarifications and/or references are required and the language needs to be more precise - e.g. words like "attempt" or "about" should be avoided. In many places the results are qualified with "about" and I wonder if it is possible to be more precise, or more confident in what is described. I have highlighted these places below.

I recommend numbering all the equations - this makes it much easier when others discuss your paper!

Thank you for the valuable suggestion. In the revised manuscript, we have numbered all the equations.

In the results section, the information on the number of elements, polynomial orders and resulting equatorial resolution is hard to read
- could this information be summarized in a table for each test?

We have summarized the number of elements, the polynomial order, and the resulting equatorial resolution in Table 5 of the revised manuscript. In addition, we have simplified the description of spatial resolution in the main text.

Many of the comments below require only minor changes to clarify the text. However, I have

selected "major revisions" because I have requested a lot of clarifications and in particular, I think it is very important that the novelty of the method is clarified with reference to other DG dycore publications.

As for the novelty, please see our response to your comment for line 58 (Comment 11).

**Introduction:**

1. line 18: "inertia subrange" should be "inertial subrange"

We have modified it in line 20 of the revised manuscript.

2. lines 26-28: This sentence is confusing and the `e-folding time' is not defined or referred to again. As the use of high order methods is motivated by this study, I think it is worth adding another sentence or two here to adequately explain this previous result.

Thank you for your suggestion. As in the introduction of Kawai and Tomita (2023, MWR), we agree with you to explain our numerical criteria more adequately. In lines 30-34 of the revised manuscript, we have modified the corresponding statements as

"In particular, the study derived two ratios associated with numerical diffusion and numerical dispersion: the ratio of decay time with the SGS terms to that of the numerical diffusion error terms and that of phase speed due to the error in advection terms to that of the SGS terms. Moreover, we pointed out that the advection scheme requires at least seventh- or eighth-order accuracy to ensure that both ratios are less than $10^{-1}$ at wavelengths longer than eight grid lengths for grid spacing simulations of $O(10)$ m."

3. line 30: "which is recognized as a local spectral method" - what do you mean by "local spectral method" or can you cite something here?

Our meaning of the term "local spectral method" is that if we see the formulation of nodal DGM in the elementwise level, the discretization strategy looks like a spectral collocation method. For example, Chen et al. (2013) referred the spectral element method and DGM used in Giraldo and Restelli (2008) as a local spectral method. However, in the revised manuscript, we have removed this sentence because we do not consider it to be essential.

[References]
- Chen, X., N. Andronova, B. Van Leer, J. E. Penner, J. P. Boyd, C. Jablonowski, & S. Lin (2013): A Control-Volume Model of the Compressible Euler Equations with a Vertical Lagrangian Coordinate. *Monthly Weather Review.*, 141, 2526–2544. https://doi.org/10.1175/MWR-D-12-00129.1
- Giraldo, F. X., & M. Restelli (2008): A study of spectral element and discontinuous Galerkin methods

for the Navier–Stokes equations in nonhydrostatic mesoscale atmospheric modeling: Equation sets and test cases. *Journal of Computational Physics.*, **227**, 3849–3877. https://doi.org/10.1016/j.jcp.2007.12.009

4. line 35: Again this description of the previous result needs either simplifying or clarifying - as it is written it provokes questions: What was special about the case with upwinded numerical flux and sufficiently high order modal filter? What is "sufficiently high order" in this case? What happens more generally?

Thank you for your comment. The upwind numerical flux and modal filters are used to ensure numerical stability. The reason why we mentioned "sufficiently high order" is because the low-order filter can contaminate the flow structure at the wavelength range longer than the eight grid lengths even for a small decay coefficient of the filter. Based on your comment, we have modified the corresponding sentence as

"…when the upwind numerical fluxes and sufficiently scale-selective modal filters are used to ensure numerical stability."
in lines 41-42 of the revised manuscript.

5. line 44: Clarify what you mean by "effective resolution significantly apart from the grid spacing". What is your definition of "effective resolution".

Our definition of "effective resolution" is the shortest wavelength fully resolved by discretization methods based on the dependence of numerical diffusion and numerical dispersion errors on the wavelength (e.g., Walters (2000); Kent et al. (2014)). Based on your comment, we have modified the corresponding statement as

"… shortest wavelength fully resolved by discretization methods, so called effective resolution, ...".
in line 52 of the revised manuscript.

[References]

- Walters, M. K. 2000: Comments on "The differentiation between grid spacing and resolution and their application to numerical modeling". *Bulletin of the American Meteorological Society*, *81*(10), 2475-2477. 10.1175/1520-0477(2000)081<2475:CAACOT>2.3.CO;2
- Kent, J., Whitehead, J. P., Jablonowski, C., & Rood, R. B. (2014). Determining the effective resolution of advection schemes. Part I: Dispersion analysis. *Journal of Computational Physics*, 278, 485-496. https://doi.org/10.1016/j.jcp.2014.01.043

6. line 46: "eight grid spacing" might be clearer as "eight grid lengths". Do you have a reference

for this claim?

Thank you for your suggestion. We have cited Kent et al. (2014). In lines 53-54 of the revised manuscript, we have modified the statement as

"The low-order spatial scheme typically leads to significant discretization errors at wavelengths shorter than eight grid lengths (Kent et al., 2014)."

[References]

Kent, J., Whitehead, J. P., Jablonowski, & C., Rood, R. B. (2014). Determining the effective resolution of advection schemes. Part I: Dispersion analysis. *Journal of Computational Physics*, 278, 485-496. https://doi.org/10.1016/j.jcp.2014.01.043

7.  line 49: "than that in plane domains" the "that" is not needed

In the revised manuscript, we have removed "that".

8.  line 50: "archive" - I think you mean "achieve"

We have modified it in line 58 of the revised manuscript.

9.  line 52: "the pole problem" - add a short description of this for those not familiar with the issue.

In lines 60-61 of the revised manuscript, we have modified this statement as

"the problem of restrictive timestep near the poles due to the convergence of meridians".

10. line 53: "we can suffer..." the "we" here is confusing as I don't think you mean the work you are describing in the paper. I suggest rephrasing as "However, significantly high resolution global simulations can suffer from..."

Thank you for your suggestion. We have modified the statement as "However, high-resolution global simulations can suffer from.." in line 61 of the revised manuscript.

11.  1ine 58: "The Climate Machine..." What order is used in this model? How is what you have done different? This section describes a some other DG dycores but it is not clear how the work presented here differs from each one, especially as line 70-71 says "This study includes several progresses from previous studies..." Firstly, "progresses" is not quite the right word here. You could say "This study build on progress from previous studies..." or "This model includes several algorithmic features from previous studies..." but more importantly it needs to be clearer what "progresses" or "features" and what previous studies.

Although Sridhar et al. (2022) showed the results using the polynomial order $p$=4, we expect

that arbitrary polynomial orders are available in the ClimateMachine based on their codes in https://github.com/CliMA/ClimateMachine.jl.

The spatial discretization used in SCALE-DG follows a nodal DGM (e.g., Hesthaven and Warbuton, 2007) and the discretization is mostly similar with that in previous studies with DG dynamical cores such as ClimateMachine and NUMA. Although this study has little novelty in the context of numerical methods of DGM, we consider that the following points to be our unique contributions:

1) Introduction of a turbulent model to a global DG dynamical core on cubed-sphere coordinates:

   To construct a global LES model, we formulated SGS eddy viscous and diffusion terms with a Smagorinsky-Lilly type turbulent model on cubed-sphere geometry in the DGM framework. Several previous studies (Ullrich, 2014; Guba et al., 2014) presented strategies for the vector Laplacian operator in element-based global shallow water models on the cubed-sphere coordinates. For our purpose of introducing the turbulent model, we treated the Laplacian operator acting on the component of vector fields in the cubed-sphere coordinates and the eddy viscosity dependent on local flow fields. Furthermore, we introduced the turbulence model to our DG dynamical core and verified its behavior by conducting an LES experiment of idealized planetary boundary layer turbulence.

2) Modification of test cases for high-order dynamical core:

   We modified existing test cases to investigate the numerical convergence associated with high-order dynamical cores. When using the totally second-order dynamical cores, due to relatively large discretization errors, the problem of ill-posed experimental setting might not be essential. However, we modified the experimental setup to evaluate numerical features, such as the convergence rate, of high-order dynamical cores.

3) Evaluation of numerical convergence with global dynamical core based on DGM:

   By conducting several standard tests, we quantitatively evaluated the numerical convergence of a global nonhydrostatic dynamical core based on DGM and indicated the high-order convergence rate. Although such investigations for regional DG dynamical cores can be found (e.g., Giraldo and Restelli, 2008; Bardar et al., 2013; Blaise et al., 2016), few studies are available on global DG dynamical cores.

Based on your comment, we have modified the corresponding statements in lines 86-88 of the revised manuscript as

"By building on progresses from the previous studies showing the applicability of the element-based methods to atmospheric flow simulations, this study attempted to develop a high-order global dynamical core using a nodal DGM both horizontally and vertically for future global atmospheric simulations with O(10–100 m) grid spacing."

Furthermore, we have modified the statement before mentioning our progresses in lines 89-92 of the revised manuscript as

"Although the numerical methods used in our dynamical core are similar to those used in previous studies that developed global DG dynamical cores such as NUMA and ClimateMachine, we consider that the following points are the unique contributions of the current study: 1)…"

Because we consider that the most important contribution is the development of a high-order global dynamical core for LES, we have mentioned it first in the three points.

[References]

- Blaise, S., Lambrechts, J., & Deleersnijder, E. (2016): A stabilization for three-dimensional discontinuous Galerkin discretizations applied to nonhydrostatic atmospheric simulations. *International Journal for Numerical Methods in Fluids*, *81*(9), 558-585. https://doi.org/10.1002/fld.4197

- Brdar, S., Baldauf, M., Dedner, A., & Klöfkorn, R. (2013): Comparison of dynamical cores for NWP models: comparison of COSMO and Dune. *Theoretical and Computational Fluid Dynamics*, *27*, 453-472. https://doi.org/10.1007/s00162-012-0264-z

- Giraldo, F. X., & Restelli, M. (2008): A study of spectral element and discontinuous Galerkin methods for the Navier–Stokes equations in nonhydrostatic mesoscale atmospheric modeling: Equation sets and test cases. *Journal of Computational Physics*, *227*(8), 3849-3877. https://doi.org/10.1016/j.jcp.2007.12.009

- Guba, O., Taylor, M. A., Ullrich, P. A., Overfelt, J. R., & Levy, M. N. (2014). The spectral element method (SEM) on variable-resolution grids: Evaluating grid sensitivity and resolution-aware numerical viscosity. *Geoscientific Model Development*, *7*(6), 2803-2816. https://doi.org/10.5194/gmd-7-2803-2014

- Ullrich, P. A. (2014): A global finite-element shallow-water model supporting continuous and discontinuous elements. *Geoscientific Model Development*, *7*(6), 3017-3035. https://doi.org/10.5194/gmd-7-3017-2014

12. line 76: "provide a chance to modify the experimental setup" - Make it clear that the standard

tests were modified for a particular reason related to your aims and the features of your method.

Based on your comment, we have modified the corresponding statements as

"We modified experimental settings of idealized test cases to demonstrate the numerical convergence with high-order dynamical cores."

in lines 106-107 of the revised manuscript.

13. line 79: "Even when the aim of this study..." I am not sure what this sentence means.

We would like to mean that an evaluation framework using high-order dynamical cores would be beneficial even when research interests are not in the dynamics. In line 110 of the revised manuscript, we have modified the statement as

"Even when research interests do not include the dynamics, …".

14. line 86: should be "cost required for numerical stabilization" (the "for" is missing)

We have added "for" it in line 96 of the revised manuscript.

15. line 88: "semi-discretization" should be "semi-discretized"

We have modified it in line 103 of the revised manuscript.

16. line 89: "which is an extension of..." - more detail would clarify the novelty, e.g. "which extends... by..."

Thank you for your comment. To clarify the novelty, we have modified the statement as

"In particular, we extended a numerical experiment of idealized planetary boundary turbulence used in regional plane models (KT2021 and KT2023) to spherical geometry by slightly changing the initial condition."

in lines 105-107 of the revised manuscript.

**Model description:**

17. line 101: "the Jacobian are denoted" - "are" should be "is". On line 104 you give the expression for the Jacobian of the vertical coordinate transformation - you could do the same here for the horizontal coordinate transformation, for consistency and clarity.

We have replaced "are" by "is" in line 130 of the revised manuscript. Moreover, as suggested, we have presented the expression for the Jacobian and metric tensor of the horizontal coordinate transformation as

"The horizontal Jacobian is defined as $\sqrt{G_h} = |G_h^{ij}|^{-1/2}$."

18. line 109: This clarification of the different notation for the coordinate variables is distracting - can you move it to where you actually use this different notation for the first time (I think in the next section (2.2), line 180 onwards)?

Thank you for your suggestion. However, because the notation $(\xi^m)$ is used in the horizontal pressure gradient terms (in Eq. (8) of the revised manuscript), we think that it would be better to position the clarification near the statements that introduce the coordinates.

19. equation 1: should S_{SGS} also depend on grad(q) (as in line 137)?

Thank you very much for pointing out $\nabla q$ should be added as a dependent variable of $S_{SGS}$. We have corrected it in the revised manuscript.

20. line 151: "where $\delta_S$ is an index..." I think it is a switch rather than an index.

Thank you for suggesting a better word. We have replaced "an index" by "a switch" in line 177 of the revised manuscript.

21. line 208: "effective horizontal grid spacing" - how is this related to the "effective resolution" you talk about elsewhere? Or is it just the spacing between the nodes? In which case, is "effective" the right word?

We agree with you that the term "effective grid spacing" is possible to confuse the term "effective resolution". We are referring to a representative grid spacing which is equivalent to that in the grid-point methods. In lines 235-236 of the revised manuscript, we have modified the statement as

"We defined a representative grid spacing at the equator which approximately corresponds to that in the grid-point methods as …"

22. line 253: should the j be a second subscript of s? it looks like it isn't.

Thank you very much for suggesting our mistake. The second index of $s$ should be an index of the element face associated with the gradient operator in the $\tilde{x}^j$-direction, i.e., $f'$. We have modified it in line 282 of the revised manuscript.

23. line 270: "severely restrict to the timestep" should be "severely restrict the timestep"

We have removed "to" in the revised manuscript.

24. line 319: Should ¥alpha_m be ¥alpha_i where i is the index of the highest mode? Or is m the index of the highest mode?

We apologize that we used inappropriate symbol for the decay coefficient for the modal filter and confused the readers. We should denote the symbol as $\alpha_m$ because we assume it is independent on the mode. In Eq. (41) of the revised manuscript, $\alpha_i$ has been replaced by $\alpha_m$. We appreciate you pointing it out

We would like to inform you that the amplification factor with the cutoff matrix for the highest mode ($i = p$) is $\sigma_p = \exp(-\alpha_m)$ during one timestep. Thus, we described the corresponding decay time scale for the highest mode as $\Delta t/\alpha_m$ approximately.

**Validation of dynamical core:**

25. line 332: as "for this test case" after "Thus" to clarify. Also, do you mean "by focussing on the energy spectra"? This would be clearer if you define what you mean by "effective resolution" and how it relates to the spectra.

Thank you for the valuable suggestion. As mentioned in one of previous replies, our definition of the effective resolution is the shortest wavelength fully resolved by discretization methods. In the energy spectra, we investigated the shortest wavelength at which the spectra begin to separate from that in the reference experiment with the highest spatial resolution. In lines 375-376 of the revised manuscript, we have modified the statement as

"Thus, for the test cases, we mainly investigated the impact of the polynomial order on the shortest wavelength at which the energy spectra began to separate from that in the reference solution."

26. line 348: "we set to D=..." should be "we set D=..."

We have removed "to" in line 391 of the revised manuscript.

27. line 355: "Figure 1 shows the numerical errors..." from this I was expecting a spatial plot of the error - the figure shows the dependence of the errors at 12 days on horizontal resolution (as described in the caption) - this description should be in the text too. I'm not sure that the qualifying "about" is necessary in the next sentence.

In line 396 of the revised manuscript, we have modified the statement as

"Figure 1 shows the dependence of the $L_1$, $L_2$, and $L_{\text{inf}}$ errors at 12 days on the horizontal spatial resolution."

In addition, we have removed "about" by modifying the corresponding sentence as

"…, we obtained $p+1$-order spatial accuracy for $p=1$, 3, and 7. For $p=11$, …"

28. line 360: "there is less difference between the angles" - less difference in what?

Our intended meaning is that the numerical errors for $p=1$ are almost independent of the angle

of the rotation axis. We have modified the statement in lines 401-402 of the revised manuscript as

"the numerical errors were almost independent of the angle of the rotation axis."

29. line 367: "This study considered..." should be "This study considers..." or even "This study presents..."

Thank you for your suggestion. In line 421 of the revised manuscript, we have modified the statement as "This study presents...".

30. line 375: "effective grid spacing" - is this the same as N_{e, h} as described in the previous test setup? Make sure you are consistent.

Please note that $N_{e,h}$ is the number of finite elements in the horizontal direction mentioned in Sect. 2.3. We agree that the description of spatial resolution should be consistent in Sect. 3. Thus, in the revised manuscript, we have removed the extra information that explained the number of DOF in the one-dimensional direction on one cubed-sphere panel.

31. line 378: "we set the Courant number against the..." What does this mean? "against" doesn't make sense here - do you mean the acoustic Courant number? Can you define C_{rh, cs} with a formula?

Thank you for your suggestion. We were referring to the acoustic Courant number associated with the horizontally propagating sound waves.

We agree to present the formula for defining the Courant number. We have added the definition in Sect. 2.4 as follows:

"We introduce two types of Courant number, which are used to explain the timestep setting in Sect. 3. For the horizontal advection test, the advective Courant number associated with the horizontal wind is defined as $C_{r,adv} = U_0 \Delta t / \Delta_{h,eq}$ where $U_0$ is the representative wind speed. For other numerical experiments, the acoustic Courant number associated with the sound wave propagation is defined as $C_{r,c_s} = c_s \Delta t / \Delta$, where $\Delta$ is the effective grid spacing; In particular, for the HEVI approach, $\Delta = \Delta_{h,eq}$."

in lines 297-301 of the revised manuscript. Then, using this definition, we have modified the statement in Sect. 3.2 as

"For the HEVI scheme, we set the timestep such that $C_{r,c_s} = 1.34 \times 10^{-1}$ for p=1,3,7 and ..."

in lines 431-432 of the revised manuscript.

32. line 402: "is well known..." Do you have any citations here?

Thank you for your suggestion. In line 456 of the revised manuscript, we have cited Zängl (2012) as a reference which mentioned the problem of the pressure gradient terms when using the basic terrain-following coordinate and the influence of discretization errors. In particular, the introduction section of Zängl (2012) has summarized the problem.

[Reference]

Zängl, G., 2012: Extending the Numerical Stability Limit of Terrain-Following Coordinate Models over Steep Slopes. *Mon. Wea. Rev.*, **140**, 3722–3733. https://doi.org/10.1175/MWR-D-12-00049.1

33. Figure 4: The x axis labels should be the same units in each figure.

We agree that it is better to use the same units in the figures. In Fig. 5(a) of the revised manuscript, we have determined to use [km] as the unit of the lateral coordinate and denoted the longitude using a secondary axis.

34. line 422: "at about z < 15km" - why "about"?

This is because a top elevation of the computational region unaffected by the vertical stretched grid is slightly different depending on the polynomial order and the number of vertical elements. Because it is not essential to our discussion about the numerical convergence and we mentioned the vertical stretching in the explanation of the sponge layer, we have removed this sentence.

35. line 423: "a fully explicit" - do you mean the HEVE scheme described earlier? If so, then refer to it.

Yes, we mean the HEVE scheme described in Sect. 2.4.2. In lines 476-477 of the revised manuscript, we have modified the corresponding statements as

"... RK scheme described in Sect. 2.4.2. For the HEVE scheme, we set the timesteps such that ..."

36. line 424: "against the" - same comment as earlier - do you mean acoustic Courant number?

Yes, we are referring to the acoustic Courant number. In lines 476-477 of the revised manuscript, we have modified the statement as

"For the third-order HEVE scheme, we set the timesteps such that $C_{r,c_s} = 2.63 \times 10^{-1}$."

37. line 422: "at about z > 15km" - why "about"?

The reason was described in our response to Comment 34. Because this statement referred to a sponge layer, we reconsider that the lowest altitude at which the decay coefficient is larger than zero is more important information. Thus, in lines 478 of the revised manuscript, we have modified the statement as

"… by introducing a sponge layer at z>15 km where the vertical element size linearly increases with the altitude."

38. line 426: "the 1/4 sector" - what is this?

[Figure]

R1: The distribution of lateral sponge layer which corresponds to the blue region.

Thank you for pointing out the wording. We wanted to mean a spherical Lune with the dihedral angle of $\pi/2$ radian as shown in the blue region of Fig. R1. In the revised manuscript, we have removed the phrase because the detail of the lateral sponge layer is described in Appendix B.2 of the revised manuscript.

39. line 429: "to ensure the numerical stability" should be "to ensure numerical stability; "which are summarized" should be "which is summarized"

Thank you for your suggestion. We have modified it in lines 481-482 of the revised manuscript.

40. line 437: "We consider..." I'm not sure what this sentence means.

We carefully studied the difference between the numerical solution obtained by our global dynamical core (Fig. 4(a) of the previous manuscript) and the linear analytic solution on a flat plane (Fig. 4(b) of the previous manuscript). For example, the vertical wavelength of the large-scale wave for the global dynamical core is shorter compared to the linear analytic solution on a flat plane.

[Figure]

R2: The spatial distribution of vertical wind from a mountain wave test with a Schär mountain: (a) Numerical solution obtained from a regional dynamical core obtained from $(\Delta_h,\Delta_v)$=(625 m, 500 m), (b) Two-dimensional linear analytic solution on a flat plane.

When using our regional dynamical core based on the two-dimensional nonlinear Euler equations with the gravity on a flat plane, the obtained numerical solution well reproduced the linear analytic solution as shown in Fig. R2. Thus, we consider that a cause of the difference in

Fig. 4(a), (b) of the previous manuscript is related to the spherical geometry. If we increase the planetary radius while unchanging the spatial scale of the mountain, the two wave patterns would become closer.

In lines 489-492 of the revised manuscript, we have modified the corresponding statement as "For example, the vertical wavelength of the large-scale wave in Fig. 5(a) is shorter compared to Fig. 4(b). Based on the consideration using our regional dynamical core, a cause of the difference is related to the spherical experimental setup. Thus, we expect this difference to decrease as the planetary radius increases while the spatial scale of the mountain remains unchanged."

41. line 446: "in mid-latitude" should be "in the mid-latitudes"

We have modified this sentence in line 503 of the revised manuscript.

42. line 447: Remove "are included"

We apologize our carelessness. We have removed the extra words.

43. line 450-1: should be "the adiabatic inviscid primitive equations"

We have modified it in lines 507-508 of the revised manuscript.

44. line 455: "stretch" should be "stretched" and could you please specify what stretching you used?

In the baroclinic wave and Held-Suarez tests, a function form for stretching the vertical element size is similar to that used in Ullrich and Jablonowski (2012, JCP). We calculate the vertical coordinate $\zeta$ at the top element boundary of $k'$-th element as

$$\zeta_{k'+1/2} = z_T \frac{1}{\sqrt{b+1}-1}\left[\sqrt{b\,(\frac{k'}{N_{e,z}})^2 + 1} - 1\right]$$

where $b$ is a positive parameter and $N_{e,z}$ is the number of elements in the vertical direction. As $b$ decreases, the vertical element size near the surface becomes small compared to the upper domain of the model. In this test case, we set $b=20$ for $p=3$, 7 and $b=5$ for $p=11$. In Sect. 3.4, we have described it briefly as

"We used a stretched vertical grid based on Eq. (102) in Ullrich and Jablonowski (2012). The stretching parameter was set such that the vertical grid spacing near the surface $\Delta_v$ took values of ..."

in lines 512-514 of the revised manuscript. For the detail on the stretching strategy, we have described it in Appendix C of the revised manuscript.

[Reference]

- Ullrich, P. A., & Jablonowski, C. (2021): MCore: A non-hydrostatic atmospheric dynamical core utilizing high-order finite-volume methods. *Journal of Computational Physics*, 231(15), 5078-5108.. https://doi.org/10.1016/j.jcp.2012.04.024

**45. line 456: "about 350m" - why about?**

Thank you for your suggestion. In the previous manuscript, we wanted to write "about 350 m" as an averaged vertical grid size near the surface for $p$=3, 7, 11 (We apologize that there is a slight error in this value). In the revised manuscript, we decided to remove "about". To do so, we have modified the corresponding statement as

"… such that the vertical grid spacing $\Delta_v$ near the surface took values of 305 m, 523 m, and 426 m for $p = 3$, 7, and 11, respectively."

in lines 512-514 of the revised manuscript.

**46. line 457: "against the" - same comment as before - is this the acoustic Courant number?**

Yes, we mean the acoustic Courant number. In line 515 of the revised manuscript, we have modified the statement as

"… we set the timesteps such that $C_{r,c_s} = 1.68 \times 10^{-1}$ for p=3, 7 and $C_{r,c_s} = 1.26 \times 10^{-1}$ for p=11."

**47. line 473: "for" should be "of the"**

We have modified it in line 530 of the revised manuscript.

**48. line 475: "cancellation of vertical errors" - I don't understand why this happens - is it really that clean? Can you explain more?**

This investigation focused on the numerical convergence associated with the horizontal resolution. For this purpose, we left the vertical DOF unchanged while increasing the horizontal resolution for each polynomial order $p$. Thus, we expected that the vertical discretization errors remain almost identical in the experiments using different horizontal resolutions at the same $p$. If we regarded the highest resolution experiment using the same $p$ as the reference experiment when evaluating the $L_2$ error, the contribution of vertical discretization errors would virtually cancel out.

Based on your comment, we think that it is better to move the explanation of this cancellation in the second paragraph of Sect. 3.4 into the description about the $L_2$ errors. In lines 533-534 of the revised manuscript, we have modified the corresponding statement as

"This is because the vertical spatial errors have similar values among different horizontal resolution cases with the same $p$ and these errors virtually cancel out when the $L_2$ error is evaluated."

49. line 478: "sufficiently small compared to... for example" - you cite a specific example here so could you also give the magnitude in that specific example so the reader can compare themselves?

We agree with your suggestion. In lines 536-537 of the revised manuscript, we have mentioned the magnitude as

"For example, in the horizontal grid spacing of 50 km (0.5 degrees), the $L_2$ error was $1 \times 10^{-2}$ hPa for the FV dynamical core and $5 \times 10^{-3}$ hPa for Mcore (Ullrich and Jablonowski, 2012b)."

50. Figure 7: Make it clear in the caption that the top plot is the "reference" solution, or order the plots so that the resolution is increasing either moving upwards or downwards - at the moment it is confusing comparing the different figures.

Thank you for your suggestion. In Fig. 8 of the revised manuscript, we have reordered the panels such that the spatial resolution increases as we move downward.

51. line 496: "against the" - same comment as before, do you mean the acoustic Courant number?

Yes, we mean the acoustic Courant number. In lines 554-555 of the revised manuscript, we have modified the statement as

"… the acoustic Courant number of $C_{r,c_s} = 1.3 \times 10^{-1}$ for $p$=3,7 and $C_{r,c_s} = 7.56 \times 10^{-1}$ for $p$=11."

52. line 500: say that these are averaged fields.

We have added "zonally and temporally averaged" in line 558 of the revised manuscript.

53. line 501: "using nearly spatial resolution" - do you mean "nearly the same spatial resolution"? Please clarify.

Thank you for suggesting a better representation. We would like to say that the spatial distribution show in Fig. 9 is similar to that obtained from the numerical experiments with the horizontal grid spacing of ~200 km in the previous studies.

We have modified this sentence based on your suggestion as "nearly the same horizontal spatial resolution." in lines 559-560 of the revised manuscript.

54. line 509 and 510: "resolutions" should be "resolution" in both cases

We have corrected it in lines 568 and 569 of the revised manuscript.

55. line 510: "about 50km" - why about?

This is because we worried about the convergence behavior in the eddy heat flux and eddy momentum flux. However, since we specified the convergence for eddy temperature variance and eddy kinetic energy in this statement, we think the use of "about" is unnecessary. Thus, we have removed it in the revised manuscript.

56. line 513: "by" should be "using"

We have modified it in line 572 of the revised manuscript.

57. Figure 9: "averaging" should be "averaged"

We have replaced "averaging" by "averaged" in the caption of Fig. 10 of the revised manuscript.

58. line 523: "about 10-20 grids" - what does this mean? grid cells rather than grids?

We would like to mean the spectra for $p$=3 overlap with that of the reference experiment at a wavelength range longer than $10\,\Delta_{h,eq}$~$20\,\Delta_{h,eq}$. Using your suggesting term "grid lengths" before, we have modified the sentence as

"... at a wavelength range longer than 10~20 grid length."

in lines 582-583 of the revised manuscript.

59. line 524: "Thus, ..." Could you clarify this sentence? If you replaced "in relatively small polynomial order" with "when using lower polynomial order", would that say what you mean?

Thank you for your suggestion. We would like to mention that the strength of modal filters should be carefully set in $p \leq 3$ based on the energy spectra obtained from the Held-Suarez test. In line 585 of the revised manuscript, we have replaced the corresponding sentence by "when using $p \leq 3$."

60. line 540: "with 200" should be "of 200"

We have replaced "with" by "of" in line 599 of the revised manuscript.

61. Figure 12: What is the polynomial order and resolution for these results?

Thank you for pointing out that we should mention the polynomial order and spatial resolution in the caption. The figure was drawn using the results for $\Delta_{h,eq}$= 10 m using $p$=7. In Fig 13 of the revised manuscript, we have added the following statement:

"in the LES of an idealized planetary boundary layer turbulence for the case of $\Delta_{h,eq}$= 10 m using $p$=7".

62. line 542: What is the form of the sponge layer? Refer to appendix A2 again.

We used a sponge layer based on a half-cosine function. In the sponge layer, the vertical wind was decayed by the Rayleigh damping with an e-folding time of 10 s at the model top. Because Appendix A.2 of the previous manuscript described the sponge layer used in the mountain wave, we have added a new statement in lines 602-603 of the revised manuscript as

"… using a sponge layer, where the vertical wind was decayed by the Rayleigh damping. The *e*-folding time varied as the half cosine function from zero at $z$ = 2km to 10 s at the model top."

63. line 547: Do you mean the acoustic Courant number? Why "about" 0.438?

Yes, we are referring to the acoustic Courant number. By using the Courant number defined in Sect. 2.4 of the revised manuscript, we can remove "about".

In line 608 of the revised manuscript, we have slightly modified the corresponding statement as "We set the timesteps such that $C_{r,c_s} = 4.38 \times 10^{-1}$."

64. line 558: "eight grids" do you mean "eight grid cells"?

In line 619 of the revised manuscript, we have modified it as "eight grid lengths" to be consistent to our response of your comment for the line 523 of the original manuscript.

65. line 560: "required polynomial order is p>3" - required for what?; "which is true for results obtained in this study" - make it clear you have shown this by describing the difference for p=3.

[Figure]

Fig. R3: Kinetic energy spectra normalized by the result of *p*=7 in LES of planetary boundary layer turbulence.

We would like to mean that p>3 is necessary to ensure that the effect of numerical diffusion term is sufficiently small compared to that of the SGS eddy viscosity term at the wavelength longer than eight grid lengths. We agree that the difference among *p*=3, 4, 7 needs to be shown more clearly. Accordingly, we determined the energy spectra normalized by the result of *p*=7 as Fig. R3. In the revised manuscript, we have showed such figures in Figs. 15(b) and C3(b) instead of the spectra normalized by the -5/3 power law. In addition, in lines 621-623 of the revised manuscript, we have modified the corresponding statements as

"… *p*>3 is required that the effect of numerical diffusion term is sufficiently small compared to that of the SGS eddy viscosity term at the wavelength longer than eight grid length. This is true for global LES as shown in Fig. 15(b)."

66. Figure 13: "averaging" should be "averaged"; "variable" should be "variance"

Thank you for pointing out our mistake. We have corrected it in the revised manuscript.

**Conclusions:**

67. line 563: "our previous studies" - cite specifically which ones.

In line 625 of the revised manuscript, we have cited Kawai and Tomita (2021, MWR).

68. line 567: "considering advantages" - what does this mean? do you mean that the DGM method has advantages? I don't think "considering" is the right word.

Thank you for pointing out the unclear statement. As you indicated, we wanted to mean that we focus on DGM because it has several advantages including a simple strategy for high-order discretization and higher computational efficiency in parallel computers. In line 629 of the revised manuscript, we have modified the statement as

"… because DGM has several advantages over grid-point methods, including …"

69. lines 593, 594 and 600: "grids" should, I think, be "grid cells"

Thank you for your suggestion. In lines 655, 656, and 663 of the revised manuscript, we have changed it as "grid lengths" for consistency with the corresponding modification.

**Code and data availability:**

70. "crate" should be create

We have corrected it in line 685 of the revised manuscript.

71. Could the data go on GitHub LFS?

We appreciate your suggestion. We understand that it is better that the output data would be in a public repository. However, the file size significantly exceeds the storage limit of GitHub LFS which is 1 GiB for a free account.

**Appendix A:**

We would like to inform you that Appendix A has become Appendix B in the revised manuscript because new appendices were added.

72. line 627: Can you cite the previous studies that did this?

In line 717 of the revised manuscript, we have cited two papers, Durran (1986) and Sachsperger

et al. (2016).

[Reference]

Durran, D. R., 1986: Another Look at Downslope Windstorms. Part I: The Development of Analogs to Supercritical Flow in an Infinitely Deep, Continuously Stratified Fluid. *Journal of the Atmospheric Sciences*, **43**, 2527–2543, https://doi.org/10.1175/1520-0469(1986)043<2527:ALADWP>2.0.CO;2.

Sachsperger, J., Serafin, S. & Grubišić, V. (2016), Dynamics of rotor formation in uniformly stratified two-dimensional flow over a mountain. *Quarterly Journal of the Royal Meteorological Society*, **142**: 1201-1212. https://doi.org/10.1002/qj.2746

73. line 638: Say that alpha is given below.

In line 730 of the revised manuscript, I have added the sentence as "… which is provided in this subsection."

74. line 670: why does this equation go on to the next line and what is the dot at the start of line 671 for?

In the original manuscript, we broke the line in the middle of the equation to make it easier to distinguish between the longitudinal and latitudinal dependence of the coefficient $\alpha_{s,h}$. The dot means the sign of multiplication. Because we felt that the unnecessary line break could cause confusion, we have removed the line break in the revised manuscript.

**Appendix B:**

We would like to inform you that Appendix B has become Appendix D in the revised manuscript because new appendices were added.

75. line 730: "inertia subrange" should be "inertial subrange"

We have modified it in line 826 of the revised manuscript.

**References:**

76. Check formatting - some titles are in all caps.

In the revised manuscript, we have modified the titles in all capital letters.

**Further notable modifications made in the revised manuscript**

- In the linear advection test in Sect 3.1 of the previous manuscript, the axis angles with the solid-body rotation flow $\varphi_0$ were incorrectly set for the cases of $\varphi_0=\pi/4, \pi/2$. Using the results

obtained from the modified experiment, we have replaced Fig. 1 in the revised manuscript. No qualitative changes were observed although the numerical errors for $\varphi_0=\pi/2$ become identical to that for $\varphi_0=0$. Thus, we have left most descriptions of the results unchanged.

- We added a new figure, Fig. 2 in the revised manuscript, which shows the impact of the modal filters on the numerical convergence in the linear advection test in Sect 3.1.

- In the Held-Suarez test, we adjusted the vertical element size for the case of $\Delta_{h,eq}$=280 km using $p$=7 for the vertical stretching to be consistent to all cases using $p$=7 and corrected the vertical grids for the case of $\Delta_{h,eq}$=52 km using $p$=3. In addition, we extended the temporal integration of the highest resolution case to 1000 days based on the suggestion of a reviewer. In Figs. 10, 11, and 12 of the revised manuscript, we have presented the results of the modified experiments. Fortunately, because no significant changes were observed, the claims made in the previous manuscript were left unchanged.

- We added a new appendix, Appendix A. Here, following the suggestions of a reviewer, we have described the results of a linear advection test, Case 4 presented in Nair and Lauritzen (2010).

---

## Author Comment (AC3)

**Response to reviewer 3 (egusphere-2024-1477 manuscript)**

Thank you for your careful review. We appreciate your thoughtful comments to improve our paper. We copied your comments in the blue text and have provided our responses in the black text. We have revised the manuscript according to your suggestions. Our point-by-point responses to the reviewer's comments are provided below. We hope that these improvements satisfactorily address the issues pointed out by you.

**General comments**

1. The authors make the claim that for global LES modeling (100km grid spacing), high-order DG methods will be important in this context. I dont object to this argument (and dont request any changes), but I will mention that I dont find the arguments persuasive.   If the arguments are correct, I think DG methods would be more common in regional models, which often run in the LES regime.

We agree that currently there are less studies of atmospheric LES using DGM. This may be because the numerical behavior was not well investigated in the LES regime. However, recent studies (Sridhar et al., 2022; Kawai and Tomita, 2023; Souza et al., 2024) indicate the possibility of DG dynamical cores to the atmospheric LES. Furthermore, in the CFD community, DGM seems to be regarded as a promising method for turbulent simulations using explicit and implicit LES in terms of high-order accuracy, flexibility of complex geometry, and scalability of parallel computations. These features would be benefit for the future high-resolution atmospheric simulations with O(10-100 m) grid spacing where the complex structure of small-scale topography need to be treated. Thus, we expect that global dynamical cores based on the high-order element-based method will become more common in the LES regime.

[References]

- Kawai, Y. & Tomita, H. (2023): Numerical Accuracy Necessary for Large-Eddy Simulation of Planetary Boundary Layer Turbulence Using the Discontinuous Galerkin Method. *Monthly Weather Review*, 151(6), 1479-1508. https://doi.org/10.1175/MWR-D-22-0245.1

- Sridhar, A., Tissaoui, Y., Marras, S., Shen, Z., Kawczynski, C., Byrne, S., ... & Schneider, T. (2022). Large-eddy simulations with ClimateMachine v0.2.0: a new open-source code for atmospheric simulations on GPUs and CPUs. *Geoscientific Model Development*, 15(15), 6259-6284. https://doi.org/10.5194/gmd-15-6259-2022

- Souza, A. N., He, J., Bischoff, T., Waruszewski, M., Novak, L., Barra, V., ... & Schneider, T. (2023). The Flux-Differencing Discontinuous Galerkin Method Applied to an Idealized Fully Compressible Nonhydrostatic Dry Atmosphere. *Journal of Advances in Modeling Earth Systems*, 15(4),

e2022MS003527. https://doi.org/10.1029/2022MS003527

2. One issue not address in this paper is the timestep. DG methods with the values of p proposed here will be quite expensive.   A good comparison showing how expensive high order DG can be compared to finite volumes is given in Brdar et al, https://doi.org/10.1007/s00162-012-0264-z which compares the DG based DUNE model with the finite volume (operational weather forecast model), COSMO. See also my comment below in the conclusions about numerical efficiency.

Thank you for informing us about an important work, Bardar et al. (2011), who discussed the computational time to reach a given error tolerance for DG and conventional FV dynamical cores, the DUNE and the COSMO. First, please notice that the temporal scheme is quite different between the two dynamical cores. DUNE adopted a fully explicit Runge-Kutta (RK) method for the inviscid terms. On the other hand, the COSMO adopted a sophisticated time-splitting approach in which the slow processes are integrated with an explicit RK method, while the fast processes are integrated with a small timestep horizontally by a forward-backward scheme and vertically by an implicit Crank-Nicholson scheme. Thus, it is difficult to directly evaluate the computational overhead due to the timestep restriction with DGM. (We would like to emphasize that, as described in the conclusion of Bardar et al. (2011), the different treatment of temporal scheme was not the focus in their experiments. We think that their comparison of the behavior of numerical convergence between the two dynamical cores is very valuable.)

On the other hand, it is well known that the timestep restriction with the explicit Runge-Kutta DGM is more severe compared to that in the grid-point methods (e.g., Cockburn and Shu, 2001). When we use a polynomial order $p$ for the spatial discretization, an approximate allowable timestep for an explicit $p+1$ stage RK method with $p+1$ order is given in the form

$$\Delta t \leq \frac{1}{\lambda_{max}} \frac{h_e}{2p + 1}$$

where $p$ is the polynomial order, $h_e$ is the element size, $\lambda_{max}$ is the maximum eigenvalue of Jacobian matrix with the advection terms. This means that we need to set the time step for the DGM such that it is approximately smaller by a factor of 1/2 compared to the grid-point methods with an approximately same DOF, as you have pointed out. However, it is possible that the computational overhead can be ignored in several situations: i) the spatial errors for high-order DGM rapidly reach to a given error tolerance due to the fast numerical convergence compared to conventional low-order methods with totally second-order accuracy. By using a coarser grid, we can significantly reduce the computational cost in three-dimensional problems. Bardar et al. (2011) also pointed out such situation. ii) In massively parallel computations, we consider the situation where there is little DOF per computational node. For conventional high-order grid point methods, the communication of halo data can occupy most of the execution time.

To accelerate DG dynamical cores in other situations discussed above, we agree that relaxing the severe timestep restriction with DG is an important topic. In fact, previous studies have attempted to extend the allowable timestep, for example, by optimizing the stability region of RK methods (e.g., Jahdali et al., 2022) or by using co-volume grids (Warburton, 2008). However, we would like to leave it as a future work.

Based on your comment, we have mentioned the issue of the timestep restriction in lines 673-677 of the revised manuscript as

"Furthermore, a severe timestep restriction for explicit temporal schemes is one of the unsolved issues in high-order DGM. We expect that the computational overheads would be ignored in several cases; A coarser spatial resolution can be used due to the high-order numerical convergence or the small communication cost in DGM is taken advantage of. However, to accelerate DG dynamical cores in all situations, developing sophisticated temporal treatments is an important future work."

[References]

- Cockburn, B. & Shu, C. W. (2001). Runge–Kutta discontinuous Galerkin methods for convection-dominated problems. *Journal of scientific computing*, 16, 173-261. https://doi.org/10.1023/A:1012873910884
- Al Jahdali, R., Dalcin, L., Boukharfane, R., Nolasco, I. R., Keyes, D. E., & Parsani, M. (2022): Optimized explicit Runge–Kutta schemes for high-order collocated discontinuous Galerkin methods for compressible fluid dynamics. *Computers & Mathematics with Applications*, 118, 1-17. https://doi.org/10.1016/j.camwa.2022.05.006
- Warburton, T., & Hagstrom, T. (2008): Taming the CFL number for discontinuous Galerkin methods on structured meshes. *SIAM Journal on Numerical Analysis*, 46(6), 3151-3180. https://doi.org/10.1137/06067260

3. While reading the text, it was clear that the authors use different settings (timestepping, filtering, and Smagorinsky diffusion) for the different test cases.  This can be good practice during the development process in order to test specific characteristics of the dycore.  But it is also useful to present results with the dycore configured as it would be used in practice. As the authors mention in their conclusions, they have not yet run the model with realistic topography (which is well known to create a lot of problems with high order element methods), and thus the "operational" configuration of SCALE-DG, especially with regards to how much filtering/diffusion will ultimately be needed, may not be known. One suggestion would be to also include all test results with the same configuration used for the planetary boundary layer turbulence test.  If the authors consider that beyond the scope of this paper, I would request to

add a table summarizing all the settings used for each test.

[Figure]

Fig. R1: Impact of the order $p_\mathrm{m}$ and the coefficient $\alpha_\mathrm{m}$ in the modal filter on the numerical convergence in a linear advection test: (a) $p_\mathrm{m}$=64, (b) $p_\mathrm{m}$=32, (c) $p_\mathrm{m}$=16, and (d) $p_\mathrm{m}$=8. In each $p_\mathrm{m}$, we changed $\alpha_\mathrm{m}$ as 0 (without the filter), $10^{-3}$, $10^{-1}$, and $10^1$. Please note that the results for $\alpha_\mathrm{m}$=0 are identical to those obtained for $\varphi_0$=0 in Fig. 1 of our paper.

We agree that a series of numerical experiments using the turbulent model will provide useful information about numerical and physical dissipation mechanisms necessary for operational runs wherein the realistic topography is included. But we would like to leave comprehensive numerical experiments as a future work.

Instead, in a linear advection test, we discussed how much the strength of modal filter can degrade high-order numerical convergence in Sect. 3.1 of the revised manuscript. Fig. R1 shows the impact of order $p_m$ and the decay coefficient $\alpha_m$ in the modal filter on the numerical convergence. Based on these results, when the scale-selective strong modal filters which immediately remove two-grid scale structure, it is possible to decrease the original convergence rate by 1~3 for $p$=7, 11. For $p$=3, although the degradation of convergence rate appears less obvious, the errors without the modal filter were much larger. Thus, for $p$=3, the effect of the increased error due to the filter may be more pronounced in the representation of the flow fields. It is difficult to determine filter levels required for numerical stability in realistic simulations a priori because they depend on various factors including nonlinearity, spatial resolution, turbulence parametrization, and smoothing of topography. However, we expect that the information about the sensitivity of filters is useful for readers who want to be careful about how much the strong filters can contaminate the quality of flow fields represented by high-order dynamical cores.

In addition, we have summarized all settings of the dissipation mechanism for each test case in Table. 6 of the revised manuscript.

**Specific comments:**

4. line 36: (or line 55) "... some researchers have successfully developed global nonhydrostatic atmospheric dynamical cores based… element-based methods". The authors mention some research codes, but neglect recent and larger efforts using high-order element based methods from major modeling centers.   These include E3SM: ( Caldwell et al., JAMES 2021 e2021MS002544, Donahue et al, JAMES 2024 e2024MS004314 ), the Korean KIM model (Hong et al, 2018, https://link.springer.com/article/10.1007/s13143-018-0028-9),        and       NRL's     NEPTUNE NEPTUNE Model, Kelly et al, 2024, https://arxiv.org/abs/2405.06076.

Thank you very much for informing us that we missed several important works with global nonhydrostatic dynamical cores based on the element-based method. In the introduction, we should refer to HOMME-NH and a spectral-element nonhydrostatic dynamical core in Korean Integrated Model. Based on other reviewer's comment, we have mentioned NUMA in the revised manuscript. Thus, we would like to referred to NEPTUNE which utilizes and extends the numerical methods prototyped in NUMA.

In lines 68-75 of the revised manuscript, we have added new statements as

"In the Nonhydrostatic Unified Model of the Atmosphere (NUMA; Kelly and Giraldo, 2012; Giraldo et al., 2013), which is applicable for both limited-area and global atmospheric simulations, the continuous and discontinuous Galerkin methods are adopted for the spatial discretization. The numeric prototyped in the NUMA is utilized and extended to a global spectral-element dynamical core in the Navy Environmental Prediction System Utilizing a Nonhydrostatic Engine (NEPTUNE) for both horizontal and vertical discretization (e.g., Zaron et al., 2022). SEM is also used for the nonhydrostatic High Order Method Modeling Environment (HOMME-NH; Dennis et al., 2005, 2012; Taylor et al., 2020) included in the Energy Exascale Earth System Model (E3SM), and for the nonhydrostatic dynamical core in the Korean Integrated Model (KIM) system (Hong et al., 2018)."

[References]

- Dennis, J., Fournier, A., Spotz, W. F., St-Cyr, A., Taylor, M. A., Thomas, S. J., & Tufo, H. (2005): High-resolution mesh convergence properties and parallel efficiency of a spectral element atmospheric dynamical core. *The International Journal of High Performance Computing Applications*, *19*(3), 225-235. https://doi.org/10.1177/1094342005056108

- Dennis, J. M., Edwards, J., Evans, K. J., Guba, O., Lauritzen, P. H., Mirin, A. A., ... & Worley, P. H. (2012): CAM-SE: A scalable spectral element dynamical core for the Community Atmosphere Model. *The International Journal of High Performance Computing Applications*, *26*(1), 74-89. https://doi.org/10.1177/1094342011428142

- Giraldo, F. X., Kelly, J. F., & Constantinescu, E. M. (2013): Implicit-explicit formulations of a three-dimensional nonhydrostatic unified model of the atmosphere (NUMA). *SIAM Journal on Scientific Computing*, *35*(5), B1162-B1194. https://doi.org/10.1137/120876034

- Hong, S. Y., Kwon, Y. C., Kim, T. H., Esther Kim, J. E., Choi, S. J., Kwon, I. H., ... & Kim, D. I. (2018): The Korean Integrated Model (KIM) system for global weather forecasting. *Asia-Pacific Journal of Atmospheric Sciences*, *54*, 267-292. https://doi.org/10.1007/s13143-018-0028-9

- Kelly, J. F., & Giraldo, F. X. (2012): Continuous and discontinuous Galerkin methods for a scalable three-dimensional nonhydrostatic atmospheric model: Limited-area mode. *Journal of Computational Physics*, 231(24), 7988-8008. https://doi.org/10.1016/j.jcp.2012.04.042

- Taylor, M. A., Guba, O., Steyer, A., Ullrich, P. A., Hall, D. M., & Eldred, C. (2020): An energy consistent discretization of the nonhydrostatic equations in primitive variables. *Journal of Advances in Modeling Earth Systems*, *12*(1), e2019MS001783. https://doi.org/10.1029/2019MS001783

- Zaron, E. D., Chua, B. S., Reinecke, P. A., Michalakes, J., Doyle, J. D., & Xu, L. (2022): The tangent-linear and adjoint models of the NEPTUNE dynamical core. *Tellus A: Dynamic Meteorology and Oceanography*, *74*(1). DOI: 10.16993/tellusa.146

5. line 75: "Few such studies for global nonhydrostatic dynamical cores are available although the numerical convergence characteristics of DGM was investigated for regional dynamical core..." For the citations of regional DG dynamical cores, the authors should also cite: Brdar et al, https://doi.org/10.1007/s00162-012-0264-z. This part of the introduction focuses only on three dimensional models, and gives the impression there is limited work on DG for global atmospheric modeling. There are quite a few papers looking at DG on the cubed-sphere grid for the shallow water equations, such as Nair MWR 2005, Ullrich GMD 2014, and the very recent entropy stable formulations: Ricardo et al, 2024,

https://www.sciencedirect.com/science/article/pii/S0021999124000123. I also think that the NUMA model from Giraldo et al. (cited in this text for their regional configuration) has a global version that runs both DG and CG, but I dont have a reference for that. A key model that needs to be mentioned is NEPTUNE, which is a global high order element based method that uses CG and DG, making it quite

similar to SCALE-DG. NEPTUNE is one of the few models I know that is using 3D higher order elements (as proposed here). (See NEPTUNE references in Kelly et al, 2024, https://arxiv.org/abs/2405.06076)

Thank you for your suggestion. As a previous study that investigated the numerical convergence with DGM in regional nonhydrostatic dynamical cores, we have added Bardar et al. (2013) in line 115 of the revised manuscript.

To mention previous studies with the element-based methods for the global shallow water equations, we have cited Nair (2005, MWR) and Ullrich (2014, GMD) in lines 64-66 of the revised manuscript as

"…developed global nonhydrostatic dynamical core based on high-order grid point and element-based methods. The essence of the numerical methods can be found in the horizontal discretization of the global shallow water equations; For example, Ullrich et al. (2010) for a high-order finite volume method (FVM), while Nair et al. (2005a) and Ullrich (2014) for high-order element-based methods."

In addition, we have cited Ricardo et al. (2024) in lines 79-80 of the revised manuscript as

"A similar method was successfully applied to a global shallow water model in Ricardo et al. (2024) and to a global nonhydrostatic dynamical core…"

We have added a new statement referring to NUMA in lines 68-71 of the revised manuscript as "In the Nonhydrostatic Unified Model of the Atmosphere (NUMA; Kelly and Giraldo, 2012; Giraldo et al., 2013), which is applicable for both limited-area and global atmospheric simulations, …"

As for NEPTUNE, we have mentioned it in lines 70-73 of the revised manuscript as

"The numerical method prototype used in NUMA is utilized and extended to a global spectral-element dynamical core in the Navy Environmental Prediction System Utilizing a Nonhydrostatic Engine (NEPTUNE) for both horizontal and vertical discretization (e.g., Zaron et al., 2022)."

[References]

- Brdar, S., Baldauf, M., Dedner, A., & Klöfkorn, R. (2013): Comparison of dynamical cores for NWP models: comparison of COSMO and Dune. *Theoretical and Computational Fluid Dynamics*, *27*, 453-472. https://doi.org/10.1007/s00162-012-0264-z
- Nair, R. D., Thomas, S. J., & Loft, R. D. (2005): A discontinuous Galerkin global shallow water model. *Monthly weather review*, *133*(4), 876-888. https://doi.org/10.1175/MWR2903.1
- Ricardo, K., Lee, D., & Duru, K. (2024): Conservation and stability in a discontinuous Galerkin method for the vector invariant spherical shallow water equations. *Journal of Computational Physics*, *500*, 112763. https://doi.org/10.1016/j.jcp.2024.112763
- Ullrich, P. A. (2014): A global finite-element shallow-water model supporting continuous and discontinuous elements. *Geoscientific Model Development*, *7*(6), 3017-3035. https://doi.org/10.5194/gmd-7-3017-2014

6. line 84: "However, they did not consider the vector Laplacian operator for the vector quantities (for 85 example, momentum). This might be because the rigorous form of vector Laplacian is so complex that it may not be worth the computational cost required numerical stabilization." The authors should note that vector viscosity for both DG and CG was developed in: Ullrich 2014, https://gmd.copernicus.org/articles/7/3017/2014/gmd-7-3017-2014.pdf as well (for CG) in Guba et al., https://gmd.copernicus.org/articles/7/2803/2014/gmd-7-2803-2014.pdf

Thank you very much for informing us about the references with vector Laplacian operator in the element-based methods. We agree with you that the two papers should be mentioned.

In lines 96-100 of the revised manuscript, we have added new statements as

"On the other hand, Ullrich (2014) presented a discretization strategy for the vector Laplacian operator with the continuous and discontinuous Galerkin methods. This approach can distinguish the divergence damping and vorticity damping with constant viscous coefficients. Guba et al. (2014) proposed a strategy of hyperviscosity with variable viscous coefficients in SEM where the vector Laplacian operator acts on the Cartesian component of the vector fields. For our purpose of introducing the turbulent model, ..."

[References]

- Guba, O., Taylor, M. A., Ullrich, P. A., Overfelt, J. R., & Levy, M. N. (2014). The spectral element

method (SEM) on variable-resolution grids: Evaluating grid sensitivity and resolution-aware numerical viscosity. *Geoscientific Model Development*, *7*(6), 2803-2816.
https://doi.org/10.5194/gmd-7-2803-2014

7. line 470 " In addition, the effective resolution is apparently higher than that of the low-order global dynamical core." See my general comment above. The authors are running different dissipation/fiter settings for each test case. As the authors mention, SCALE-DG needs stronger filtering when running more realistic test cases presented later. It is also very likely that even more dissipation will be needed when realistic topography is added. For the baroclinic instability test case, (as well as Held-Suarez) models are recommend to run with their operational diffusion

[Figure]

Fig. R2: Impact of the decay coefficient $\alpha_m$ of modal filter with $p_m$=16 on the surface pressure [hPa] at day 9 in the baroclinic wave test changed $\alpha_m$ as $10^2$, $10^1$, and $10^0$. This figure focused on the results where $\Delta_{h,eq}$=250, 125 km using p=7. For the comparison, we presented the results obtained from the FV dynamical core shown in Fig. 6 of Jablonowski and Williamson (2006) at the top panels. The lowest panels show our result obtained from $\Delta_{h,eq}$=31 km using $p$=7 as a reference solution.

settings, which is the case for the FV results. Thus, I would qualify this statement, and note that this might be due to the SCALE-DG's high order discretization, but it could also be due to using filtering levels that would not be practical in realistic problems.

We would like to inform you that high-order modal filters with large decay coefficients such as $p_{m,h}=16$ and $\alpha_{m,h}=O(1)$ for $p=7$ were used in the baroclinic wave and the Held-Suarez tests where small-scale flow structures develop. For the filtering levels, the flow structures at the short wavelength range near two grid scale are immediately dissipated after one timestep in HEVI temporal scheme. The total numerical dissipation of upwind numerical flux and modal filter near two grid scales is never weaker than the inherent numerical diffusion with the monotonic third-order piecewise parabolic method in the FV dynamical core or the explicit hyperdiffusion in the GME shown in Jablonowski and Williamson (2006).

To check the sensitivity of modal filter, we conducted additional experiments of the baroclinic wave test where the decay coefficient in the modal filters changed as $\alpha_{m,h}=10^0$, $10^1$, $10^2$ while $p_m$ was fixed to 16. Figure R2 shows the impact of the decay coefficient on the surface pressure at day 9. Based on Figs. R2(a)-(c), the development of low pressure systems weakened as the decay coefficient increased in the coarsest resolution of $\Delta_{h,eq}=250$ km. For the case of $\alpha_{m,h}=10^2$, the extent of numerical dissipation was comparable to that in the FV dynamical core. However, as shown Figs. R2(e)-(h), the sensitivity of decay coefficient in the modal filter was not significant for the amplitude and phase of high- and low-pressure systems with the increase in the spatial resolution. This is because we adopted the scale-selective filter with $p_m=16$.

It is unclear that how much we need to strengthen the filtering level to treat realistic steep terrain in high-order DGM. It must depend on how much the topography is coarsened compared to the inherent effective resolution with DGM. As a preliminary investigation, we recently conducted a Held-Suarez test with the realistic topography smoothed by approximately 4 grid lengths. Based on the numerical experiments, a stable long integration seems to be maintained if the decay coefficients of filter $\alpha_{m,h}$ used in the baroclinic wave and the original Held-Suarez test increase by a few factors of 2~4. If we assume the required decay coefficient is at most $\alpha_{m,h}=O(10)$ for $p_m=16$ when introducing the realistic topography, we expect such filter level will not fully change high-order DG solutions into low-order solutions based on Fig. R2.

As a future work, we need to further investigate how the strength of modal filters used in realistic atmospheric simulations (with complex surface geometry and the forcing of physics processes) can contaminate the quality of numerical solutions with the high-order DGM. Thank you for your comment.

9. line 567: "and high computational efficiency in recent parallel supercomputers, over grid-point methods." I doubt this statement is true - given the small timestep required by high order DG (see comment above). One might be able to make the case that the methods achieve higher FLOP counts, but most people would interpret computationally efficiency in terms of time-to-solution.

As mentioned in our reply to Comment 2, if the same DOF and temporal method are used, allowable timesteps for the DGM would be shorter by a factor of 2 compared to the grid-point methods. The overhead would be low in several situations where we can utilize the advantage of DGM associated with the high-order numerical convergence or small communication cost in massively parallel computers. In this case, to reach a given error tolerance, the time-to-solution for the DGM can be shorter than the conventional grid-point methods. However, we need to further investigate whether this holds true for various situations. Therefore, following your suggestion, we determined to describe a high FLOPS count in the DGM here. In line 630 of the revised manuscript, we have modified the statement as

"… the high floating-point operations per second (FLOPS) count in recent parallel supercomputers, over …"

10. Terminology: The authors use the phrase ""eight grids" and "10~20 grids" several times. It's clear what they mean, but this is an unusual phrasing and I think technically incorrect because they are referring to the grid spacing or grid cell width, not the grid itself. I'd suggest changing to $8 ¥Delta x$.

Thank you for suggesting an improvement of our terminology. We understood that the term "grids" is incorrect because what we would like to mean here is the wavelength corresponding to the several grid spacing. In the revised manuscript, we have determined to use "grid length" based on an advice of other reviewer.

---

## Referee Report (RR1)

**Second round of review:**

General comments:
The authors have significantly improved the manuscript and addressed most of the reviewers' comments. However, the scientific significance of the work and novelty compared to other DG dycores, in my opinion, remain the weak points of this paper.

Specific comments:
The authors have acknowledged and cited previous studies in their revised manuscript's Introduction and listed what they believe are their unique contributions (lines 92-116) in three key points. However, none of the three points is a real breakthrough. They mentioned other studies who have followed the same approaches before, without really emphasizing why their proposed approach should be better.

Also, it would be nice if the authors mentioned which configuration in terms of numerical settings and choice of dissipation mechanisms they plan to use in the final "operational" version of SCALE-DG, even if subject to change in the future.

Technical comments:
Line 100: Change "For introducing the turbulent model" with "To introduce the turbulent model"

---

## Author Response (AR2)

**Response to topic editor (egusphere-2024-1477 manuscript)**

Dear Dr. Sylwester Arabas,

Thank you for addressing reviewers' remarks and congratulations for receiving "Excellent" marks for Scientific quality from both Referees.
I am accepting the paper requesting the following technical corrections:
- please follow Referee #1's comments,
- please correct seven occurrences of bogus "https://doi.org/https://doi.org" URLs in references.

Thank you for providing your thoughtful comments. We also appreciate the Referee#1's review for improving our paper. Based on Referee#1's comments, we have modified the manuscript as mentioned below. In addition, we have corrected seven URLs in references.

**Referee#1's comments**

General comments:

The authors have significantly improved the manuscript and addressed most of the reviewers' comments. However, the scientific significance of the work and novelty compared to other DG dycores, in my opinion, remain the weak points of this paper.

We are grateful to notice the improvements compared to the previous manuscript. In the sight of numerical aspect of atmospheric dynamical cores toward global LES, we consider that completing our model description paper of DG dynamical core is an important step following global simulations with sub-kilometer horizontal spacing, such as Miyamoto et al. (2013, GRL). By building on progresses from previous studies with the element-based methods, we have constructed a global dynamical core based on DGM for LES. By quantitatively investigating numerical performance through various test cases, we have advanced the fundamental understanding of numerical behavior of global nonhydrostatic dynamic cores using conventional DGM for both horizontal and vertical discretization. Based on the findings and using a set of simulation codes developed here, we will attempt to improve DGM to optimize for atmospheric flows also considering dynamics-physics coupling strategies. We would like to enhance the uniqueness of our model by such research activities.

Specific comments:

The authors have acknowledged and cited previous studies in their revised manuscript's Introduction and listed what they believe are their unique contributions (lines 92-116) in three key points. However, none of the three points is a real breakthrough. They mentioned other studies who have followed the same approaches before, without really emphasizing why their proposed approach should be better.

Although we appreciate and understand most of reviewer#1's opinions, we believe there are certain contributions as we mentioned above.

We focus on the first key point, which mentioned numerical approaches following several previous studies, since the second and third key points in lines 107-116 of the previous manuscript mentioned the modification of experimental setup to evaluate high-order numerical convergence and the investigation of numerical performance of global DG dynamical core, respectively. We agree that it is better to add information on why the approaches are necessary when introducing a turbulent model to our global DG dynamical core. In lines 100-104 of the revised manuscript, we have modified the corresponding statements as

"However, we did not directly use these approaches to the vector Laplacian when introducing the turbulent model. This is because we need to treat eddy viscous and diffusion coefficients dependent on local wind shear and stratification. In addition, we consider that the vector Laplacian operator applied to the vector component on the cubed-sphere coordinates can be convenient for straightforward distinction between horizontal and vertical directions. Using tensor analysis, …".

Also, it would be nice if the authors mentioned which configuration in terms of numerical settings and choice of dissipation mechanisms they plan to use in the final "operational" version of SCALE-DG, even if subject to change in the future.

We agree that it is better to mention possible numerical setting and choice of dissipation mechanism in operational runs. To stabilize realistic simulations, we may need to increase the strength of modal filters, as discussed in our response to other reviewer's comment. However, it is difficult to generalize the strength of dissipation mechanisms because it depends on the extent of smoothing the topography and the dissipation with turbulent models. Thus, in lines 685-687 of the revised manuscript, we have added statements as

"For actual operational runs including physical processes and other factors, such as realistic topography, the degree of numerical filters depends on situations and cannot be generalized now. It is an important issue to produce a kind of criterion for the numerical stabilization."

**Technical comments:**

Line 100: Change "For introducing the turbulent model" with "To introduce the turbulent model"

Because we made the modification mentioned above, we have changed the corresponding statement as "when introducing the turbulent model".